# SciPredict: Can LLMs Predict the Outcomes of Scientific Experiments in Natural Sciences?

**Udari Madhushani Sehwag** [1]   **Elaine Lau** [† 1]   **Haniyeh Ehsani Oskouie** [2 3]   **Shayan Shabihi** [4]   **Erich Liang** [5 3]
**Andrea Toledo** [1]   **Guillermo Mangialardi** [1]   **Sergio Fonrouge** [1]   **Ed-Yeremai Hernández Cardona** [1]   **Paula Vergara** [1]
**Utkarsh Tyagi** [1]   **Chen Bo Calvin Zhang** [1]   **Pavi Bhatter** [† 1]   **Nicholas Johnson** [1]   **Furong Huang** [4]
**Ernesto Gabriel Hernández Montoya** [1]   **Bing Liu** [† 1]

## Abstract

Accelerating scientific discovery requires the identification of which experiments would yield the best outcomes before committing resources to costly physical validation. While existing benchmarks evaluate LLMs on scientific knowledge and reasoning, their ability to predict experimental outcomes—a task where AI could significantly exceed human capabilities —remains largely underexplored. We introduce SciPredict, a benchmark comprising 405 tasks derived from recent empirical studies in 33 specialized sub-fields of physics, biology, and chemistry. SciPredict addresses two critical questions: (a) *can LLMs predict the outcome of scientific experiments with sufficient accuracy?* and (b) *can such predictions be reliably used in the scientific research process?* Evaluations reveal fundamental limitations on both fronts. Model accuracies are 14-26% and human expert performance is $\approx$20%. Although some frontier models exceed human performance model accuracy is still far below what would enable reliable experimental guidance. Even within the limited performance, models fail to distinguish reliable predictions from unreliable ones, achieving only $\approx$20% accuracy regardless of their confidence or whether they judge outcomes as predictable without physical experimentation. Human experts, in contrast, demonstrate strong calibration: their accuracy increases from $\approx$5% to $\approx$80% as they deem outcomes more predictable without conducting the experiment. SciPredict

establishes a rigorous framework demonstrating that superhuman performance in experimental science requires not just better predictions, but better awareness of prediction reliability. For reproducibility all our data and code are provided at https://github.com/scaleapi/scipredict.

## 1. Introduction

Reasoning deeply about the expected outcome of experiments before running them is central to scientific progress. Researchers routinely make such predictions, deciding which hypotheses to test and parameter regimes to pursue under resource constraints. A system that could reliably predict the experimental results would reshape the scientific process, accelerating discovery by filtering out suboptimal directions, identifying gaps in current frameworks, and suggesting much needed empirical investigations. LLMs appear well-suited for this task (illustrated in Fig. 2), as they encode vast scientific knowledge (Taylor et al., 2022), can reason about complex systems, and demonstrate strong performance on scientific question-answering tasks (Wang et al., 2025).

Due to the lack of comprehensive benchmarks, the progress toward improving the ability of LLMs to predict the outcomes of practical experiments has been slow. Among benchmarks that explore the use of LLMs to aid the scientific research, most focus on areas such as literature review and paper composition/drafting (Li et al., 2025; Xu et al., 2025; Laurent et al., 2024), reproducing methods and computational simulation results (Starace et al., 2025; Zhao et al., 2025; Yan et al., 2025; Lin et al., 2025; Ali-Dib & Menou, 2024; Shojaee et al., 2025; Xia et al., 2025), or generating hypotheses for scientific experiments (Yang et al., 2025; Ke et al., 2025; Abdel-Rehim et al., 2025).

To address this gap, we introduce SciPredict, a benchmark designed to evaluate the capabilities of LLMs in predicting the outcomes of empirical experiments in natural sciences. We extract tasks from recently published empirical studies,

† *Work done while at Scale AI.*

[1]Scale AI [2]University of California, Los Angeles [3]Human Frontier Collective, Scale AI [4]University of Maryland [5]Princeton University. Correspondence to: Udari Madhushani Sehwag <udari.sehwag@scale.com>.

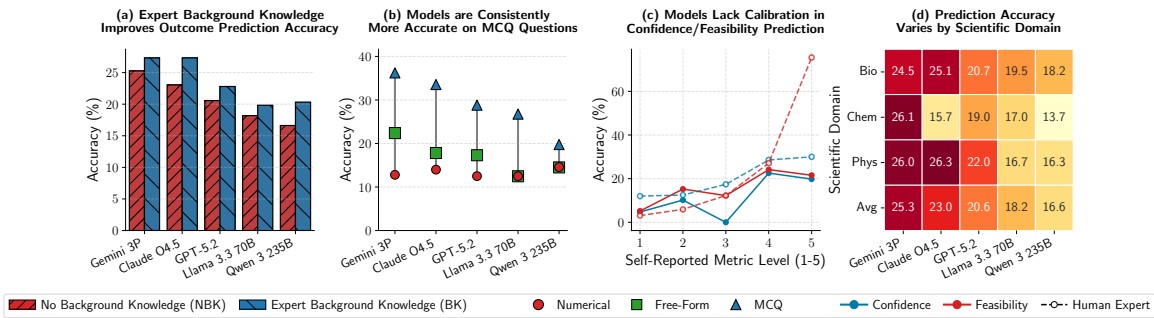

Figure 1. **Key findings of SciPredict.** Frontier models exhibit fundamental gaps in accuracy and calibration robustness in scientific experiment outcome prediction. We highlight four key failure modes using a representative subset of SOTA models: Claude O4.5 (Claude Opus 4.5), OpenAI GPT-5.2, Gemini 3P (Gemini 3 Pro), Llama 3.3 (Meta Llama 3.3 70B), and Qwen 3 235B. (a) Providing expert-curated background knowledge (BK) consistently boosts performance over No Background Knowledge (NBK). (b) Accuracy generally degrades when moving from multiple-choice questions (MCQ) to questions requiring free-form answers (Free-Form) to Numerical value questions. (c) Unlike Human Experts (dashed lines), models show poor calibration in SciPredict tasks; the accuracy of the models' answers to tasks do *not* correlate with their self-reported Confidence and perceived task prediction Feasibility. (d) Model performance varies across different domains. The Avg field shown represents the weighted average of scores according to the number of questions per domain.

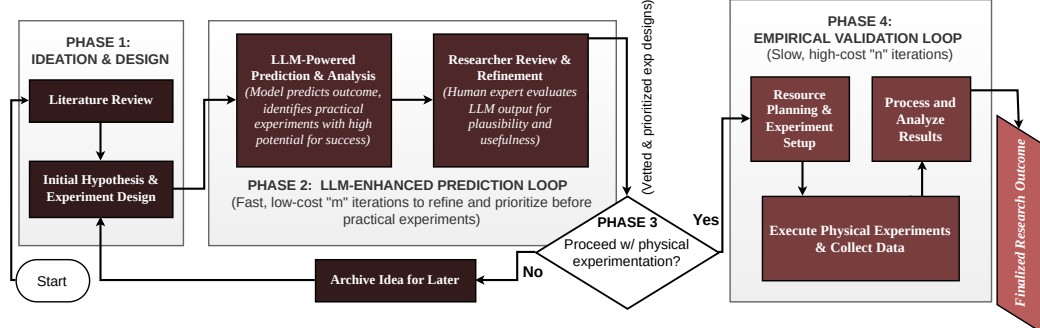

Figure 2. **LLM-enhanced efficient scientific research workflow.** The figure illustrates how LLM-powered experimental outcome prediction can be integrated into the scientific research process. Phase 1 involves ideation and experimental design through literature review and hypothesis formulation. Phase 2 represents a fast, low-cost prediction loop where LLMs predict experimental outcomes and identify high-potential experiments for physical validation, which researchers then review for plausibility. Based on this evaluation, researchers either proceed to Phase 3 (resource planning and experiment setup) and Phase 4 (empirical validation through physical experimentation), or archive the idea for later consideration. This workflow demonstrates how reliable LLM predictions could accelerate scientific discovery by filtering suboptimal experimental directions before committing to costly empirical validation.

from post-March 31, 2025, postdating the cutoff dates of frontier models. SciPredict is comprised of 405 experimental prediction tasks, spanning 33 specialized sub-fields: 9 under physics, 10 under chemistry, and 14 under biology. For each task, domain-expert human annotators extract relevant information, including experimental setups, measurements taken by the research team, empirical results, etc. from the target publications along with the relevant background knowledge from prior literature. Prediction questions come in three possible formats of multiple-choice (MCQ), free-format (FF), or numerical value (NUM) depending on the specific task. This variation allows us to effectively capture the different aspects of models' capabilities in scientific reasoning. For free-format questions SciPredict includes 1-10 expert-written rubrics used to judge the accuracy of provided predictions. For MCQ and NUM questions, respectively, the correct choice(s) and acceptable numerical

ranges are provided as the ground-truth labels. Each task underwent a multi-stage expert review process. The curation process overall costed $336k and 7,380 human expert hours, reflecting the difficulty of constructing a high-quality benchmark for experimental outcome prediction.

Our findings show that SOTA LLMs achieve prediction accuracy between $14\%$-$26\%$ while human experts achieve $\approx 20\%$. Although exceeding human performance, these accuracy levels remain insufficient for reliable experimental planning. In practice, the *reliability* of the outcome prediction process is crucial because researchers want to invest their limited resources in sufficiently compelling experimental directions. To account for this, we require the models and human experts to provide prediction *feasibility* scores along with their predictions, measuring whether the targeted outcomes are perceived to be reliably predictable given the contextual information (e.g., experimental setup,

background information), without physically conducting the experiments. Models show poor calibration of such scores with their measured prediction accuracy: their accuracy does *not* meaningfully improve with higher self-reported feasibility scores. Human experts, on the other hand, demonstrate *strong* calibration of their prediction accuracy and their rated prediction feasibility scores (increase in accuracy from $\approx 5\%$ to $\approx 80\%$ as rated feasibility rises).

To understand what types of prior scientific knowledge aids accurate outcome predictions, we used different variations of *background knowledge* in our evaluations. While expert-curated background knowledge (mainly extracted by experts from prior studies cited in the target publication) improved accuracy by $\approx 3\%$ on average ($1.2 - 5.8\%$ depending on the model), the models' self-generated background knowledge often resulted in accuracy degradation. Interestingly, even combining such self-generated background knowledge items with the expert-curated knowledge still yielded underperformance in most cases. We note that this pattern reveals a critical limitation: models struggle to identify what background information and prior scientific knowledge would be helpful for task outcome predictions, often introducing misleading assumptions or irrelevant details in their self-generated background knowledge that degrades accuracy. Fig. 1 summarizes some of our primary findings.

Our key contributions are summarized as follows:

- We introduce SciPredict, the first benchmark for evaluating LLMs in experimental outcome prediction tasks in natural sciences (biology, chemistry, physics). This dataset is comprised of 405 expert-curated tasks with three prediction question types (multiple-choice, free-form, and numerical) directly derived from empirical studies published after March 31, 2025, ensuring no data leakage from the model pre-training data.
- We conduct a comprehensive evaluation of 15 SOTA LLMs and human experts, analyzing the accuracy and reliability (confidence, difficulty, feasibility). We analyze the effectiveness of 4 types of relevant background knowledge being provided in context for effective predictions (expert-curated, self-generated, filtered, combined).
- We identify a critical calibration gap: unlike human experts who demonstrate strong calibration of their confidence/difficulty/feasibility ratings with their prediction accuracy, LLMs mostly do not show such meaningful correlations, making their deployment in real-world scientific experimentation pipelines untrustworthy.
- We demonstrate that the models benefit from expert-curated background knowledge provided in context for predictions, while they fail to generate such background knowledge autonomously. We also reveal primary causes for models prediction failures are due to factual and logical reasoning flaws rather than misunderstanding the task.

## 2. Related Works

**AI/ML research benchmarks.** Recent benchmarks have begun evaluating LLMs on tasks that simulate the AI research cycle itself, extending beyond problem-solving or knowledge recall. (Starace et al., 2025; Zhao et al., 2025; Yan et al., 2025; Lin et al., 2025) evaluate LLMs for their ability to reproduce masked or full code repositories and experiment results given existing ML papers. (Hua et al., 2025) takes this a step further by evaluating how well LLMs can write experiment code for novel research ideas not seen during training. (Huang et al., 2024; Jiang et al., 2025; Chan et al., 2025) evaluate agents on machine learning engineering tasks, assessing their ability to iteratively modify algorithms and improve performance across various datasets and tasks. (Li et al., 2025) focuses on research methodology, requiring LLMs to predict masked out methodological details of AI research papers. (Xu et al., 2025) evaluates LLM agents' ability to provide technical details, literature review, and open consulting to AI-related questions. (Chen et al., 2025; Zhang et al., 2025; Kon et al., 2025) extend evaluation to the end-to-end AI research cycle. Our benchmark explicitly focuses on assessing LLMs' ability to understand and predict empirical scientific outcomes.

**Non-ML scientific research benchmarks.** LLMs have also been evaluated for their performance on scientific research tasks outside of AI. For example, (Ali-Dib & Menou, 2024) assesses LLMs on coding and problem-solving tasks in computational physics. (Shojaee et al., 2025) uses LLMs, leveraging their extensive domain knowledge and reliable program synthesis, to infer scientific equations directly from datasets; extending this, (Xia et al., 2025) turns LLMs into autonomous scientists that code, evaluate, and iteratively optimize the simulated equations. Similarly, (Bersenev et al., 2024) provides LLM agents with written biology papers and evaluates their ability to reproduce the methodology, code, and results. (Laurent et al., 2024) tests LLMs on their ability to do literature review and data analysis for biology research questions. While these benchmarks are valuable for evaluating LLMs' abilities in problem-solving, coding, and scientific writing, they do not directly measure an LLM's capacity to predict empirical scientific outcomes. Work on outcome prediction has so far focused mainly on behavioral and social sciences. (Cui et al., 2024) and (Saynova et al., 2025) evaluate LLMs on predicting experimental outcomes or reproducibility, but they operate in domains where measurements are often less precise and quantitative. In contrast, our benchmark targets the hard sciences, emphasizing quantitative prediction of empirical results. (Lu et al., 2025) provides qualitative analysis of how well LLMs can answer theoretical physics questions. Compared to this work our paper provides a standardized benchmark for quantitative evaluation.

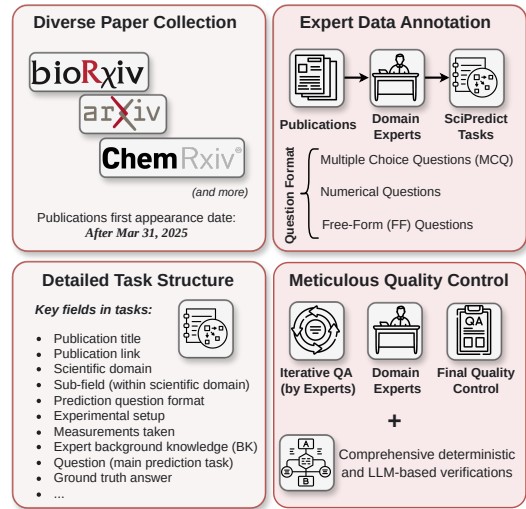

*Figure 3.* **SciPredict curation pipeline.** The benchmark construction involves four integrated stages: (Top-Left) **Data Collection** from preprint repositories ensuring a post-March 2025 cutoff to prevent data leakage; (Top-Right) **Expert Annotation** where domain specialists convert raw papers into MCQ, numerical, and free-form prediction tasks; (Bottom-Left) **Task Structure** enforcement, ensuring every sample includes granular fields such as experimental setup, measurements, and expert-curated background knowledge; and (Bottom-Right) **Quality Control/Assurance**, combining iterative expert review with deterministic and LLM-based verifications.

**LLM-driven scientific hypothesis generation.** While some benchmarks ask LLMs to generate hypotheses for scientific experiment settings, these works differ from our work in important ways. (Yang et al., 2025) provides a benchmark where LLMs have to produce and rank novel hypotheses in chemistry when prompted with background information and a set of hand-picked inspiration facts. (Ke et al., 2025) proposes a multi-agent framework that combines language-model reasoning with biomedical knowledge graphs and an automated literature retrieval engine to generate and iteratively refine grounded, novel hypotheses in biomedicine. (Abdel-Rehim et al., 2025) examines the applicability of large language models for hypothesis generation, focusing their experiments on breast cancer therapy. (Brunnsåker et al., 2025) introduces an LLM-driven approach to automating experimental design that fuses relational learning–generated hypotheses with real-world lab constraints and and is deployed on an automated cell and metabolomics platform.

## 3. SciPredict Curation

SciPredict consists of 405 prediction tasks derived from empirical studies published after March 2025 across physics, biology, and chemistry. The construction process balances competing requirements. Questions must be challenging enough to distinguish model capabilities yet tractable enough that expert-curated background knowledge could plausibly aid prediction. Experimental setups must be described with sufficient precision for informed reasoning without simply revealing the answer. Ground truth outcomes must be objectively verifiable while accounting for the inherent variability in empirical measurements.

### 3.1. Design Principles

Our focus domains are experimentally rich and empirical validation is central to knowledge creation. To evaluate scientific reasoning, we use three question formats—multiple-choice (MCQ), free-form, and numerical—to cover discrete, explanatory, and quantitative prediction. Domain-selection criteria, rubric designs, and evaluation specifics are detailed in Appendix A.2. Example data are given in Appendix B.1.

### 3.2. Data Collection

We recruited domain experts across biology, physics, and chemistry, representing diverse educational and geographic backgrounds (details in Appendix A.1). Experts selected papers published after March 31, 2025 ensuring tasks represent genuine predictive reasoning challenges. They extracted experimental setups, measurements, prediction questions, and ground-truth outcomes, and curated background knowledge relevant for informed reasoning. Full details of selection criteria and extraction methods are provided in Appendix A.3.

### 3.3. Quality Control

All tasks underwent rigorous multi-stage review. Initial screening removed ambiguous, simulated, theoretical, or outdated (pre-March 2025) tasks. Two rounds of domain experts verified the clarity and completeness of experimental details, background knowledge relevance, ground truth clarity, and task difficulty. Reviewers additionally ensured that MCQ distractors represented scientifically sound but incorrect alternatives, comprehensive yet flexible free-form evaluation rubrics, and realistic numerical precision ranges. See Appendix A.5 for full reviewer guidelines and checks.

### 3.4. Data Diversity

SciPredict spans 33 sub-fields of physics, biology, and chemistry. Task distribution across these domains are provided in Appendix A.7 Tab. 2. Tasks systematically vary in complexity, from single-step causal reasoning to complex multi-hop inference requiring advanced expertise. Background knowledge items range from undergraduate-level to very specialized (expertise held by active researchers). Task distribution ensures sufficient representation across domains (physics 25%, biology 50%, chemistry 25%) and question formats (MCQ 40%, free-form 32%, numerical 28%). See Appendix A.6 for additional details.

### 3.5. Human baseline

In addition to the experts who constructed the benchmark, we recruited a separate group of domain experts to provide a human baseline performance. Each expert answered benchmark questions, provided reasoning for their answers, and reliability scores. Mirroring the LLM evaluation, experts completed two rounds: first without, then with provided background knowledge (further details are provided in Appendix A.4). To ensure a high quality human baseline, we match experts to tasks based on their expertise. Additional details on the mapping and example human baseline responses are given in Appendix A.7 Tab. 1 and B.2.

## 4. Evaluation Setup and Metrics

Our dataset $\mathcal{D}$ contains 3 subsets for multiple-choice questions ($\mathcal{D}_{\text{MCQ}}$), free-form questions ($\mathcal{D}_{\text{FF}}$), and numerical questions ($\mathcal{D}_{\text{NUM}}$). For each task $i$, evaluated models $m \in \mathcal{M}$ provide a prediction $\hat{y}_i^{(m)}$ and 3 reliability assessments.

### 4.1. Accuracy Metrics

**Multiple-choice (MCQ).** Each question $i \in \mathcal{D}_{\text{MCQ}}$ presents 3-4 options with ground truth answer $g_i \subseteq \{A, B, C, D\}$ provided by domain expert annotators ($g_i$ is comprised of more than one choice in $12.35\%$ of the MCQ tasks only). Accuracy is the proportion of questions answered correctly:

$$\text{Acc}_{\text{MCQ}}^{(m)} = \frac{1}{|\mathcal{D}_{\text{MCQ}}|} \sum_{i \in \mathcal{D}_{\text{MCQ}}} \mathbb{1}[\hat{y}_i^{(m)} = g_i]. \quad (1)$$

**Free-form (FF).** Each question $i \in \mathcal{D}_{\text{FF}}$ has a reference answer $y_i$ and an expert-written evaluation rubric. We employ an LLM judge $J_\theta$ with a fixed prompt to assess whether the model's response $\hat{y}_i^{(m)}$ demonstrates correct scientific reasoning, as evaluated against the provided rubric:

$$s_i = J_\theta(\hat{y}_i^{(m)}, y_i) \in \{0, 1\}, \text{Acc}_{\text{FF}}^{(m)} = \frac{1}{|\mathcal{D}_{\text{FF}}|} \sum_{i \in \mathcal{D}_{\text{FF}}} s_i. \quad (2)$$

**Numerical value (NUM).** For each question $i \in \mathcal{D}_{\text{NUM}}$, domain experts specify an acceptable range $[L_i, U_i]$ accounting for measurement precision and experimental variability. Accuracy reflects whether predictions fall within this scientifically reasonable interval:

$$\text{Acc}_{\text{NUM}}^{(m)} = \frac{1}{|\mathcal{D}_{\text{NUM}}|} \sum_{i \in \mathcal{D}_{\text{NUM}}} \mathbb{1}[L_i \leq \hat{y}_i^{(m)} \leq U_i]. \quad (3)$$

This metric captures whether the model's quantitative prediction is sufficiently accurate for experimental planning, rather than demanding exact numerical matches.

### 4.2. Reliability Calibration

Reliable deployment in experimental science requires the ability to distinguish trustworthy predictions from unreliable ones. We assess reliability through three measures.

- **Confidence.** Models report confidence $\hat{c}_i^{(m)} \in \{1, \ldots, 5\}$ regarding their prediction's correctness. If well-calibrated, this metric is expected to *positively* correlate with the prediction accuracy ($\hat{c} \uparrow$ correlates with Acc $\uparrow$)
- **Difficulty.** Models' perceived task prediction hardness $\hat{z}_i^{(m)} \in \{1, \ldots, 5\}$ given the provided context. Difficulty assesses the self-awareness of models regarding their own prediction limitations. If well-calibrated, this metric is expected to *negatively* correlate with the prediction accuracy ($\hat{z} \uparrow$ correlates with Acc $\downarrow$).
- **Feasibility.** Models assess if an outcome can be predicted via reasoning without running the practical experiment ($\hat{f}_i^{(m)} \in \{1, \ldots, 5\}$). If well-calibrated, this metric is expected to *positively* correlate with the prediction accuracy ($\hat{f} \uparrow$ correlates with Acc $\uparrow$).

### 4.3. Experimental Conditions

To determine the information requirements for accurate predictions, we systematically vary the **context** provided to the model. Each task's BK in SciPredict is comprised of multiple atomic knowledge bullet points.

- **No Background Knowledge (NBK).** The context contains only the experimental setup, measurements, and the prediction question. This assesses whether the model's internal *parametric knowledge* is sufficient for prediction.
- **Background Knowledge (BK).** The context additionally includes expert-curated BK . This measures the performance gain when relevant, high-quality background information is explicitly surfaced in the context.
- **Self-generated Background (SBK).** The model is prompted to generate its own BK before predicting. This assesses the model's ability to autonomously identify and articulate the necessary scientific context.
- **Self-generated + Annotator Background (SABK).** The context includes both the expert-curated (BK) and self-generated background knowledge (SBK). This assesses whether combining such information sources provides additive benefits or introduces noise/interference.
- **Filtered Background Knowledge (FBK).** For each model, the context includes expert BK *minus* the facts the model already *knows*. We convert the BK items into questions and remove any BK items from the final prediction context where the model is able to answer the corresponding questions. This isolates whether stating known information in context improves prediction even when that information is theoretically accessible from parameters.

## 4.4. Evaluation Protocol and Robustness

Free-form predictions were evaluated by Gemini-3-Pro against expert rubrics. We validated the robustness of such evaluation pipeline by replicating evaluations using GPT-5.2 as well, where we found *no statistically significant* differences in accuracy scores. We also replicated predictions using various decoding strategies (temperature settings from 0.0 to 1.0, top-p sampling with $p \in \{0.9, 0.95, 1.0\}$. Performance variations remained statistically insignificant. Reported accuracy metrics represent means and with error bars indicate one standard deviation within 3 trials.

## 5. Main Results

We assess 15 SOTA LLMs analyzing if frontier models can predict experimental outcomes with sufficient accuracy and reliability for scientific use. All the prompts used in evaluations are given in Appendix D.

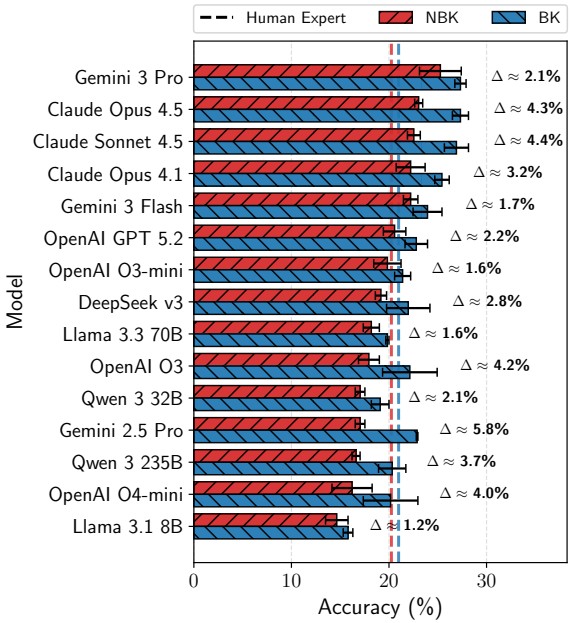

*Figure 4.* **Accuracy with and without background knowledge.** Accuracy (%) of each evaluated model under two input conditions: **(a) W/o background knowledge**: the model receives only the experimental setup, measurements, and the question; **(b) W/ background knowledge**: the same information as previous case with the addition of expert-curated background knowledge.

> **Finding #1:** Human performance is close to the average model performance.

We emphasize that expert human baseline performance serves as a calibration reference point, *not* an upper bound; the models can exceed human prediction capabilities by inte-

grating vast cross-domain knowledge and reasoning power. Human baseline ($\approx$20% accuracy; Fig. 4) reflects the inherent difficulty of predicting novel experimental outcomes without real-world scientific experimentation or validation.

> **Finding #2:** Providing curated background knowledge consistently improves the outcome prediction accuracy.

A key factor in answering the questions correctly, is access to relevant background knowledge. We test two conditions, model performance with and without background knowledge. As shown in Fig. 4, providing background knowledge improves accuracy $\text{Acc}^{(m)}$ across all models $m \in \mathcal{M}$, though the size of the increase varies by model. On average, BK improves accuracy by $\approx$3%. Curated background knowledge helps narrow the space of plausible outcomes. It is noted that confidence scores $\hat{c}^{(m)}$ remain roughly the same across NBK and BK, suggesting that background information primarily benefits correctness rather than improving confidence.

> **Finding #3:** Across nearly all models, accuracy is higher with the full annotator background than with the filtered version, implying that including knowledge the models *already know* still boosts performance.

Fig. 5 shows that restating known facts in the input context enhances model performance, even when those facts are *not* strictly missing from the models' parametric knowledge. By filtering the background knowledge—removing any expert background knowledge bullet points that the models already demonstrate knowledge of; see §Sec. 4.3—the x-axis approximates performance when the context contains only the "unknown" background knowledge. Most models fall in the upper triangle (above the y = x line), illustrating accuracy $\text{Acc}^{(m)}$ is higher when the full curated background is provided, including facts the model demonstrably knows. Additional results are given in Appendix C Tab. 3.

> **Finding #4:** Models cannot reliably generate useful background knowledge: self-generated background usually reduces accuracy. When combined with expert-curated background knowledge it rarely improves performance.

We evaluate settings where models self-generate background knowledge (SBK) and then answer, as well as a combined condition that appends this self-generated context to expert-curated background (SABK). Fig. 6 shows that, in contrast to the clear gains from expert-curated background knowledge, self-generated background is unreliable and often counterproductive: for most models, SBK lowers

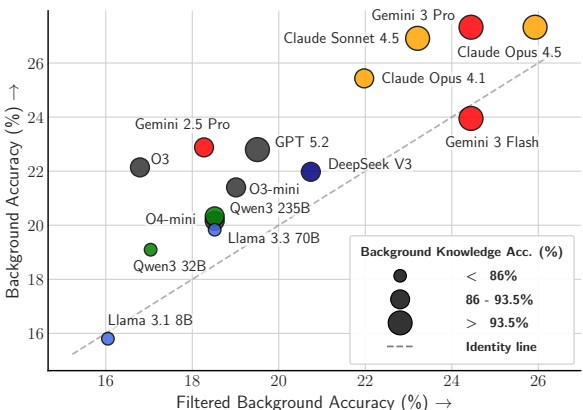

*Figure 5.* **Restating known facts in context enhances performance.** We present a scatter plot comparing model accuracy given full expert-curated background (BK, y-axis) versus a filtered version of such background (FBK, x-axis) in context (see definitions under §Sec. 4.3). Each point represents a model, colored by family, with marker size indicating the percentage of the tasks' background knowledge bullet points the model already knew ($> 70\%$ for all). Most models lie above the dashed identity line ($y = x$), showing that explicitly restating already known required background knowledge in the context yields higher accuracy than relying on originally learned knowledge alone.

accuracy compared to providing no background at all, implying that the generated content is frequently irrelevant or misleading and can steer predictions away from the correct experimental outcome. Moreover, supplementing expert-curated background knowledge with self-generated background (SABK) typically fails to yield consistent improvements, indicating that models struggle not only to generate helpful knowledge, but also to avoid introducing distracting or harmful information when additional context is available. Additional results are given in Appendix C Tab. 3.

> *Finding #5:* Self-assessed confidence/difficulty/feasibility by models are not aligned with accuracy, indicating calibration gaps. Humans, in contrast, show strong calibration of their rated confidence/difficulty/feasibility scores with their accuracy.

Fig. 7 demonstrates if models $m \in \mathcal{M}$ can reliably anticipate their own prediction errors by comparing accuracy $\text{Acc}^{(m)}$ to self-reported confidence $\hat{c}^{(m)}$, difficulty $\hat{z}^{(m)}$, and feasibility $\hat{f}^{(m)}$ ratings. While informative self-assessments would by expected to imply that $\text{Acc}^{(m)} \uparrow$ as $\hat{c}^{(m)} \uparrow$, $\hat{f}^{(m)} \uparrow$, and $\hat{z}^{(m)} \downarrow$, our results show that these relationships are very weak and non-monotonic for the models, which indicates substantial *miscalibration*. Conversely, human ratings of confidence, difficulty, and feasibility closely track accuracy in the expected directions (strong positive or negative correlation). Interestingly, as the bottom-left subfigure of Fig. 7 shows, models systematically achieve

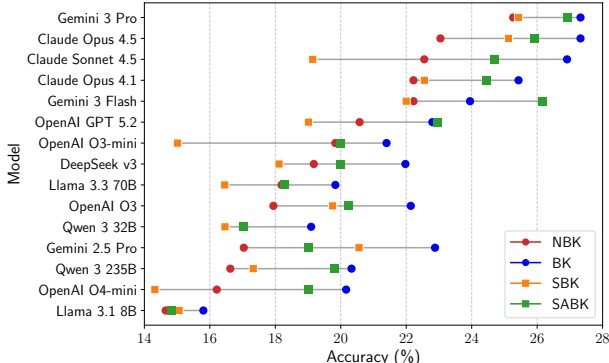

*Figure 6.* **Human vs self-generated background knowledge.** Evaluated accuracy (%) for the models under the four prediction conditions defined in §Sec. 4.3. *BK* generally yields the highest accuracy, while *SBK* frequently degrades accuracy relative to *NBK*, indicating that models fail to reliably generate useful predictive context. Furthermore, *SABK* rarely improves upon *BK*, suggesting that adding synthetic information likely introduces noise or misleading cues even when the correct expert background information is available in context.

higher accuracy ($\text{Acc}_i^{(m)} \uparrow$) on tasks where human experts rate as more feasible ($f_i^{(m)} \uparrow$) and less difficult ($z_i^{(m)} \downarrow$), demonstrating that the human experts can much more reliably capture the predictability of evaluated tasks. Additional results are given in Appendix C Tab. 5 and Tab. 6.

> *Finding #6:* Model failures are primarily driven by factual and extraction errors (avg. 80.14%) and logical reasoning flaws (avg. 87.42%) rather than comprehension issues, frequently manifesting as information fabrication and false certainty.

We classify model failures into 16 error types and 5 categories (Fig. 8) using an LLM judge. Results show failures concentrate in Factual and Extraction errors (avg. 80.1%) and Logical and Reasoning flaws (avg. 87.4%). Prevalent fine-grained errors include Factual Contradiction (avg. 52.3%) and Information Fabrication (avg. 54.0%), indicating that models frequently fail to incorporate relevant experimental information and basic scientific facts when making predictions. Deficiencies in Scientific Rigor (avg. 47.9%) appear primarily as False Certainty (avg. 43.6%), where models express probabilistic or rough predictions as certain facts. Models also on average fail to acknowledge their limitations in providing predictions in about 19.4% of the tasks. Basic comprehension/scope (avg. 10.0%) and formatting/mechanical (avg. <0.6%) errors remain rare, confirming models understand the tasks but lack reasoning capabilities for effective predictions. These patterns persist when only considering tasks which human experts rate as feasible ($\hat{f} \in \{4, 5\}$) as shown in Fig. 13. Tab. 8 provides detailed definitions for the error categories.

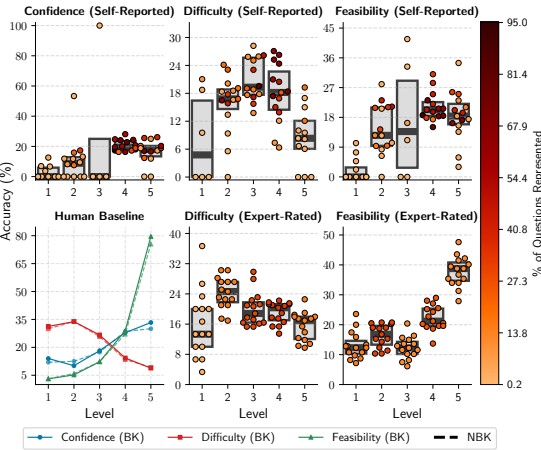

*Figure 7.* **Models are poorly calibrated in self-reported confidence, difficulty, and feasibility, whereas human calibration correlated with accuracy.** The top row plots empirical accuracy against model-provided confidence/difficulty/feasibility metrics; the expected trends (accuracy rising with confidence/feasibility and falling with difficulty) are weak and non-monotonic. The bottom row shows the prediction accuracy of models and humans plotted against the human calibration metrics. The bottom-left subplot shows human accuracy with significant positive correlation with reated confidence and feasibility levels, while showing a significant negative correlation with the rated difficulty as expected. The bottom-middle and bottom-right subplots reveal that model accuracy recovers the expected trends when plotted against *human-rated* difficulty and feasibility levels, confirming that human judgment provides a superior signal for task outcome predictability compared to models' self-reports. Circle colors correspond to the percentage of the number of total questions assigned to that calibration level (x-axis) by each model (top row) or human expert (bottom row). Each circle corresponds to a distinct model.

---

> **Finding #7:** MCQs are substantially easier than free-form and numerical value tasks.

As shown in Fig. 9 we find that model accuracy is highly sensitive to answer format, with multiple-choice questions substantially easier than open-ended generation and especially numerical prediction. This gap is not merely a matter of "MCQs being easier because the correct option is visible," but appears to reflect a broader dependence on recognition over generation: MCQs let models compare candidates and pick the closest match, while free-form and numerical formats require constructing a specific claim/value and committing to it. To isolate format from content, we convert MCQs into matched free-form prompts (MCQ→FF) and re-run evaluation. The resulting drop, visible across essentially all model families, shows that simply removing the provided options degrades accuracy even when the underlying experimental scenario is unchanged. This suggests that headline MCQ accuracy $\text{Acc}_{\text{MCQ}}^{(m)}$ can overestimate how reliably a model would perform in realistic scientific workflows, where predictions are typically produced in open

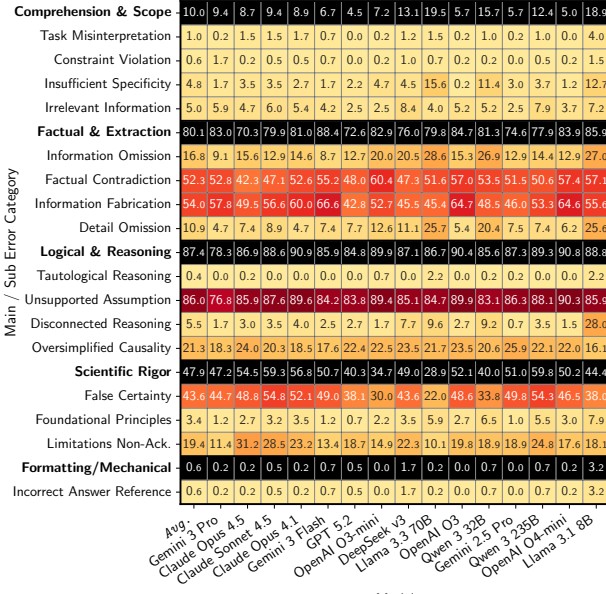

| Main / Sub Error Category | Avg. | Gemini 3 Pro | Claude Opus 4.5 | Claude Sonnet 4.5 | Claude Opus 4.1 | Gemini 3 Flash | GPT 5.2 | OpenAI O3-mini | DeepSeek v3 | Llama 3.3 70B | OpenAI O3 | Qwen 3 32B | Gemini 2.5 Pro | Qwen 3 235B | OpenAI O4-mini | Llama 3.1 8B |
|---|---|---|---|---|---|---|---|---|---|---|---|---|---|---|---|---|
| **Comprehension & Scope** | 10.0 | 9.4 | 8.7 | 9.4 | 8.9 | 6.7 | 4.5 | 7.2 | 13.1 | 19.5 | 5.7 | 15.7 | 5.7 | 12.4 | 5.0 | 18.9 |
| Task Misinterpretation | 1.0 | 0.2 | 1.5 | 1.5 | 1.7 | 0.7 | 0.0 | 0.2 | 1.2 | 1.5 | 0.2 | 1.0 | 0.2 | 1.0 | 0.0 | 4.0 |
| Constraint Violation | 0.6 | 1.7 | 0.2 | 0.5 | 0.5 | 0.7 | 0.0 | 0.2 | 1.0 | 0.7 | 0.2 | 0.2 | 0.0 | 0.5 | 0.2 | 1.5 |
| Insufficient Specificity | 4.8 | 1.7 | 3.5 | 3.5 | 2.7 | 1.7 | 2.2 | 4.7 | 4.5 | 15.6 | 0.2 | 11.4 | 3.0 | 3.7 | 1.2 | 12.7 |
| Irrelevant Information | 5.0 | 5.9 | 4.7 | 6.0 | 5.4 | 4.2 | 2.5 | 2.5 | 8.4 | 4.0 | 5.2 | 5.2 | 2.5 | 7.9 | 3.7 | 7.2 |
| **Factual & Extraction** | 80.1 | 83.0 | 70.3 | 79.9 | 81.0 | 88.4 | 72.6 | 82.9 | 76.0 | 79.8 | 84.7 | 81.3 | 74.6 | 77.9 | 83.9 | 85.9 |
| Information Omission | 16.8 | 9.1 | 15.6 | 12.9 | 14.6 | 8.7 | 12.7 | 20.0 | 20.5 | 28.6 | 15.3 | 26.9 | 12.9 | 14.4 | 12.9 | 27.0 |
| Factual Contradiction | 52.3 | 52.8 | 42.3 | 47.1 | 52.6 | 55.2 | 48.0 | 60.4 | 47.3 | 51.6 | 57.0 | 53.5 | 51.5 | 50.6 | 57.4 | 57.1 |
| Information Fabrication | 54.0 | 57.8 | 49.5 | 56.6 | 60.0 | 66.6 | 42.8 | 52.7 | 45.5 | 45.4 | 64.7 | 48.5 | 46.0 | 53.3 | 64.6 | 55.6 |
| Detail Omission | 10.9 | 4.7 | 7.4 | 8.9 | 4.7 | 7.4 | 7.7 | 12.6 | 11.1 | 25.7 | 5.4 | 20.4 | 7.5 | 7.4 | 6.2 | 25.6 |
| **Logical & Reasoning** | 87.4 | 78.3 | 86.9 | 88.6 | 90.9 | 85.9 | 84.8 | 89.9 | 87.1 | 86.7 | 90.4 | 85.6 | 87.3 | 89.3 | 90.8 | 88.8 |
| Tautological Reasoning | 0.4 | 0.0 | 0.2 | 0.0 | 0.0 | 0.0 | 0.0 | 0.7 | 0.0 | 2.2 | 0.0 | 0.2 | 0.2 | 0.0 | 0.0 | 2.2 |
| Unsupported Assumption | 86.0 | 76.8 | 85.9 | 87.6 | 89.6 | 84.2 | 83.8 | 89.4 | 85.1 | 84.7 | 89.9 | 83.1 | 86.3 | 88.1 | 90.3 | 85.9 |
| Disconnected Reasoning | 5.5 | 1.7 | 3.0 | 3.5 | 4.0 | 2.5 | 2.7 | 1.7 | 7.7 | 9.6 | 2.7 | 9.2 | 0.7 | 3.5 | 1.5 | 28.0 |
| Oversimplified Causality | 21.3 | 18.3 | 24.0 | 20.3 | 18.5 | 17.6 | 22.4 | 22.5 | 23.5 | 21.7 | 23.5 | 20.6 | 25.9 | 22.1 | 22.0 | 16.1 |
| **Scientific Rigor** | 47.9 | 47.2 | 54.5 | 59.3 | 56.8 | 50.7 | 40.3 | 34.7 | 49.0 | 28.9 | 52.1 | 40.0 | 51.0 | 50.9 | 50.2 | 44.4 |
| False Certainty | 43.6 | 44.7 | 48.8 | 54.8 | 52.1 | 49.0 | 38.1 | 30.0 | 43.6 | 22.0 | 48.6 | 33.8 | 49.8 | 54.3 | 46.5 | 38.0 |
| Foundational Principles | 3.4 | 1.2 | 2.7 | 3.2 | 3.5 | 1.2 | 0.7 | 2.2 | 3.5 | 5.9 | 2.7 | 6.5 | 1.0 | 5.5 | 3.0 | 7.9 |
| Limitations Non-Ack. | 19.4 | 11.4 | 31.2 | 28.5 | 23.2 | 13.4 | 18.7 | 14.9 | 22.3 | 10.1 | 19.8 | 18.9 | 18.9 | 24.8 | 17.6 | 18.1 |
| **Formatting/Mechanical** | 0.6 | 0.2 | 0.2 | 0.5 | 0.2 | 0.7 | 0.5 | 0.0 | 1.7 | 0.2 | 0.0 | 0.7 | 0.0 | 0.7 | 0.2 | 3.2 |
| Incorrect Answer Reference | 0.6 | 0.2 | 0.2 | 0.5 | 0.2 | 0.7 | 0.5 | 0.0 | 1.7 | 0.2 | 0.0 | 0.7 | 0.0 | 0.7 | 0.2 | 3.2 |

Model

*Figure 8.* **Analysis of model errors.** We classify errors into a hierarchical taxonomy spanning five top-level (in black background) categories and 16 specific error types. The heatmap shows the percentage of incorrect responses containing each error type for each evaluated model. Error categories (as defined in Tab. 8) progress from surface-level issues (e.g., Comprehension & Scope) to deeper reasoning failures (e.g., Logical & Reasoning flaws) to fundamental scientific deficiencies (Scientific Rigor flaws). Models can exhibit multiple error types simultaneously, so accumulative percentage scores within top-level categories may exceed 100%. SciPredict tasks contribute to top-level category percentages if flagged with at least one underlying error type. Fig. 13 shows the same chart only for tasks with human rated $feasibility \in \{4, 5\}$.

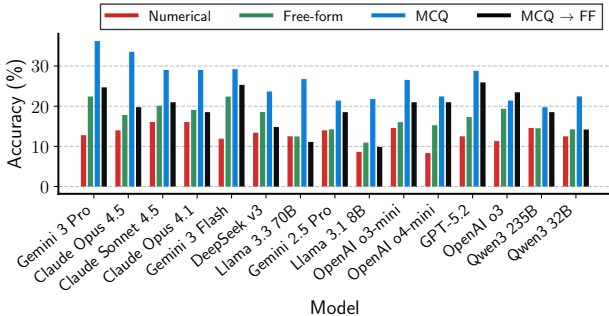

*Figure 9.* **Accuracy is highly sensitive to the question format.** Even when the underlying prediction tasks are identical, performance is generally the highest for multiple-choice questions (MCQs), lower for free-form (FF), and lowest for numerical (NUM) predictions. Crucially, when MCQs are rewritten as free-form (MCQ → FF) questions to remove the answer options, accuracy drops systematically. This shows that a model's reported performance depends heavily on how the prediction is requested.

form $\text{Acc}_{\text{FF}}^{(m)}$ (and often as quantities). Finally, the steepness of the MCQ→free-form drop varies by model, implying meaningful differences in robustness to output constraints. Additional results are given in Appendix C Tab. 4.

*Finding #8:* Performance varies by scientific domain, with Chemistry typically being the most challenging.

Fig. 10 shows that Chemistry has the lowest accuracy on average compared to Biology and Physics. This domain gap is particularly visible for the human baseline. Accuracy gains of models are not uniform across domains, indicating that scaling or general instruction-following ability does not fully translate into robust empirical reasoning in Chemistry. This pattern suggests that our benchmark is sensitive to domain-specific experimental knowledge and intuitions. Additional results are given in Appendix C Tab. 3.

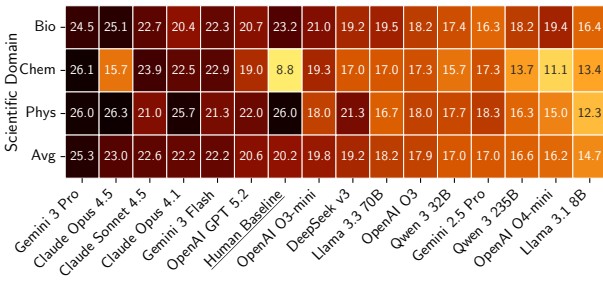

*Figure 10.* **Domain specific accuracy.** Heatmap of model accuracy (%) on benchmark questions, broken down by scientific domain. Results are provided for the evaluated models and human baseline. Overall, frontier models achieve the highest accuracies, but performance varies by domain; Chemistry tends to be the most challenging subset, and several models (including the human baseline) exhibit performance degradation on Chemistry relative to Biology/Physics. The `Avg` shown represents the weighted average respective to number of questions per domain.

*Finding #9:* Performance on this benchmark has a strong correlation with performance on HLE benchmark.

Fig. 11 helps disentangle how much performance on SciPredict (NBK) reflects broad hard-reasoning capability versus a more task-specific ability to anticipate empirical outcomes from experimental descriptions. Although the overall association with HLE is positive, the dispersion around the trendline is substantial: models with similar HLE text-only accuracy can differ by several points on NBK accuracy. This residual structure is informative some models overperform relative to what their HLE score would predict (e.g., DeepSeek v3 achieves comparatively strong NBK accuracy despite very low HLE, and Claude Sonnet 4.5 / Claude Opus 4.1 sit above the fitted line), while others underperform given their HLE level (e.g., Gemini 2.5 Pro, OpenAI O3, and GPT-5.2 fall below the line). These deviations suggest that, beyond general text-only reasoning, strong results on SciPredict also depend on scientific priors and experimental intuition: identifying which intervention details are causally relevant, mapping measurements to plau-

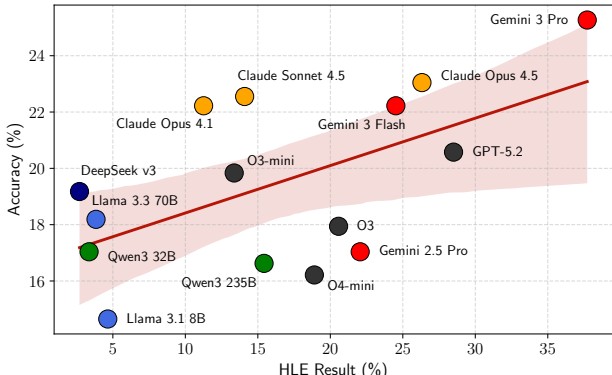

*Figure 11.* **Model accuracy on SciPredict correlates with performance on the HLE benchmark.** Benchmark performance correlates with general hard-reasoning performance. This is a scatter plot of each evaluated model's accuracy on SciPredict in the no-background-knowledge (NBK) setting (y axis) versus its HLE text-only accuracy (x axis). The solid line shows a linear fit and the shaded region indicates the corresponding confidence bands. Overall, SciPredict NBK accuracy exhibits a moderate positive correlation with HLE performance (Pearson r ≈ 0.46), suggesting that broader reasoning capability explains some-but not all-variance in empirical outcome prediction.

sible mechanisms, and remaining robust when background context is withheld in the NBK setting.

## 6. Discussion and Conclusions

Our work reveals fundamental gaps between current LLM capabilities and the requirements for reliable experimental guidance. While frontier models achieve 14-26% accuracy, comparable to human expert baselines around 20%, performance remains insufficient for guiding resource-intensive experimental decisions. Models exhibit severe miscalibration: unlike human experts whose accuracy ranges from ∼5% on infeasible questions to ∼80% on feasible ones, models maintain uniform ∼20% performance regardless of self-reported confidence or feasibility. Expert-curated background knowledge provides modest gains (∼3%), but models cannot autonomously identify or generate helpful context. These findings demonstrate that achieving superhuman scientific assistance requires not merely better predictions, but systems that accurately assess their own reliability.

**Limitations.** While SciPredict establishes a rigorous framework for evaluating experimental outcome prediction, some limitations constrain the scope and generalizability. The benchmark focuses on 3 natural science domains, excluding engineering and computational fields where prediction tasks may exhibit different characteristics. Our temporal cutoff (March 2025) ensures data freshness but limits historical coverage, and the 405-question scale, though substantial, may not capture the full diversity of experimental paradigms within each subdomain.

## Impact Statement

This work introduces SciPredict, a benchmark for evaluating AI systems' ability to predict scientific experimental outcomes, a capability that could significantly accelerate scientific discovery by helping researchers prioritize experiments and allocate resources more efficiently. The potential benefits of achieving reliable experimental outcome prediction are substantial: reduced costs and time in experimental research, faster iteration cycles in scientific discovery, and more efficient allocation of limited research resources, particularly in domains where experiments are costly or time-consuming. However, our work also highlights critical risks that must be addressed before deployment, most notably severe miscalibration where models express high confidence on incorrect predictions and cannot distinguish reliable forecasts from unreliable ones. Deploying such systems prematurely could lead researchers to pursue unproductive experimental directions, waste valuable resources, or—in high-stakes domains like medicine or materials science—make decisions with serious real-world consequences based on unreliable AI predictions. We emphasize that overcoming the calibration gap we identify is as important as improving raw accuracy, and our benchmark provides the measurement framework necessary to track progress toward AI scientific assistants that are not only capable but also trustworthy and appropriately cautious about their own limitations.

## Acknowledgments

Shabihi and Huang are supported by DARPA Transfer from Imprecise and Abstract Models to Autonomous Technologies (TIAMAT) 80321, DARPA HR001124S0029-AIQ-FP-019, DOD-AFOSR-Air Force Office of Scientific Research under award number FA9550-23-1-0048, National Science Foundation TRAILS Institute (2229885). Private support was provided by Peraton and Open Philanthropy. The Authors acknowledge the National Artificial Intelligence Research Resource (NAIRR) Pilot for contributing to this research result.

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

# A. Additional Dataset Details

## A.1. Additional details about task contributors / human baseline participants

We provide additional visualizations of the degree, expertise, and country of origin diversity of the experts recruited for benchmark construction and human baseline. Overall, our experts have strong credentials in their respective fields. For the human baseline, we match experts with relevant expertise to task domains and subdomains; see Tab. 1 for more details.

**Expert recruitment**    To construct our benchmark, we recruited a large cohort of experts in biology, physics, and chemistry. Among them, 54.5% hold a doctoral degree (PhD or equivalent), 34.3% hold a master's degree, and 11.2% hold a bachelor's degree. The experts represent a diverse set of countries, including the United States (14.3%), India (14.3%), United Kingdom (13.6%) , Argentina (7.3%), and more. See Fig. 12 for more details.

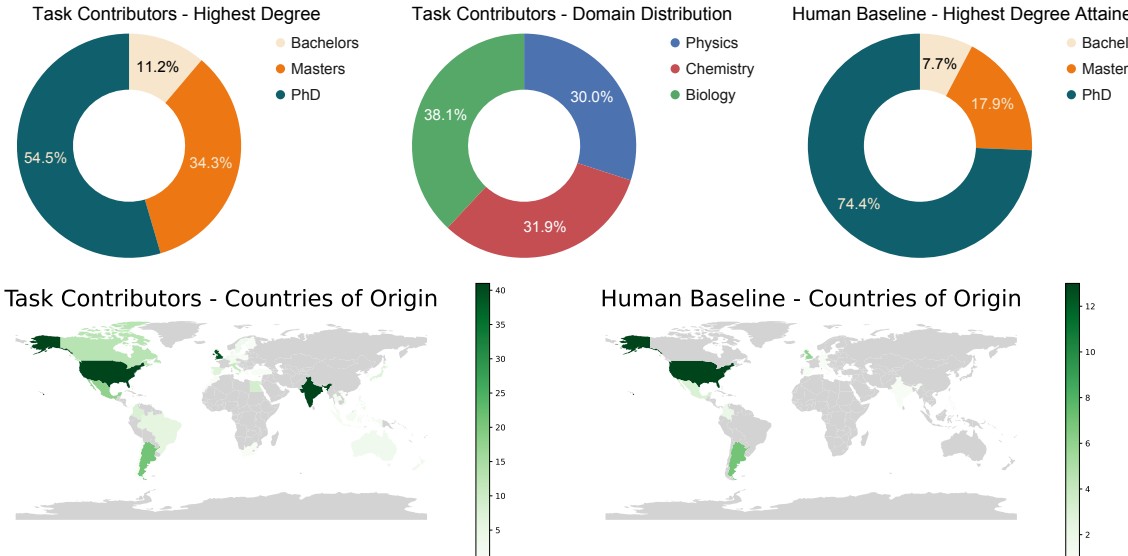

*Figure 12.* **Diversity of the experts recruited for benchmark construction and human baseline.** *Top left*: A plot of the highest degree distribution of experts recruited for benchmark construction. *Top center*: A plot of the domain expertise of experts recruited for benchmark construction. *Top right*: A plot of the highest degree distribution of experts recruited for human baseline. *Bottom left*: A heatmap of the countries of origin of experts recruited for benchmark construction. *Bottom right*: A heatmap of the countries of origin of experts recruited for human baseline.

## A.2. Domain Selection and Question Design Details

**Domain Selection Criteria.**    We selected physics, biology, and chemistry based on three key criteria. First, these domains involve high-stakes applications in engineering, medicine, and materials science, where incorrect predictions can incur significant real-world costs. Second, experimental protocols in these domains are typically well documented, enabling structured extraction of experimental setups, controlled conditions, and measured outcomes. Third, the domains provide sufficient diversity in experimental systems and reasoning styles to evaluate whether models can generalize predictive reasoning across distinct scientific contexts.

**Question Formats and Evaluation.**    Our benchmark includes multiple-choice (MCQ), free-form, and numerical value questions, each with domain-appropriate evaluation procedures. For MCQs, ground truth specifies the correct option or set of correct options. For free-form questions, domain experts design detailed evaluation rubrics that capture the essential scientific reasoning and expected outcomes. For numerical value questions, experts define acceptable answer ranges based on measurement precision and inherent experimental variability, and model predictions are evaluated based on whether they fall within these ranges.

## A.3. Detailed expert task curation

To prevent data leakage from existing pretraining data, recruited domain experts selected empirical research papers published exclusively after March 31, 2025. Selected studies explicitly avoided purely theoretical analyses or computational simulations, focusing solely on clearly documented empirical experiments. Papers are selected from domain specific open source venues that are widely recognized in the scientific community such as bioRxiv, chemRxiv, arXiv, PubMed Central (PMC), Nature, Science.

For each chosen paper, experts explicitly extracted and documented: (1) the domain and specialized subdomain classification, (2) experimental setup details, (3) specific measurements obtained from the experiment, (4) a clear prediction question targeting the experiment's outcome, and (5) the ground truth answer directly sourced from the paper, formatted according to the task type (MCQ, numerical, or free-form). Experts additionally curated background knowledge necessary for informed prediction, selecting relevant domain principles, previously established findings, and theoretical frameworks from the source papers or from expert domain knowledge. Fig. 3 provides a representative example of this extraction and background curation process.

## A.4. Human Baseline Recruitment Details

In addition to the experts involved in benchmark construction, we recruited a separate group of experts to serve as human baseline subjects. These participants were selected to represent an expert-level baseline for the prediction tasks. Human baseline subjects were presented with benchmark questions and asked to provide an answer, explain their reasoning, and report their confidence. To mirror the evaluation protocol used for LLM baselines, each subject completed a second round of the same questions after being provided with the curated background knowledge associated with the task.

The human baseline cohort consists primarily of domain experts, with 74.4% holding doctoral degrees, 17.9% holding master's degrees, and 7.7% holding bachelor's degrees. In terms of primary area of expertise, 48.7% specialize in biology, 33.3% in chemistry, and 17.9% in physics. The cohort also reflects broad geographic diversity, including participants from the United States (33.3%), Argentina (17.9%), the United Kingdom (15.4%), Mexico (7.7%), and Colombia (5.1%). Fig. 12 provides a detailed demographic breakdown.

To ensure that human baseline performance reflects expert-level reasoning rather than domain mismatch, we performed a rigorous assignment process aligning each subject's area of expertise with the corresponding task subdomains. The resulting expertise-to-task mapping is summarized in Tab. 1.

## A.5. Quality Control Details

All data undergoes a multi-stage review process to ensure scientific rigor. Initial screening filters questions where the first version of the paper appeared online on or before March 31, 2025, experiments are simulations or theoretical derivations, answers are directly stated in experimental setup descriptions, phrasing is ambiguous, required predictions exceed available information, or ground truth conflicts with source papers. Questions passing initial screening go through two layers of domain expert reviewers who verify experimental setup precision sufficiency for informed reasoning, background knowledge necessity and sufficiency, ground truth clarity and proper sourcing, and appropriate difficulty level.

For multiple-choice questions, reviewers ensure distractors represent plausible alternatives arising from reasonable but incorrect assumptions rather than obviously wrong options. For free-form questions, reviewers confirm that evaluation rubrics capture essential scientific reasoning without being overly prescriptive about phrasing, and that rubric criteria are mutually exclusive and collectively exhaustive, with each criterion validated to a binary outcome. For numerical value questions, reviewers verify acceptable ranges are neither unrealistically narrow nor trivially broad, reflecting realistic experimental measurement precision and variability. Questions flagged during review undergo revision or removal if fundamental problems cannot be resolved.

## A.6. Data Diversity Details

The benchmark spans 33 specialized subdomains across physics, biology, and chemistry, ensuring models encounter the full spectrum of experimental reasoning required in modern scientific practice. Within physics, questions draw from 9 subdomains such as experimental condensed matter physics, quantum and atomic physics, and high energy particle physics. Biology questions cover 14 subdomains such as molecular biology, neuroscience, plant biology, and ecology. Chemistry

spans 10 subdomains such as organic chemistry, catalysis, and polymer chemistry.

Question complexity varies systematically along multiple axes. Experimental systems range from controlled laboratory setups with few interacting components to complex biological systems with emergent properties. Some questions require single-step causal reasoning, while others demand multi-hop inference chains such as integrating thermodynamics, kinetics, and material properties. Background knowledge requirements span a continuum from questions answerable via undergraduate-level principles to those requiring specialized domain expertise typically held only by active researchers in the relevant subdomain.

Domain distribution remains balanced to prevent overfitting to particular experimental contexts, with 25% of questions from physics, 50% from biology, and 25% from chemistry. Question format distribution is similarly controlled, with 40% multiple-choice, 32% free-form, and 28% numerical value questions. Together, these diversity dimensions ensure the benchmark probes models' general capacity for experimental outcome prediction rather than narrow pattern matching on specific experimental templates or domain conventions.

### A.7. Human baseline expert - Task subdomain mapping

*Table 1.* Subfield expertise of human annotators, grouped by the task domains (Physics, Chemistry, Biology) and subdomains.

| Task Domain | Subdomain | Human Baseline Subfields |
|---|---|---|
| Physics | All Physics | Advanced Chemical Engineering, Applied And Interdisciplinary Physics, Applied Physics And Interdisciplinary, Chemical Engineering, Classical And Mechanical Physics, Condensed Matter And Materials, Electromagnetism And Optics, Engineering Physics, High-energy And Nuclear Physics, Radiophysics & Electronics, Theoretical Physics, Zoology |
| | Condensed Matter & Materials Physics | Advanced Chemical Engineering, Applied Physics And Interdisciplinary, Chemical Engineering, Condensed Matter And Materials, Electromagnetism And Optics, Engineering Physics, Radiophysics & Electronics |
| | Materials Chemistry | Condensed Matter And Materials, Engineering Physics |
| | Optics, Photonics & Laser Physics | Applied Physics And Interdisciplinary, Condensed Matter And Materials, Electromagnetism And Optics, Engineering Physics, Radiophysics & Electronics, Zoology |
| | High-Energy / Nuclear / Particle Physics | Engineering Physics, High-energy And Nuclear Physics, Radiophysics & Electronics, Theoretical Physics, Zoology |
| | Applied & Instrumentation Physics | Applied And Interdisciplinary Physics, Applied Physics And Interdisciplinary, Classical And Mechanical Physics, Condensed Matter And Materials, Electromagnetism And Optics, Engineering Physics, High-energy And Nuclear Physics, Radiophysics & Electronics |
| | Quantum & Atomic Physics | Applied Physics And Interdisciplinary, Condensed Matter And Materials, Electromagnetism And Optics, Engineering Physics, Radiophysics & Electronics, Zoology |
| | Plasma & Nonlinear Physics | Applied Physics And Interdisciplinary, Classical And Mechanical Physics, Electromagnetism And Optics, Engineering Physics, Radiophysics & Electronics |
| | Biophysics | Advanced Chemical Engineering, Applied Physics And Interdisciplinary, Chemical Engineering, Condensed Matter And Materials, Electromagnetism And Optics, Radiophysics & Electronics |
| | Mechanical / Energy / Thermo / Fluid Physics | Classical And Mechanical Physics, Condensed Matter And Materials, Engineering Physics, Radiophysics & Electronics |
| Chemistry | All Chemistry | Advanced Chemical Engineering, Analytical Chemistry, Antimicrobial Resistance, Bio-organic Chemistry, Biochemistry, Biochemistry And Molecular Biology, Catalysis And Environmental Chemistry, Chemical Biology, Chemical Engineering, Chemical Sciences, Digital Technologies Applied To Education, Electrochemistry, Engineering Physics, Green Chemistry, Materials And Inorganic Chemistry, Molecular And Cellular Biology, Molecular Biology And Genetics, Organic And Biological Chemistry, Principles Of Biochemistry, Pure Chemistry, Zoology |
| | Analytical Chemistry | Advanced Chemical Engineering, Analytical Chemistry, Antimicrobial Resistance, Bio-organic Chemistry, Biochemistry And Molecular Biology, Chemical Biology, Chemical Engineering, Chemical Sciences, Digital Technologies Applied To Education, Electrochemistry, Engineering Physics, Materials And Inorganic Chemistry, Molecular And Cellular Biology, Molecular Biology And Genetics, Organic And Biological Chemistry, Principles Of Biochemistry, Pure Chemistry |
| | Materials Chemistry | Analytical Chemistry, Bio-organic Chemistry, Biochemistry And Molecular Biology, Chemical Biology, Chemical Engineering, Digital Technologies Applied To Education, Electrochemistry, Materials And Inorganic Chemistry, Organic And Biological Chemistry |
| | Catalysis | Biochemistry, Biochemistry And Molecular Biology, Catalysis And Environmental Chemistry, Chemical Biology, Chemical Engineering, Chemical Sciences, Digital Technologies Applied To Education, Electrochemistry, Green Chemistry, Materials And Inorganic Chemistry, Principles Of Biochemistry, Pure Chemistry |
| | Physical Chemistry | Advanced Chemical Engineering, Analytical Chemistry, Chemical Engineering, Chemical Sciences, Digital Technologies Applied To Education, Materials And Inorganic Chemistry, Organic And Biological Chemistry, Principles Of Biochemistry, Pure Chemistry |

| Task Domain | Subdomain | Human Baseline Subfields |
|---|---|---|
|  | Organic Chemistry | Analytical Chemistry, Bio-organic Chemistry, Biochemistry And Molecular Biology, Catalysis And Environmental Chemistry, Chemical Biology, Chemical Engineering, Digital Technologies Applied To Education, Electrochemistry, Materials And Inorganic Chemistry, Organic And Biological Chemistry, Zoology |
|  | Nanotechnology / Nanochemistry | Analytical Chemistry, Biochemistry, Biochemistry And Molecular Biology, Catalysis And Environmental Chemistry, Chemical Biology, Chemical Engineering, Digital Technologies Applied To Education, Electrochemistry, Green Chemistry, Materials And Inorganic Chemistry, Organic And Biological Chemistry, Principles Of Biochemistry, Pure Chemistry |
|  | Biochemistry | Antimicrobial Resistance, Biochemistry, Electrochemistry, Molecular And Cellular Biology, Molecular Biology And Genetics, Organic And Biological Chemistry, Principles Of Biochemistry, Pure Chemistry |
|  | Inorganic Chemistry | Analytical Chemistry, Catalysis And Environmental Chemistry, Materials And Inorganic Chemistry |
|  | Environmental Chemistry | Advanced Chemical Engineering, Analytical Chemistry, Chemical Engineering, Materials And Inorganic Chemistry, Zoology |
|  | Polymer Chemistry | Chemical Engineering, Digital Technologies Applied To Education, Materials And Inorganic Chemistry, Organic And Biological Chemistry |
| Biology | All Biology | Antimicrobial Resistance, Bio-organic Chemistry, Biochemistry, Biochemistry And Molecular Biology, Biological Engineering, Biological Sciences, Biomedical Engineering, Biomedical Sciences, Biotechnology, Cell Biology, Chemical Biology, Chemical Engineering, Clinical Drug Development, Developmental Biology, Ecology, Genetics, Green Chemistry, Immunology, Microbiology, Microbiology And Cell Science, Molecular And Cellular Biology, Molecular Biology, Molecular Biology And Genetics, Neurobiology And Behavior, Observational Oceanography, Physiology, Plant Sciences, Research And Data Analysis, Software Engineering, Systems And Synthetic Biology, Taxonomy And Biodiversity, Zoology |
|  | Microbiology | Antimicrobial Resistance, Biochemistry, Biological Engineering, Biological Sciences, Biomedical Engineering, Biomedical Sciences, Cell Biology, Chemical Engineering, Ecology, Microbiology, Microbiology And Cell Science, Molecular And Cellular Biology, Molecular Biology And Genetics, Neurobiology And Behavior, Software Engineering, Systems And Synthetic Biology, Taxonomy And Biodiversity |
|  | Cancer Biology / Oncology | Antimicrobial Resistance, Biochemistry, Biological Engineering, Biological Sciences, Biomedical Engineering, Biomedical Sciences, Cell Biology, Chemical Engineering, Clinical Drug Development, Genetics, Immunology, Microbiology And Cell Science, Molecular And Cellular Biology, Molecular Biology, Molecular Biology And Genetics, Research And Data Analysis, Software Engineering, Taxonomy And Biodiversity |
|  | Neuroscience / Neurobiology | Antimicrobial Resistance, Biochemistry, Biological Engineering, Biomedical Engineering, Cell Biology, Chemical Engineering, Clinical Drug Development, Developmental Biology, Genetics, Immunology, Molecular And Cellular Biology, Molecular Biology, Molecular Biology And Genetics, Neurobiology And Behavior, Physiology, Systems And Synthetic Biology |
|  | Ecology | Biochemistry, Biological Engineering, Biological Sciences, Biomedical Engineering, Biomedical Sciences, Cell Biology, Chemical Engineering, Ecology, Genetics, Microbiology, Microbiology And Cell Science, Observational Oceanography, Plant Sciences, Research And Data Analysis, Systems And Synthetic Biology, Taxonomy And Biodiversity |
|  | Immunology | Bio-organic Chemistry, Biochemistry, Biological Engineering, Biomedical Engineering, Biomedical Sciences, Chemical Engineering, Immunology, Microbiology And Cell Science, Software Engineering, Systems And Synthetic Biology, Zoology |
|  | Molecular Biology | Antimicrobial Resistance, Bio-organic Chemistry, Biochemistry, Biological Engineering, Biological Sciences, Biomedical Engineering, Biomedical Sciences, Cell Biology, Chemical Engineering, Genetics, Microbiology And Cell Science, Molecular And Cellular Biology, Molecular Biology, Molecular Biology And Genetics, Research And Data Analysis, Software Engineering, Taxonomy And Biodiversity |
|  | Pharmacology / Toxicology | Biochemistry, Biological Sciences, Biomedical Sciences, Cell Biology, Clinical Drug Development, Genetics, Immunology, Microbiology And Cell Science, Observational Oceanography, Physiology, Research And Data Analysis, Software Engineering |
|  | Plant Biology | Biochemistry, Biological Sciences, Developmental Biology, Ecology, Genetics, Observational Oceanography, Plant Sciences, Research And Data Analysis, Systems And Synthetic Biology, Taxonomy And Biodiversity |
|  | Animal Behavior | Biochemistry, Biological Sciences, Cell Biology, Clinical Drug Development, Developmental Biology, Genetics, Microbiology, Molecular Biology, Observational Oceanography, Physiology, Systems And Synthetic Biology, Taxonomy And Biodiversity, Zoology |
|  | Cell Biology | Antimicrobial Resistance, Bio-organic Chemistry, Biochemistry, Biological Engineering, Biological Sciences, Biomedical Engineering, Biomedical Sciences, Cell Biology, Chemical Engineering, Clinical Drug Development, Developmental Biology, Genetics, Immunology, Microbiology And Cell Science, Molecular And Cellular Biology, Molecular Biology, Molecular Biology And Genetics, Neurobiology And Behavior, Physiology, Research And Data Analysis, Software Engineering, Taxonomy And Biodiversity |
|  | Physiology | Biochemistry, Biological Engineering, Biological Sciences, Biomedical Engineering, Biotechnology, Cell Biology, Chemical Engineering, Clinical Drug Development, Genetics, Microbiology, Molecular And Cellular Biology, Molecular Biology, Neurobiology And Behavior, Observational Oceanography, Physiology, Plant Sciences, Systems And Synthetic Biology, Taxonomy And Biodiversity |

| Task Domain | Subdomain | Human Baseline Subfields |
|---|---|---|
| | Biochemistry | Biochemistry, Biochemistry And Molecular Biology, Biological Engineering, Biomedical Engineering, Cell Biology, Chemical Biology, Chemical Engineering, Clinical Drug Development, Genetics, Molecular Biology, Physiology, Software Engineering, Zoology |
| | Genetics | Biochemistry, Biological Sciences, Biomedical Sciences, Cell Biology, Clinical Drug Development, Genetics, Microbiology, Microbiology And Cell Science, Molecular Biology, Observational Oceanography, Plant Sciences, Systems And Synthetic Biology, Taxonomy And Biodiversity |
| | Bioengineering / Biomaterials | Antimicrobial Resistance, Biochemistry, Biological Sciences, Biomedical Sciences, Cell Biology, Green Chemistry, Microbiology And Cell Science, Molecular And Cellular Biology, Molecular Biology, Molecular Biology And Genetics, Observational Oceanography, Physiology, Systems And Synthetic Biology |

*Table 2.* Task distribution by scientific subfield: number of tasks per Biology, Physics, and Chemistry subdomain.

| Field | Subfield | Count |
|---|---|---|
| Physics | Condensed Matter & Materials Physics | 33 |
| | Materials Chemistry | 17 |
| | Optics, Photonics & Laser Physics | 16 |
| | High-Energy / Nuclear / Particle Physics | 15 |
| | Applied & Instrumentation Physics | 13 |
| | Quantum & Atomic Physics | 10 |
| | Plasma & Nonlinear Physics | 5 |
| | Biophysics | 3 |
| | Mechanical / Energy / Thermo / Fluid Physics | 2 |
| Chemistry | Analytical Chemistry | 18 |
| | Materials Chemistry | 17 |
| | Catalysis | 16 |
| | Physical Chemistry | 14 |
| | Organic Chemistry | 13 |
| | Nanotechnology / Nanochemistry | 10 |
| | Biochemistry | 8 |
| | Inorganic Chemistry | 6 |
| | Environmental Chemistry | 4 |
| | Polymer Chemistry | 3 |
| Biology | Microbiology | 36 |
| | Cancer Biology / Oncology | 28 |
| | Neuroscience / Neurobiology | 19 |
| | Ecology | 17 |
| | Immunology | 16 |
| | Molecular Biology | 14 |
| | Pharmacology / Toxicology | 13 |
| | Plant Biology | 13 |
| | Animal Behavior | 13 |
| | Cell Biology | 10 |
| | Physiology | 9 |
| | Biochemistry | 8 |
| | Genetics | 8 |
| | Bioengineering / Biomaterials | 3 |

# B. Example Data

## B.1. Task examples

<div>

**Physics: Free-Form Question**

**Paper Title:** Compact Continuous Cold Atomic Beam from a Single Cell with 3D Cooling and Ultra-low Light Shift

**Link to The Paper:** https://arxiv.org/abs/2510.13126

**Experimental Setup:** Researchers investigated a compact single-cell source of a continuous cold-atom beam ($^{87}$Rb) that achieves simultaneous 3D cooling by integrating a two-dimensional magneto-optical trap (2D MOT) with an off-axis moving optical molasses (OM). A vapor-cell apparatus (overall length $\approx$170 mm) provided transverse MOT cooling with circularly polarized beams detuned by $\Delta$MOT = -4$\Gamma$ from the F = 2 $\rightarrow$ F' = 3 D$_2$ transition and a cylindrical quadrupole field ($\approx$10 G cm$^{-1}$), where $\Gamma$ is the natural linewidth. Longitudinal cooling and velocity control were realized with two pairs of lin$\perp$lin OM beams oriented 20° to the extraction axis, detuned by $\Delta$OM = -5$\Gamma$ and symmetrically shifted by $\pm\delta$OM to set the mean atomic speed ($\approx$5–20 m s$^{-1}$) over an OM interaction length lOM $\approx$ 50 mm. Custom in-vacuum mirrors formed the off-axis geometry and incorporated a 0.8 mm output aperture to collimate the beam (cooling length lc $\approx$ 50 mm) while suppressing near-resonant stray light. The setup included permanent-magnet field generation, state-preparation "plug" lasers 40 mm downstream for sharp time-of-flight (TOF) edges, and fluorescence detection at 294 mm with a calibrated photomultiplier tube (PMT) to extract longitudinal temperature, velocity, and flux. For coherence diagnostics, two $\pi$/2 Raman beams separated by L = 100 mm in a magnetically shielded region produced spatial-domain Raman–Ramsey fringes, enabling quantification of decoherence and ultra-low light shift (typ. -0.51 Hz) under operating MOT power.

**Measurements Taken:**
- Time-of-flight (TOF) time series and distribution obtained from the emitted fluorescence from the atoms in F=2 state, collected with imaging optics and recorded by a calibrated PMT at a primary detection distance of 294 mm.

**Outcome Prediction Question:** Researchers investigated the longitudinal temperature and atomic flux of a continuous cold $^{87}$Rb beam using a time-of-flight (TOF) method. The temperature was extracted from the FWHM of the TOF distribution, while the flux was obtained from the integrated spectral density. Based on measurements for a saturation intensity of 1.67 mW/cm$^2$, what outcome would researchers expect for the change in longitudinal temperature and atomic flux when the MOT power is increased?

**Ground Truth Answer:** Increasing MOT power raises the flux but affects the temperature only weakly.

**Background Knowledge:**
- Combining a 2D MOT with an off-axis moving OM yields a high-flux beam with significantly reduced longitudinal temperature compared to conventional MOT-based sources.
- Continuous operation of cold-atom beam sources eliminates the dead time inherent to pulsed sources and thus suppresses aliasing noise from undersampling.

**Rubrics:**
- Response states that increasing the magneto-optical trap power increases atomic flux.
- Response states that increasing the magneto-optical trap power has a little influence on temperature.

</div>

---

Physics: Multiple-Choice Question

**Paper Title:** Ionization and temperature measurements in warm dense copper using x-ray absorption spectroscopy

**Link to The Paper:** https://arxiv.org/abs/2509.13272

**Experimental Setup:** Researchers investigated the ionization and temperature of warm dense copper (Cu) using X-ray absorption spectroscopy (XAS) at the OMEGA Laser Facility to characterize plasmas at several times solid density. The experimental configuration consists of a planar target and a separate backlighter positioned 3 mm away. A series of 60 laser beams, delivering 3.4–5.4 kJ per side of 351 nm light, and the achieved laser intensity is 161 - 770 TW/cm$^2$ over the three pulse length configurations, was symmetrically focused onto a planar buried-layer target composed of 125 $\mu$m CH ablators enclosing a 10 $\mu$m-thick Cu foil (8.96 g/cm$^3$ solid density) with a 500 $\mu$m diameter, surrounded by an Au washer. The laser spot ($\approx$ 880 $\mu$m diameter) was smoothed with distributed phase plates and spectral dispersion to generate uniform counter-propagating shocks. A 6 $\mu$m Ge backlighter foil, coated on graphite and irradiated with six additional beams ($\approx$1.2 kJ, 500 ps pulse), is produced at a spot diameter of 140 $\mu$m. The transmitted x-rays were recorded using the EFX flat-crystal spectrometer (Si 111) over the 6.3–11.4 keV range on an image plate with Mn, Fe, and W filters serving as fiducial markers. Shock timing and planarity, as well as shock break-in and break-out of the Cu layer, were verified through a line-imaging VISAR system and a streaked optical pyrometer (SOP) on one-sided targets, ensuring symmetric compression and precise backlighter synchronization. 3 VISAR measurement is done with 1 ns, 2 ns, or 3 ns square pulses using 14 beams per side, respectively. Each measurement has two VISAR channels with different sensitivities; one leg was set with 33.66 $\mu$m/ns/fringe, and the second with 13.538 $\mu$m/ns/fringe.

**Measurements Taken:**
- Shock breakout times (in ns) and planarity were measured with the VISAR system.
- Shock velocity time history as a function of position across the target measured with the VISAR system.

**Outcome Prediction Question:** An investigation into shock breakout times and shock velocity time histories as a function of position across the target of warm dense copper (Cu) plasma is conducted using a VISAR system. The experimental configuration consists of a planar target and a separate backlighter positioned 3 mm away. A series of 60 laser beams was symmetrically focused onto a planar buried-layer target surrounded by an Au washer. The laser spot was smoothed with distributed phase plates and spectral dispersion to generate uniform counter-propagating shocks, compressing the Cu layer. A Ge backlighter foil, coated on graphite and irradiated with six additional beams, is produced. The transmitted X-rays were also recorded using the EFX flat-crystal spectrometer. Which behavior is most likely observed?
A. Shocks were non-planar over the target region, and warm dense copper shows Ionization Potential Depression (IPD).
B. Shocks were highly planar over the target region, and the absorption spectra of warm dense copper features blue shift of both the K-edge and the bound-bound resonance 1s→3p absorption relative to the cold edge.
C. Shocks were highly planar over the target region, and the absorption spectra of warm dense copper features red shift of both the K-edge and the bound-bound resonance 1s→3p absorption relative to the cold edge.
D. Shocks were highly planar over the target region, and the absorption spectra of warm dense copper features blue shift of the K-edge relative to the cold edge, but no shift for the bound-bound resonance 1s→3p absorption.

**Ground Truth Answer:** B

**Background Knowledge:**
- Generating warm dense matter in the laboratory often involves significant temporal and spatial gradients that complicate the analysis of experimental observables. Incorporating gradients in the analysis of experimental data, while possible, increases the uncertainties in the inferred plasma conditions.
- At these high-density conditions, the measured Cu K-edge exhibits sensitivity to the electron temperature, allowing for a direct inference of the temperature from the slope of the Cu K-edge.
- Temperature sensitivity of the K-edge can still be the dominant edge effect, in general, as the temperature nears the Fermi energy, the K-edge shape of the non-degenerate material becomes unsuitable as a temperature inference.



**Physics: Numerical Value Question**

**Paper Title:** A sub-volt near-IR lithium tantalate electro-optic modulator

**Link to The Paper:** https://arxiv.org/abs/2505.00906

**Experimental Setup:** Researchers fabricated a TFLT MZM operating at a near-IR wavelength of 737 nm. The fabricated unbalanced MZM consists of a directional coupler as an input beamsplitter and a L = 5 mm long electrode in the ground-signal-ground configuration, followed by another directional coupler at the output. Grating couplers are used to couple light on and off the chip to near-IR single-mode fibers. The optical layer of the device is defined using 150 keV electron-beam lithography with 500 nm-thick ma-N2405 resist on top of a 200 nm-thick x-cut TFLT-on-SiO2 layer. The waveguide width is designed to be 600 nm. The SiO2 layer is 2 $\mu$m-thick and is on a Si substrate. The TFLT is etched by 100 nm using an Ar+-based inductively coupled plasma reactive ion etching. Etch-induced re-deposition is removed using a high-pH solution. The devices are then annealed in an O2 atmosphere at 520°C for 2 h to mitigate etch-induced imperfections. For the MZMs, an 800 nm-thick SiO2 cladding layer is then deposited by plasma-enhanced chemical vapor deposition. The DC bias stability of two electro-optic Mach-Zehnder modulators is compared. The first modulator is fabricated using thin-film lithium tantalate (TFLT), and the second, serving as a counterpart, is fabricated with a similar process using thin-film lithium niobate (TFLN). For the test, each modulator is subjected to a constant on-chip optical power of 4.3 dBm at a wavelength of 737 nm. A DC step voltage is applied to each device to set its operating point at quadrature bias. The output optical power from the modulator is then monitored over 16 minutes in ambient conditions to measure any drift from this bias point. To measure the DC bias stability of MZM over long timescales. First, it applied a 0.1 Hz-frequency square wave to the modulator using an on-chip optical power, and measured the modulator response with a photodetector.

**Measurements Taken:**
- The output optical power as a function of time over 16 minutes for the TFLT modulator.
- The output optical power as a function of time over 16 minutes for the TFLN modulator.
- The total DC bias drift, in decibels (dB), for the TFLT modulator.
- The total DC bias drift, in decibels (dB), for the TFLN modulator.

**Outcome Prediction Question:** An experiment compares the long-term stability of two Mach-Zehnder modulators, one made from thin-film lithium tantalate (TFLT) and a counterpart from thin-film lithium niobate (TFLN). Both are operated with 4.3 dBm of on-chip optical power at 737 nm and biased at quadrature. The output power is monitored for 16 minutes to quantify the DC bias drift. To measure the DC bias stability of MZM over long timescales. First, it applied a 0.1 Hz-frequency square wave to the modulator using an on-chip optical power, and measured the modulator response with a photodetector. Based on the experimental results, what is the total measured DC bias drift, in decibels (dB), for the thin-film lithium niobate (TFLN) modulator?

**Ground Truth Answer:** $\Delta$DC bias drift = [7.2–8.8] dB at 16 min for the TFLN modulator operated at 4.3 dBm optical power (737 nm). No CI/SE/SD reported → fallback ±0.8 dB applied.

**Background Knowledge:**
- In particular, the relaxation rate will increase with more applied optical power and can be exacerbated with applied DC or RF field. This effect reduces the DC stability of electro-optic circuits, such as Mach-Zehnder modulators (MZMs), and has been one of the main challenges faced by TFLN photonics

</div>

---

**Biology: Free-Form Question**

**Paper Title:** Dopamine induces fear extinction by activating the reward-responding amygdala neurons

**Link to The Paper:** https://pmc.ncbi.nlm.nih.gov/articles/PMC12067255/

**Experimental Setup:** Researchers tested whether ventral tegmental area (VTA) dopamine signaling in the basolateral amygdala (BLA) drives fear extinction by acting on reward-responding posterior BLA (pBLA) neurons versus fear-coding anterior BLA (aBLA) neurons, using adult mice (DAT-IRES-Cre; EYFP controls; subtype mapping with Rspo2-Cre for aBLA and Ppp1r1b/Cartpt-Cre for pBLA). DAT-Cre mice received bilateral VTA injections of Cre-dependent ChR2-EYFP (activation) or eNpHR3.0-EYFP (inhibition); controls received EYFP; optic fibers were implanted over pBLA or aBLA to manipulate VTA→BLA terminals. Training: Day 1 contextual fear conditioning (baseline ~3 min, then 3 footshocks, 0.60 mA, 2 s); Day 2 45-min extinction (no shocks); Day 3 10-min retrieval. Intervention (extinction only): starting 5 min into extinction, deliver 8 cycles of 3-min light separated by 2-min no-light (activation: blue 450–470 nm, 8–12 mW, 20 Hz pulses; inhibition: green 520–550 nm, 8–12 mW, continuous) with fibers targeted to pBLA or aBLA. Behavior videos were recorded with VideoFreeze software and freezing level was scored manually by experimenters who were blinded to conditions or automatically with DeepLabCut behavior analysis toolbox and custom Python code (68). Freezing was quantified in 5-min bins across extinction and again during retrieval.

**Measurements Taken:**
- Extinction learning: Percent freezing per 5-min bin across the 45-min Day 2 session (9 bins). Scored manually by experimenters who were blinded to conditions or automatically with DeepLabCut behavior analysis toolbox and custom Python code (68).
- Extinction memory: Percent freezing during the Day 3 retrieval test (10 min). Scored manually by experimenters who were blinded to conditions or automatically with DeepLabCut behavior analysis toolbox and custom Python code (68).

**Outcome Prediction Question:** Mice underwent contextual fear conditioning (Day 1: context + three 0.60 mA, 2 s shocks), 45-min extinction (Day 2, no shocks), and 10-min retrieval (Day 3). During extinction, VTA dopamine terminals in pBLA (Ppp1r1b$^+$) or aBLA (Rspo2$^+$) were optogenetically manipulated beginning 5 min into the session using 8 cycles of 3 min light separated by 2 min: activation (blue 450–470 nm, 8–12 mW, 20 Hz) or inhibition (green 520–550 nm, 8–12 mW, constant). Freezing was binned in 5-min windows across extinction and measured again at retrieval. How do these projection-specific manipulations (activation and inhibition of VTA dopamine terminals in the pBLA and in aBLA) affect fear extinction and retrieval compared with EYFP controls?

**Ground Truth Answer:** Activation of VTA dopamine terminals in the pBLA promotes faster extinction and improved retrieval, indicating an enhancement of extinction learning. In contrast, inhibition of pBLA dopamine input impairs both extinction and retrieval. Activation of VTA terminals in the aBLA leads to increased freezing later in extinction and poorer retrieval performance, suggesting interference with extinction memory formation, while inhibition of aBLA terminals produces no reliable behavioral change.

**Background Knowledge:**
- Fear extinction is a form of new learning that allows for the adaptive control of fear behaviors and is commonly studied using Pavlovian conditioning tasks.
- aBLA Rspo$^+$ neurons encode negative valence and drive aversive behaviors whereas pBLA Ppp1r1b$^+$ neurons encode positive valence and drive appetitive behaviors.
- VTA dopamine as a teaching signal: DA activity to shock omission can initiate extinction learning and is required for extinction.
- Terminal activation (ChR2, blue, pulsed) vs inhibition (eNpHR3.0, green, constant) at BLA terminals tests sufficiency/necessity of VTA→BLA pathways.
- Freezing is the behavioral measure; decreases across 5-minute bins and at retrieval indicate successful extinction.

**Rubrics:**
- The response should state that activation of ventral tegmental area dopamine terminals in the posterior basolateral amygdala of adult mice promotes faster extinction compared to control. Use of acronyms such as VTA or pBLA are acceptable.
- The response should state that activation of ventral tegmental area dopamine terminals in the posterior basolateral amygdala of adult mice improves retrieval compared to control. Use of acronyms such as VTA or pBLA are acceptable.
- The response should state that inhibition of ventral tegmental area dopamine terminals in the posterior basolateral amygdala of adult mice impairs extinction compared to control. Use of acronyms such as VTA or pBLA are acceptable.
- The response should state that inhibition of ventral tegmental area dopamine terminals in the posterior basolateral amygdala of adult mice impairs retrieval compared to control. Use of acronyms such as VTA or pBLA are acceptable.
- The response should state that activation of ventral tegmental area terminals in the anterior basolateral amygdala of adult mice leads to increased freezing later in extinction compared to control. Use of acronyms such as VTA or aBLA are acceptable.
- The response should state that activation of ventral tegmental area terminals in the anterior basolateral amygdala of adult mice leads to poorer retrieval performance compared to control. Use of acronyms such as VTA or aBLA are acceptable.
- The response should state that inhibition of ventral tegmental area terminals in the anterior basolateral amygdala of adult mice produces no reliable behavioral change compared to control. Use of acronyms such as VTA or aBLA are acceptable.

## Biology: Multiple-Choice Question

**Paper Title:** Social Tolerance and Innovation in Capuchins: socially more tolerant brown capuchins are better problem-solvers than less tolerant white-faced capuchins

**Link to The Paper:** https://www.biorxiv.org/content/10.1101/2025.09.05.674457v1.full

**Experimental Setup:** Researchers tested three groups of white-faced capuchins (Cebus capucinus)(n = 23 individuals in total) and three groups of brown capuchins (Sapajus apella) (n = 20 individuals in total) to explore and compare the relationship between social tolerance and problem-solving propensities. To measure social tolerance, they prepared an area of 1 m$^2$ per five animals in the group, in which they distributed apple pieces and measured the proportion of individuals within the co-feeding area at each scan sample. To measure problem-solving propensities, they designed three versions of novel extractive foraging devices requiring one to three steps to acquire the food reward. For the first puzzle, animals had to rotate a door to either the left or right to access a hidden reward (1/24 of an apple) by reaching into a box. For the second puzzle, animals had to pull on a chain reaching out of a box, which moved a blockade out of the way so that they could push in a door and reach into the box. For the third puzzle, animals had to pull a metal rod blocking a slider that had to be pulled upwards and held in position to reach into the box and then pull on a chain to access the hidden reward. Researchers analyzed the approaching, exploring, and solving behaviour separately.

**Measurements Taken:**
- Proportion of individuals within the co-feeding area at each scan sample (social tolerance)
- Proportion of individuals within the puzzle area at each scan sample (social tolerance)
- Number of approaches to a food puzzle area
- Approaching a food puzzle area duration
- Approaches to a food puzzle area latency
- Number of exploration events (touch, sniff, interact) during the approaches to a food puzzle area
- Number of times the capuchins successfully solved the puzzles
- Exploration of food puzzle events latency
- Time to solve a puzzle

**Outcome Prediction Question:** Researchers tested three groups of white-faced capuchins (Cebus capucinus) and three groups of brown capuchins (Sapajus apella) to explore and compare the relationship between social tolerance and problem-solving propensities. To measure social tolerance, they prepared an area of 1 m$^2$ per five animals in the group, in which they distributed apple pieces and measured the proportion of individuals within the co-feeding area at each scan sample. To measure problem-solving propensities, they designed three versions of novel extractive foraging devices requiring one to three steps to acquire the food reward. Which of the following outcomes is most likely?
A. Both species should show the same levels of social tolerance and problem-solving propensities.
B. White-faced capuchins should show the highest level of social tolerance and problem-solving propensities.
C. White-faced capuchins should show the lowest level of social tolerance and problem-solving propensities.
D. White-faced capuchins should show the highest level of social tolerance and the lowest level of problem-solving propensities.

**Ground Truth Answer:** C

**Background Knowledge:**
- Social tolerance has increasingly been linked to the facilitation of social learning across a variety of species, including chimpanzees, orangutans, macaques, capuchin monkeys, lemurs, and birds.
- White-faced capuchins (Cebus capucinus) and brown capuchins (Sapajus apella) exhibit a diverse array of traditions.
- White-faced capuchins (Cebus capucinus) are less known for using tools (but see Barrett et al., 2018), but they regularly engage in object use (Boinski, 1988).
- Robust capuchins (Sapajus spp.) have fewer documented social traditions but exhibit a wide range of foraging traditions, including tool-use, and show notable social tolerance in these contexts, tolerating close proximity of conspecifics.

---

Biology: Numerical Value Question

**Paper Title:** GsMTx4-loaded GelMA promotes tendon regeneration and suppresses heterotopic ossification via the Apelin signaling pathway

**Link to The Paper:** https://www.sciencedirect.com/science/article/pii/S0142961225004260?via%3Dihub

**Experimental Setup:** Researchers employed Male Sprague Dawley (SD) rats (10–12 weeks old, weighing 250–300 g) as animal model for studying tendon repair and regeneration. A central defect (1 mm in width and 5 mm in length) was created in the Achilles tendon using two parallel No.15 surgical blades. Subsequently, the skin was sutured using 4-0 Vicryl sutures. The rats received temgesic (0.3 mg/kg of body weight) for three consecutive days following the surgery to manage pain. The rats were randomly assigned to one of four groups: Achilles tendon defect (ATD) (no treatment), GelMA, GelMA + 50 $\mu$g GsMTx4, GelMA + 100 $\mu$g GsMTx4. At the time of injury, a mixture of GelMA and LAP (Lithium Phenyl-2,4,6-Trimethylbenzoylphosphinate) (20 $\mu$l), loaded with 50 or 100 $\mu$g GsMTx4 where appropriate, was placed within the ATD of treated animals and transformed into the gel state with a blue light source (3 W, 405 nm) for 30 s at a distance of 2 cm from the defect. These animals were euthanized at 2, 4, and 8 weeks post-treatment, with six rats per group per time point. The harvested Achilles tendons were fixed in 4% paraformaldehyde at room temperature for 24 h. Following fixation, the samples were rinsed with running water and dehydrated with an ethanol gradient, and embedded in paraffin. The blocks were sectioned at 5 $\mu$m thickness using a microtome and stained with Hematoxylin and Eosin (H&E). Semi-quantitative analysis of H&E staining results was conducted according to the modified Bonar score.

**Measurements Taken:**
- Histologic Bonar Score (ATD, GelMA, GelMA + 50 $\mu$g GsMTx4, GelMA + 100 $\mu$g GsMTx4): 2 weeks; 4 weeks; 8 weeks.

**Outcome Prediction Question:** Researchers employed Male Sprague Dawley (SD) rats (10–12 weeks old, weighing 250–300 g) as animal model for studying tendon repair and regeneration. A central defect (1 mm in width and 5 mm in length) was created in the Achilles tendon using two parallel No.15 surgical blades. The rats were randomly assigned to one of four groups: Achilles tendon defect (ATD) (no treatment), GelMA, GelMA + 50 $\mu$g GsMTx4, GelMA + 100 $\mu$g GsMTx4. At the time of injury, a mixture of GelMA and LAP (Lithium Phenyl-2,4,6-Trimethylbenzoylphosphinate) (20 $\mu$l), loaded with 50 or 100 $\mu$g GsMTx4 where appropriate, was placed within the ATD of treated animals and transformed into the gel state with a blue light source. The animals were euthanized at 2, 4, and 8 weeks post-treatment. The harvested Achilles tendons were embedded in paraffin, sectioned using a microtome, and stained with Hematoxylin and Eosin (H&E). Semi-quantitative analysis of H&E staining results was conducted according to the modified Bonar score (BS). Based on the reported values of the BS for Achilles tendon repair and regeneration, what is the predicted difference of the BS (in points) between the GelMA and the GelMA + 100 $\mu$g GsMTx4 groups 8-weeks post treatment?

**Ground Truth Answer:** $\Delta$ BS (GelMA - GelMA + 100 $\mu$g GsMTx4) 8-weeks post treatment = 4 - 6 points; derived from BS GelMA 8-weeks post treatment = ~9 points, BS GelMA + 100 $\mu$g GsMTx4 8-weeks post treatment = ~4 points. Note: No CI/SE/SD reported -> fallback $\pm$ 10% units (rounded) applied.

**Background Knowledge:**
- Tendon regeneration is highly relied on the surrounding mechanical environment.
- Studies have demonstrated the importance of Piezo1 in modulating cellular behaviors to mechanical cues, such as cell migration, differentiation, proliferation, and extracellular matrix synthesis.
- GelMA hydrogel demonstrates excellent biocompatibility and sustained release properties.
- The mechanosensitive ion channel Piezo1 is inhibited by the peptide GsMTx4

**Paper Title:** An investigation of the physical and chemical changes of Pd nanoparticles on carbon supports in response to the release of hydrogen from aqueous formate solutions

**Link to The Paper:** https://chemrxiv.org/engage/chemrxiv/article-details/68d16d29f2aff16770fa93bd

**Experimental Setup:** Researchers prepared and analyzed Pd nanoparticles supported on carbon materials to examine their structural and chemical evolution during hydrogen release from aqueous sodium formate. Three supports were used: carbon black (Vulcan XC-72), nitrogen-doped carbon (NC), and graphitic carbon nitride (g-$C_3N_4$). Nitrogen-doped carbon was obtained by heating a melamine–carbon black mixture at 700 °C under nitrogen, while g-$C_3N_4$ was synthesized by heating urea at 500 °C in air. Pd catalysts were produced by reducing $H_2PdCl_4$ with $NaBH_4$ in trisodium citrate solution at 25 °C, yielding a 1 wt% Pd loading. The product was filtered, washed, and dried at 85 °C for 24 h, and selected samples were calcined at 250 °C for 3 h in air. Structural and compositional analyses included inductively coupled plasma–optical emission spectrometry (PerkinElmer 7300 DV) to determine Pd content, X-ray diffraction (Rigaku SmartLab SE, Cu K$\alpha$, 2$\Theta$ = 2–100°) to assess crystallinity, and nitrogen physisorption (Micromeritics ASAP 2020) using BET and BJH models to measure surface area and pore volume. Pd dispersion was quantified by CO chemisorption (Micromeritics ASAP 2020C, 30 °C, pre-reduced at 100 °C for 0.5 h), and nanoparticle morphology was examined by aberration-corrected scanning transmission electron microscopy (Thermo Fisher Themis Z, 300 kV). Catalytic performance was tested in a 50 mL batch reactor containing 250 mg of catalyst and 10 mL of 1 M sodium formate at 65 °C under $N_2$ with stirring at 500 rpm for 2 h, where gas evolution was monitored by pressure change and analyzed using a micro-gas chromatograph. In-situ X-ray absorption spectroscopy was performed at the Stanford Synchrotron Radiation Lightsource beamline 4-1 to monitor Pd oxidation states during reaction using Pd K-edge XANES and EXAFS scans (24126–25238 eV, 0.5 × 4 mm beam). Catalyst reuse tests were carried out by recovering the solid after reaction, washing with deionized water, drying at 80 °C, and re-calcining at 180 or 250 °C for 3 h when required. All synthesis, characterization, and catalytic experiments were conducted under controlled temperature and atmospheric conditions to ensure reproducibility.

**Measurements Taken:**
- Pd oxidation state and local atomic structure characterized by in-situ X-ray Absorption Spectroscopy (XAS, SSRL beamline 4-1) with Pd K-edge XANES and EXAFS scans (24126–25238 eV, beam size 0.5 × 4 mm) under reaction conditions.
- Palladium loading (wt%) measured using Inductively Coupled Plasma–Optical Emission Spectroscopy (ICP-OES, PerkinElmer 7300 DV) to quantify Pd content on carbon supports.

**Outcome Prediction Question:** Palladium nanoparticles supported on carbon materials were assessed as catalysts for hydrogen release from aqueous sodium formate. Three supports- carbon black (Vulcan XC-72), nitrogen-doped carbon (NC), and graphitic carbon nitride (g-$C_3N_4$)- were employed, with NC synthesized by heating a melamine-carbon black mixture at 700 °C under $N_2$ and g-$C_3N_4$ prepared by urea pyrolysis at 500 °C in air. Pd catalysts (1 wt%) were obtained by reducing $H_2PdCl_4$ with $NaBH_4$ in trisodium citrate at 25 °C, followed by drying and optional calcination at 250 °C. Structural and chemical characterization included ICP–OES for Pd content, XRD for crystallinity, $N_2$ physisorption for surface area, CO chemisorption for Pd dispersion, and STEM for nanoparticle morphology. Catalytic performance was evaluated in a batch reactor (65 °C, 1 M sodium formate) by monitoring gas evolution and composition via micro-GC. In-situ XANES/EXAFS at the Pd K-edge tracked oxidation-state changes during reaction, and reuse tests examined catalyst stability following washing and re-calcination. What will in-situ XANES analysis reveal about the role of palladium oxide (PdO) as an active catalyst for formate dehydrogenation?

**Ground Truth Answer:** In-situ XANES experiments unambiguously demonstrate that PdO is rapidly reduced to metallic Pd and then forms Pd hydride upon exposure to a formate solution, showing that PdO does not play a direct role in the mechanism of $H_2$ formation.

**Background Knowledge:**
- Palladium nanoparticles on carbon supports (Pd/C) are effective for catalyzing hydrogen release from aqueous formate solutions but typically suffer from a gradual decrease of activity.
- Nitrogen doping of carbon supports is observed to enhance the rates of hydrogen release from aqueous formate solutions

**Rubrics:** The response must state that palladium oxide (PdO) does not play a direct role as the active catalyst in the mechanism of $H_2$ formation

Chemistry: Multiple-Choice Question

**Paper Title:** Lab-Scale Thermal Decomposition of Hydrogen Peroxide as Green Propellant over Low-Cost Catalysts Based on Copper Deposited on Different Supports

**Link to The Paper:** https://www.mdpi.com/2226-4310/12/5/440

**Experimental Setup:** Researchers investigate the thermal degradation of the $H_2O_2$ green monopropellant. Three distinct catalysts—copper supported on $\gamma$-alumina, graphite, and MNC clay—were used. Conversely, a LABSYS evo-gasorption apparatus (Category: DTA/TG/DSC, Model: Setaram Instrumentation) was used to perform differential thermal analysis– thermogravimetry (DTA–TG) measurements in order to investigate the thermal breakdown of $H_2O_2$ at constant atmospheric pressure (p = 1 atm). A syringe was used to inject a 30% (w/w) $H_2O_2$ microdroplet into the metallic sample cell. It was investigated how the three different catalysts affected the $H_2O_2$ thermogram. A microdroplet of liquid $H_2O_2$ was combined with a modest amount (a few micrograms) of powdered catalyst in the aluminum sample cell for each thermal study. Before each run, the following experimental conditions were maintained:
(i) Carrier gas: argon, with a flow rate of 50 mL·min$^{-1}$;
(ii) Heating rate: 10 °C·min$^{-1}$, from room temperature up to 250 °C;
(iii) The $H_2O_2$ droplet was added directly to the catalyst particles already placed in the aluminum cell. After sealing the apparatus, a stabilization period of approximately 2 min was allowed for the system (carrier gas and sample) to equilibrate. The thermal run was then initiated to record the DTA–TG thermograms.
Experiments were run at two constant temperatures: 0 °C and 36 °C

**Measurements Taken:**
- Differential pressure ($\Delta P$, in kPa) vs time (minutes) was recorded.
- $\Delta P$ for each catalyst (Cu/$\gamma$-alumina, Cu/graphite, Cu/clay) compared to the uncatalyzed control.
- $\Delta P$ at 0 °C and 36 °C to assess temperature effects on decomposition rate.

**Outcome Prediction Question:** Which of the following statements best describes the observed catalytic activity (as measured by differential pressure, $\Delta P$, vs time) for the decomposition of 30 % $H_2O_2$ over the three copper-supported catalysts (Cu/$\gamma$-alumina, Cu/graphite, Cu/clay) compared to the uncatalyzed decomposition, at 36 °C and 0 °C?
A. At both temperatures all three catalysts produce rates almost identical to each other; the rates follow a similar trend, with 0°C just being slower than 36 °C, each gives a large increase over the uncatalyzed reaction at both temperatures.
B. At 0°C all three catalysts give a similar rate, none of them is clearly faster than another, but at 36 °C Cu/$\gamma$-alumina gives the highest rate (largest $\Delta P$ increase), followed by Cu/graphite, then Cu/clay, each significantly faster than uncatalyzed at both temperatures.
C. At 0 °C, Cu/clay a rate that is slower than the uncatalyzed reaction at the beginning, then becomes faster than the uncatalyzed reaction, while Cu/graphite, and Cu/$\gamma$-alumina have a similar rate and are higher than the uncatalyzed reaction. At 36 °C all three are faster than uncatalyzed reaction, Cu/$\gamma$-alumina is the fastest, closely followed by Cu/graphite, then Cu/clay.
D. At 0 °C all three catalysts begin slightly faster than the uncatalyzed reaction then all three become much faster, the variability being larger than the difference between the catalysts. At 36 °C the reaction with all three catalysts is much faster than the uncatalyzed reaction, with Cu/$\gamma$-alumina being much faster than Cu/graphite, then Cu/clay lags because the copper particles came off the support particles.

**Ground Truth Answer:** C

**Background Knowledge:**
- As the world increasingly focuses on sustainable and environmentally friendly solutions, there is a growing interest in exploring greener alternative propellants that offer comparable performance while mitigating the drawbacks associated with hydrazine and its derivatives.
- The thermal decomposition of hydrogen peroxide ($H_2O_2$) as a promising green propellant was performed over free-noble metallic-based catalysts deposited on abundant supports.
- Green monopropellants have the potential for long-term cost savings due to reduced safety measures, disposal costs, and regulatory compliance requirements associated with hazardous materials such as hydrazine.

**Paper Title:** Time-resolved photo-electrochemical measurements to study band bending of BiVO4 photoanodes

**Link to The Paper:** https://chemrxiv.org/engage/chemrxiv/article-details/68b1a2e2728bf9025e19a17e?

**Experimental Setup:** Thin-film $BiVO_4$ photoanodes were investigated in a three-electrode photo-electrochemical RRDE cell under chopped AM 1.5G illumination. Light switch-ON/OFF transients were recorded over 0–2.5 V vs RHE, and the disk photocurrent during switch-ON was fit with exponentials to isolate the fast space-charge reorganization time constant ($\tau$_fast) (along with slower components).

**Measurements Taken:**
- Disk photocurrent transients at light switch-ON/OFF (current vs time) across 0–2.5 V vs RHE.
- Exponential fits of transients to extract characteristic time constants (including $\tau$_fast) in seconds; report the average $\tau$_fast (switch-ON) over the potential window.
- Steady-state J–E curves under illumination.
- RRDE ring current (Pt ring) vs time/potential for $O_2$ detection/validation.
- Assignment of $\tau$_fast to space-charge reorganization based on transient behavior.

**Outcome Prediction Question:** Thin-film $BiVO_4$ photoanodes were tested in a three-electrode photo-electrochemical RRDE cell under chopped AM 1.5G illumination. During light "switch-ON" steps over 0–2.5 V vs RHE, the disk photocurrent transients were fit with exponentials to isolate the fast space-charge reorganization process ($\tau$_fast). At these conditions, what is the average value of $\tau$_fast in seconds (s) for the switch-ON process?

**Ground Truth Answer:** 0.0022±0.002 s.

**Background Knowledge:**
- $BiVO_4$ is a semiconductor photoanode used for oxygen evolution under illumination; its behavior is probed in a three-electrode photoelectrochemical cell.
- Band bending at the semiconductor/electrolyte interface creates a space-charge region that governs carrier separation and the early transient response.
- Time-resolved photoelectrochemistry with chopped AM 1.5G illumination measures photocurrent transients at light on/off to extract characteristic time constants.
- A rotating ring–disk electrode (RRDE) uses a Pt ring to detect dissolved $O_2$ produced at the disk, distinguishing disk photocurrent from ring current.
- The flat-band potential is the potential where band bending vanishes and is estimated from cyclic-voltammetry features; potentials are reported vs RHE.
- Exponential fitting of transients yields $\tau$_fast and slower components that reflect interfacial charge reorganization and reaction kinetics.

## B.2. Example human responses

---

**Physics: Numerical Value Question**

**Paper Title:** Recent Highlights from the STAR Experiment

**Link to The Paper:** https://arxiv.org/abs/2508.08444

**Experimental Setup:** Researchers investigated the Beam Energy Scan-II (BES-II) program at the STAR experiment, which was used to measure net-proton cumulant ratios in Gold-on-Gold (Au+Au) collisions at various center-of-mass energies (from 7.7 to 27 GeV) in the Fixed-Target mode. BES-II employed a new centrality definition, RefMult3X, corresponding to pseudorapidity acceptances fulfilling $|\eta| < 1.6$. The Time-Projection Chamber (TPC) for low transverse momentum ($0.4 < pT < 0.8$ GeV/c) and the Time-Of-Flight (TOF) detector for greater transverse momentum ($0.8 < pT < 2.0$ GeV/c) were used to identify protons and anti-protons. Only particles falling within the speed window of $|y| < 0.5$ were included in the analysis. The most central collisions (0-5% centrality class) were the focus of the measurements, which were methodically adjusted for experimental variables such detector efficiency, event pile-up, and centrality bin width.

**Measurements Taken:**
- Net-proton cumulants (C1, C2, C3, C4) as a function of collision centrality and collision energy.
- The relative dynamical correlation of transverse momentum as a function of collision energy.

**Outcome Prediction Question:** In the STAR experiment's Beam Energy Scan-II (BES-II), what was the measured value of the net-proton cumulant ratio C4/C2 at the collision energy of 19.6 GeV for the 0-5% centrality class?

**Ground Truth Answer:** [0.25-0.40]
Note: The range is informed graphically in Figure 3. The range was estimated by the pixel coordinates of the error bars and axis ticks.

**Background Knowledge:**
- The upgrades done to STAR for BES-II enabled a new centrality definition, RefMult3X, which achieves better centrality resolution due to larger multiplicity within the acceptance.
- Experimentally measured proton multiplicity distributions are described by the central moments, which depend on the cumulants. In particular, the second cumulant C2 is the variance $\sigma^2$, and the ratio between the fourth and second cumulant, C4/C2, is $\kappa\sigma^2$, where $\kappa$ is the kurtosis.
- When there are no intrinsic correlations among the measured particles, all ratios of the cumulants are unity, so Poisson statistics is a trivial baseline for experimentally measured cumulant ratios.

---

---

Human Responses

**Answer (NBK):** The measured value of the net-proton cumulant ratio $C_4/C_2$ at the collision energy of 19.6 GeV for the 0-5% centrality class is 0.4.

**Reasoning (NBK):** Theoretically, the ratio of the fourth- to the second-order net-proton cumulant ($C_4/C_2$) is often called the moment product $\kappa\sigma^2$. Here, $\kappa$ is the kurtosis and $\sigma^2$ is the variance. The theoretical Poisson baseline for net-proton cumulant ratios is unity or 1. Hence, the measured value must be $\leq 1$.
Additionally, various previous experiments support the fact that the $C_4/C_2$ value is close to unity for all collision energies for the smallest rapidity acceptance, and for higher collision energies. For example, according to Adam et al. (2021), during the BES-I experiment of the STAR detector at RHIC, the mean $C_4/C_2$ ratio in the 0-5% bin is ~0.4. This result is also supported by Bleicher et. al. (1999) during the Ultra-Relativistic Quantum Molecular Dynamics (UrQMD) experiment.

**Confidence (NBK):** Somewhat confident in your answer

**Difficulty (NBK):** Easy to answer

**Answer (BK):** The measured value of the net-proton cumulant ratio $C_4/C_2$ at the collision energy of 19.6 GeV for the 0-5% centrality class is 0.4.

**Reasoning (BK):** Theoretically, the ratio of the fourth- to the second-order net-proton cumulant ($C_4/C_2$) is often called the moment product $\kappa\sigma^2$. Here, $\kappa$ is the kurtosis and $\sigma^2$ is the variance. The theoretical Poisson baseline for net-proton cumulant ratios is unity or 1. Hence, the measured value must be $\leq 1$.
Additionally, various previous experiments support the fact that the $C_4/C_2$ value is close to unity for all collision energies for the smallest rapidity acceptance, and for higher collision energies. For example, according to Adam et al. (2021), during the BES-I experiment of the STAR detector at RHIC, the mean $C_4/C_2$ ratio in the 0-5% bin is ~0.4 [Figure 8]. This result is also supported by Bleicher et. al. (1999) during the Ultra-Relativistic Quantum Molecular Dynamics (UrQMD) experiment [Figures 6 and 30].

**Confidence (BK):** Somewhat confident in your answer

**Difficulty (BK):** Easy to answer

**Feasibility:** Very feasible to answer without running the experiment

**Feasibility Reasoning:** Theoretically, the ratio of the fourth- to the second-order net-proton cumulant ($C_4/C_2$) is often called the moment product $\kappa\sigma^2$. Here, $\kappa$ is the kurtosis and $\sigma^2$ is the variance. The theoretical Poisson baseline for net-proton cumulant ratios is unity or 1. Hence, the measured value must be $\leq 1$, which can be directly concluded from the known theory on this topic.
Additionally, various previous experiments support the fact that the $C_4/C_2$ value is close to unity for all collision energies for the smallest rapidity acceptance, and for higher collision energies. For example, according to Adam et al. (2021), during the BES-I experiment of the STAR detector at RHIC, the mean $C_4/C_2$ ratio in the 0-5% bin is ~0.4 [Figure 8]. This result is also supported by Bleicher et. al. (1999) during the Ultra-Relativistic Quantum Molecular Dynamics (UrQMD) experiment [Figures 6 and 30].
Hence, using the existing literature on previously performed experiments, the measured value of $C_4/C_2$ can be logically estimated for the BES-II experiment of the STAR detector at RHIC.

# C. Additional Results

*Table 3.* **Domain-wise performance across LLM families.** Different versions of Gemini, OpenAI, Claude, Llama, Qwen, and DeepSeek evaluated on *Chemistry, Biology, Physics*, and *All Domains*. **Conf.** := Confidence Score; **Diff.** := Difficulty Level; **Feas.** := Feasibility Score.

| Model | Experimental Setup | Chemistry Accuracy (%) | Chemistry Conf. | Chemistry Diff. | Chemistry Feas. | Biology Accuracy (%) | Biology Conf. | Biology Diff. | Biology Feas. | Physics Accuracy (%) | Physics Conf. | Physics Diff. | Physics Feas. | All Domains Accuracy (%) | All Domains Conf. | All Domains Diff. | All Domains Feas. |
|---|---|---|---|---|---|---|---|---|---|---|---|---|---|---|---|---|---|
| Gemini 3-pro | NBK | 26.14±5.40 | 4.37±0.01 | 3.56±0.03 | 3.55±0.04 | 24.47±1.24 | 4.38±0.02 | 3.44±0.01 | 3.55±0.09 | 26.00±0.00 | 4.56±0.02 | 3.39±0.01 | 3.71±0.10 | 25.27±1.92 | 4.42±0.01 | 3.46±0.01 | 3.59±0.06 |
| | BK | 28.43±0.98 | 4.39±0.03 | 3.52±0.02 | 3.50±0.07 | 27.26±0.75 | 4.39±0.02 | 3.40±0.07 | 3.59±0.03 | 26.33±2.08 | 4.56±0.01 | 3.25±0.06 | 3.89±0.06 | 27.33±0.79 | 4.43±0.01 | 3.40±0.04 | 3.64±0.02 |
| | SBK | 28.43±0.00 | 4.43±0.00 | 3.45±0.00 | 3.71±0.00 | 26.11±0.00 | 4.45±0.00 | 3.19±0.00 | 3.89±0.00 | 21.00±0.00 | 4.63±0.00 | 3.24±0.00 | 4.00±0.00 | 25.43±0.00 | 4.49±0.00 | 3.27±0.00 | 3.87±0.00 |
| | SABK | 30.39±0.00 | 4.48±0.00 | 3.42±0.00 | 3.79±0.00 | 25.12±0.00 | 4.44±0.00 | 3.20±0.00 | 3.78±0.00 | 27.00±0.00 | 4.68±0.00 | 3.14±0.00 | 4.01±0.00 | 26.91±0.00 | 4.51±0.00 | 3.24±0.00 | 3.84±0.00 |
| | FBK | 25.49±0.00 | 4.35±0.00 | 3.53±0.00 | 3.47±0.00 | 24.63±0.00 | 4.32±0.00 | 3.49±0.00 | 3.35±0.00 | 23.00±0.00 | 4.63±0.00 | 3.32±0.00 | 3.75±0.00 | 24.44±0.00 | 4.40±0.00 | 3.46±0.00 | 3.48±0.00 |
| Claude Opus 4.5 | NBK | 15.69±0.98 | 3.19±0.08 | 4.04±0.03 | 2.78±0.08 | 25.12±0.49 | 3.30±0.04 | 3.98±0.02 | 2.92±0.04 | 26.33±2.08 | 3.48±0.03 | 4.01±0.02 | 3.17±0.06 | 23.05±0.51 | 3.32±0.04 | 4.00±0.02 | 2.95±0.01 |
| | BK | 22.88±1.50 | 3.20±0.01 | 4.03±0.01 | 2.85±0.06 | 27.09±0.85 | 3.38±0.02 | 3.92±0.05 | 3.00±0.01 | 32.33±2.08 | 3.51±0.04 | 3.93±0.04 | 3.14±0.03 | 27.33±0.75 | 3.37±0.02 | 3.95±0.03 | 3.00±0.01 |
| | SBK | 17.65±0.00 | 3.27±0.00 | 4.01±0.00 | 2.86±0.00 | 27.00±0.00 | 3.46±0.00 | 3.81±0.00 | 3.12±0.00 | 29.00±0.00 | 3.51±0.00 | 3.90±0.00 | 3.23±0.00 | 25.14±0.00 | 3.43±0.00 | 3.88±0.00 | 3.08±0.00 |
| | SABK | 18.63±0.00 | 3.40±0.00 | 3.91±0.00 | 3.01±0.00 | 26.60±0.00 | 3.48±0.00 | 3.83±0.00 | 3.17±0.00 | 32.00±0.00 | 3.64±0.00 | 3.77±0.00 | 3.41±0.00 | 25.93±0.00 | 3.50±0.00 | 3.83±0.00 | 3.19±0.00 |
| | FBK | 18.63±0.00 | 3.19±0.00 | 4.02±0.00 | 2.80±0.00 | 26.60±0.00 | 3.28±0.00 | 3.97±0.00 | 2.95±0.00 | 32.00±0.00 | 3.39±0.00 | 3.97±0.00 | 3.12±0.00 | 25.93±0.00 | 3.28±0.00 | 3.98±0.00 | 2.95±0.00 |
| Claude Sonnet 4.5 | NBK | 23.86±1.50 | 3.98±0.02 | 3.77±0.01 | 3.24±0.05 | 22.66±0.85 | 4.04±0.02 | 3.63±0.02 | 3.47±0.03 | 21.00±2.00 | 4.14±0.04 | 3.74±0.02 | 3.39±0.03 | 22.55±0.75 | 4.05±0.01 | 3.69±0.01 | 3.40±0.01 |
| | BK | 26.80±1.50 | 4.05±0.03 | 3.75±0.03 | 3.31±0.02 | 28.08±2.15 | 4.06±0.02 | 3.57±0.03 | 3.54±0.03 | 24.67±1.53 | 4.10±0.03 | 3.67±0.02 | 3.47±0.06 | 26.91±1.23 | 4.07±0.02 | 3.64±0.02 | 3.47±0.02 |
| | SBK | 16.67±0.00 | 4.10±0.00 | 3.73±0.00 | 3.53±0.00 | 20.00±0.00 | 4.11±0.00 | 3.53±0.00 | 3.64±0.00 | 20.00±0.00 | 4.11±0.00 | 3.79±0.00 | 3.52±0.00 | 19.16±0.00 | 4.11±0.00 | 3.64±0.00 | 3.58±0.00 |
| | SABK | 19.61±0.00 | 4.11±0.00 | 3.75±0.00 | 3.42±0.00 | 25.12±0.00 | 4.11±0.00 | 3.52±0.00 | 3.66±0.00 | 29.00±0.00 | 4.06±0.00 | 3.82±0.00 | 3.47±0.00 | 24.69±0.00 | 4.10±0.00 | 3.65±0.00 | 3.55±0.00 |
| | FBK | 23.53±0.00 | 3.99±0.00 | 3.80±0.00 | 3.16±0.00 | 23.65±0.00 | 3.94±0.00 | 3.67±0.00 | 3.40±0.00 | 22.00±0.00 | 3.95±0.00 | 3.85±0.00 | 3.33±0.00 | 23.21±0.00 | 3.96±0.00 | 3.75±0.00 | 3.32±0.00 |
| Claude Opus 4.1 | NBK | 22.55±2.59 | 4.00±0.06 | 3.76±0.03 | 2.99±0.04 | 20.36±0.28 | 4.01±0.01 | 3.61±0.01 | 3.20±0.00 | 25.67±4.16 | 4.09±0.03 | 3.77±0.01 | 3.29±0.02 | 22.22±1.48 | 4.03±0.01 | 3.69±0.01 | 3.17±0.01 |
| | BK | 22.55±0.00 | 4.02±0.03 | 3.74±0.02 | 3.10±0.07 | 26.11±1.71 | 4.05±0.01 | 3.54±0.02 | 3.28±0.04 | 27.00±1.00 | 4.11±0.01 | 3.66±0.03 | 3.37±0.02 | 25.43±0.65 | 4.05±0.01 | 3.62±0.02 | 3.26±0.02 |
| | SBK | 24.51±0.00 | 4.10±0.00 | 3.69±0.00 | 3.29±0.00 | 20.81±0.00 | 4.15±0.00 | 3.48±0.00 | 3.50±0.00 | 24.00±0.00 | 4.16±0.00 | 3.67±0.00 | 3.57±0.00 | 22.53±0.00 | 4.14±0.00 | 3.58±0.00 | 3.46±0.00 |
| | SABK | 28.43±0.00 | 4.09±0.00 | 3.74±0.00 | 3.27±0.00 | 22.17±0.00 | 4.13±0.00 | 3.47±0.00 | 3.55±0.00 | 25.00±0.00 | 4.13±0.00 | 3.65±0.00 | 3.64±0.00 | 24.44±0.00 | 4.12±0.00 | 3.58±0.00 | 3.50±0.00 |
| | FBK | 20.59±0.00 | 3.97±0.00 | 3.76±0.00 | 2.96±0.00 | 22.17±0.00 | 3.97±0.00 | 3.67±0.00 | 3.15±0.00 | 23.00±0.00 | 3.98±0.00 | 3.78±0.00 | 3.39±0.00 | 21.98±0.00 | 3.97±0.00 | 3.72±0.00 | 3.16±0.00 |
| Gemini 3-Flash | NBK | 22.88±1.50 | 4.43±0.03 | 3.58±0.01 | 4.23±0.01 | 22.33±2.48 | 4.37±0.01 | 3.32±0.04 | 4.32±0.02 | 21.33±2.31 | 4.47±0.03 | 3.45±0.03 | 4.34±0.01 | 22.22±1.08 | 4.41±0.02 | 3.42±0.02 | 4.30±0.01 |
| | BK | 24.84±4.08 | 4.42±0.03 | 3.56±0.04 | 4.24±0.00 | 23.97±1.24 | 4.41±0.01 | 3.25±0.02 | 4.35±0.02 | 23.00±1.00 | 4.52±0.04 | 3.39±0.08 | 4.37±0.05 | 23.95±1.62 | 4.44±0.01 | 3.36±0.02 | 4.33±0.01 |
| | SBK | 23.53±0.00 | 4.50±0.00 | 3.48±0.00 | 4.26±0.00 | 21.78±0.00 | 4.47±0.00 | 3.19±0.00 | 4.41±0.00 | 20.83±0.00 | 4.51±0.00 | 3.38±0.00 | 4.36±0.00 | 21.99±0.00 | 4.49±0.00 | 3.31±0.00 | 4.36±0.00 |
| | SABK | 28.43±0.00 | 4.48±0.00 | 3.49±0.00 | 4.28±0.00 | 24.14±0.00 | 4.50±0.00 | 3.13±0.00 | 4.43±0.00 | 28.00±0.00 | 4.55±0.00 | 3.40±0.00 | 4.41±0.00 | 26.17±0.00 | 4.51±0.00 | 3.29±0.00 | 4.39±0.00 |
| | FBK | 29.41±0.00 | 4.40±0.00 | 3.58±0.00 | 4.21±0.00 | 23.65±0.00 | 4.34±0.00 | 3.37±0.00 | 4.28±0.00 | 21.00±0.00 | 4.44±0.00 | 3.51±0.00 | 4.29±0.00 | 24.44±0.00 | 4.38±0.00 | 3.46±0.00 | 4.26±0.00 |
| OpenAI GPT-5.2 | NBK | 18.95±2.04 | 3.53±0.04 | 3.49±0.01 | 3.61±0.05 | 20.69±1.48 | 3.59±0.02 | 3.37±0.03 | 3.60±0.02 | 22.00±1.73 | 3.61±0.02 | 3.37±0.03 | 3.67±0.05 | 20.58±1.03 | 3.58±0.02 | 3.40±0.03 | 3.62±0.03 |
| | BK | 19.93±0.57 | 3.59±0.06 | 3.43±0.01 | 3.69±0.01 | 25.78±2.80 | 3.73±0.02 | 3.27±0.01 | 3.74±0.02 | 19.67±1.53 | 3.63±0.07 | 3.27±0.01 | 3.70±0.05 | 22.80±1.79 | 3.67±0.02 | 3.31±0.01 | 3.72±0.01 |
| | SBK | 17.65±0.00 | 3.60±0.00 | 3.36±0.00 | 3.76±0.00 | 18.72±0.00 | 3.70±0.00 | 3.20±0.00 | 3.79±0.00 | 21.00±0.00 | 3.60±0.00 | 3.32±0.00 | 3.71±0.00 | 19.01±0.00 | 3.65±0.00 | 3.27±0.00 | 3.76±0.00 |
| | SABK | 18.63±0.00 | 3.62±0.00 | 3.34±0.00 | 3.76±0.00 | 24.63±0.00 | 3.79±0.00 | 3.17±0.00 | 3.86±0.00 | 24.00±0.00 | 3.55±0.00 | 3.28±0.00 | 3.72±0.00 | 22.96±0.00 | 3.69±0.00 | 3.24±0.00 | 3.80±0.00 |
| | FBK | 20.59±0.00 | 3.49±0.00 | 3.49±0.00 | 3.58±0.00 | 19.70±0.00 | 3.60±0.00 | 3.37±0.00 | 3.56±0.00 | 18.00±0.00 | 3.64±0.00 | 3.42±0.00 | 3.68±0.00 | 19.51±0.00 | 3.58±0.00 | 3.41±0.00 | 3.60±0.00 |
| **Human Baseline** | NBK | 8.82 | 2.59 | 3.78 | 2.52 | 23.15 | 3.13 | 3.39 | 2.92 | 26.00 | 3.07 | 3.26 | 3.05 | 20.25 | 2.98 | 3.46 | 2.85 |
| | BK | 9.80 | 2.65 | 3.78 | 2.52 | 23.65 | 3.22 | 3.32 | 2.92 | 27.00 | 3.30 | 3.24 | 3.05 | 20.99 | 3.10 | 3.42 | 2.85 |
| OpenAI O3-mini | NBK | 19.28±5.91 | 4.34±0.02 | 3.25±0.06 | 4.41±0.01 | 21.02±1.50 | 4.38±0.02 | 3.08±0.04 | 4.46±0.03 | 18.00±2.00 | 4.42±0.05 | 3.16±0.03 | 4.46±0.01 | 19.84±1.49 | 4.38±0.01 | 3.14±0.03 | 4.44±0.02 |
| | BK | 21.24±2.26 | 4.42±0.03 | 3.16±0.05 | 4.48±0.05 | 22.00±1.03 | 4.48±0.01 | 2.94±0.02 | 4.56±0.00 | 20.33±0.58 | 4.51±0.03 | 3.07±0.04 | 4.53±0.03 | 21.40±0.87 | 4.47±0.01 | 3.02±0.03 | 4.53±0.02 |
| | SBK | 11.70±0.00 | 4.45±0.00 | 3.11±0.00 | 4.45±0.00 | 16.08±0.00 | 4.46±0.00 | 2.87±0.00 | 4.57±0.00 | 16.00±0.00 | 4.49±0.00 | 3.16±0.00 | 4.51±0.00 | 14.96±0.00 | 4.46±0.00 | 3.00±0.00 | 4.52±0.00 |
| | SABK | 19.61±0.00 | 4.33±0.00 | 3.12±0.00 | 4.43±0.00 | 17.24±0.00 | 4.48±0.00 | 2.85±0.00 | 4.56±0.00 | 26.00±0.00 | 4.52±0.00 | 3.00±0.00 | 4.58±0.00 | 20.00±0.00 | 4.45±0.00 | 2.95±0.00 | 4.53±0.00 |
| | FBK | 17.65±0.00 | 4.29±0.00 | 3.25±0.00 | 4.39±0.00 | 19.70±0.00 | 4.36±0.00 | 3.11±0.00 | 4.45±0.00 | 19.00±0.00 | 4.42±0.00 | 3.20±0.00 | 4.44±0.00 | 19.01±0.00 | 4.36±0.00 | 3.17±0.00 | 4.43±0.00 |
| DeepSeek v3 | NBK | 16.99±2.47 | 4.35±0.07 | 3.64±0.17 | 4.34±0.09 | 19.21±1.71 | 4.47±0.02 | 3.52±0.01 | 4.37±0.06 | 21.33±2.52 | 4.63±0.03 | 3.74±0.10 | 4.33±0.04 | 19.18±0.79 | 4.48±0.03 | 3.60±0.06 | 4.35±0.04 |
| | BK | 18.63±7.40 | 4.49±0.02 | 3.52±0.07 | 4.23±0.02 | 22.82±0.57 | 4.55±0.02 | 3.36±0.02 | 4.44±0.08 | 23.67±1.53 | 4.64±0.05 | 3.54±0.06 | 4.35±0.08 | 21.98±2.36 | 4.56±0.00 | 3.45±0.01 | 4.37±0.06 |
| | SBK | 13.73±0.00 | 4.51±0.00 | 3.71±0.00 | 4.26±0.00 | 18.32±0.00 | 4.60±0.00 | 3.51±0.00 | 4.39±0.00 | 22.22±0.00 | 4.59±0.00 | 3.76±0.00 | 4.44±0.00 | 18.12±0.00 | 4.58±0.00 | 3.62±0.00 | 4.37±0.00 |
| | SABK | 16.67±0.00 | 4.38±0.00 | 3.72±0.00 | 4.32±0.00 | 19.70±0.00 | 4.61±0.00 | 3.32±0.00 | 4.39±0.00 | 24.00±0.00 | 4.53±0.00 | 3.80±0.00 | 4.45±0.00 | 20.00±0.00 | 4.53±0.00 | 3.54±0.00 | 4.39±0.00 |
| | FBK | 16.67±0.00 | 4.31±0.00 | 3.79±0.00 | 4.34±0.00 | 21.67±0.00 | 4.49±0.00 | 3.39±0.00 | 4.48±0.00 | 23.00±0.00 | 4.47±0.00 | 3.79±0.00 | 4.30±0.00 | 20.74±0.00 | 4.44±0.00 | 3.59±0.00 | 4.40±0.00 |
| Llama 3.3 70B | NBK | 16.99±2.26 | 3.53±0.03 | 3.65±0.03 | 3.44±0.07 | 19.54±0.28 | 3.51±0.05 | 3.68±0.02 | 3.38±0.00 | 16.67±0.58 | 3.47±0.02 | 3.72±0.04 | 3.31±0.04 | 18.19±0.71 | 3.50±0.03 | 3.68±0.01 | 3.38±0.01 |
| | BK | 21.57±2.59 | 3.54±0.03 | 3.61±0.04 | 3.53±0.05 | 19.87±1.03 | 3.65±0.03 | 3.62±0.01 | 3.55±0.03 | 18.00±1.00 | 3.53±0.09 | 3.60±0.01 | 3.42±0.08 | 19.84±0.29 | 3.59±0.02 | 3.61±0.02 | 3.51±0.01 |
| | SBK | 14.14±0.00 | 3.64±0.00 | 3.57±0.00 | 3.54±0.00 | 18.59±0.00 | 3.63±0.00 | 3.62±0.00 | 3.43±0.00 | 14.43±0.00 | 3.47±0.00 | 3.73±0.00 | 3.37±0.00 | 16.44±0.00 | 3.59±0.00 | 3.63±0.00 | 3.44±0.00 |
| | SABK | 19.61±0.00 | 3.59±0.00 | 3.55±0.00 | 3.60±0.00 | 19.21±0.00 | 3.71±0.00 | 3.58±0.00 | 3.60±0.00 | 15.00±0.00 | 3.62±0.00 | 3.64±0.00 | 3.56±0.00 | 18.27±0.00 | 3.66±0.00 | 3.59±0.00 | 3.59±0.00 |
| | FBK | 18.63±0.00 | 3.50±0.00 | 3.70±0.00 | 3.33±0.00 | 18.72±0.00 | 3.50±0.00 | 3.75±0.00 | 3.34±0.00 | 18.00±0.00 | 3.43±0.00 | 3.63±0.00 | 3.29±0.00 | 18.52±0.00 | 3.48±0.00 | 3.71±0.00 | 3.33±0.00 |
| OpenAI O3 | NBK | 17.32±2.26 | 3.91±0.01 | 3.12±0.06 | 3.95±0.02 | 18.23±2.26 | 3.92±0.01 | 3.01±0.01 | 4.00±0.00 | 18.00±1.73 | 3.88±0.04 | 3.14±0.01 | 3.95±0.01 | 17.94±1.27 | 3.91±0.01 | 3.07±0.02 | 3.98±0.01 |
| | BK | 18.63±5.96 | 3.91±0.04 | 3.08±0.02 | 3.99±0.04 | 23.48±1.73 | 3.95±0.01 | 2.99±0.01 | 4.02±0.02 | 23.00±1.00 | 3.88±0.03 | 3.07±0.04 | 4.00±0.02 | 22.14±2.47 | 3.92±0.02 | 3.03±0.01 | 4.01±0.00 |
| | SBK | 21.57±0.00 | 3.93±0.00 | 3.00±0.00 | 4.05±0.00 | 20.20±0.00 | 3.99±0.00 | 2.92±0.00 | 4.07±0.00 | 17.00±0.00 | 3.92±0.00 | 3.12±0.00 | 4.04±0.00 | 19.75±0.00 | 3.96±0.00 | 2.99±0.00 | 4.06±0.00 |
| | SABK | 18.63±0.00 | 3.93±0.00 | 3.09±0.00 | 4.01±0.00 | 20.69±0.00 | 3.98±0.00 | 2.93±0.00 | 4.08±0.00 | 21.00±0.00 | 3.95±0.00 | 3.04±0.00 | 4.06±0.00 | 20.25±0.00 | 3.96±0.00 | 3.00±0.00 | 4.06±0.00 |
| | FBK | 17.65±0.00 | 3.81±0.00 | 3.12±0.00 | 3.88±0.00 | 18.23±0.00 | 3.94±0.00 | 3.04±0.00 | 3.98±0.00 | 13.00±0.00 | 3.86±0.00 | 3.15±0.00 | 3.99±0.00 | 16.79±0.00 | 3.89±0.00 | 3.09±0.00 | 3.96±0.00 |
| Qwen 3 32B | NBK | 15.69±1.96 | 3.86±0.04 | 3.95±0.04 | 3.65±0.06 | 17.41±0.75 | 3.88±0.01 | 3.83±0.05 | 3.73±0.05 | 17.67±1.53 | 3.83±0.05 | 3.90±0.04 | 3.68±0.10 | 17.04±0.49 | 3.86±0.00 | 3.88±0.02 | 3.70±0.04 |
| | BK | 19.93±2.47 | 3.88±0.07 | 3.88±0.03 | 3.74±0.07 | 18.06±1.42 | 3.96±0.03 | 3.79±0.02 | 3.86±0.02 | 20.33±0.58 | 3.94±0.02 | 3.82±0.03 | 3.86±0.05 | 19.09±1.00 | 3.94±0.01 | 3.82±0.01 | 3.83±0.01 |
| | SBK | 13.75±0.00 | 3.73±0.00 | 3.89±0.00 | 3.61±0.00 | 18.29±0.00 | 3.99±0.00 | 3.79±0.00 | 3.92±0.00 | 15.48±0.00 | 3.99±0.00 | 3.88±0.00 | 3.88±0.00 | 16.45±0.00 | 3.93±0.00 | 3.84±0.00 | 3.83±0.00 |
| | SABK | 14.71±0.00 | 4.00±0.00 | 3.90±0.00 | 3.78±0.00 | 16.26±0.00 | 3.97±0.00 | 3.71±0.00 | 3.86±0.00 | 21.00±0.00 | 3.87±0.00 | 3.86±0.00 | 3.80±0.00 | 17.04±0.00 | 3.95±0.00 | 3.80±0.00 | 3.82±0.00 |
| | FBK | 13.73±0.00 | 3.86±0.00 | 3.92±0.00 | 3.69±0.00 | 19.21±0.00 | 3.82±0.00 | 3.87±0.00 | 3.65±0.00 | 16.00±0.00 | 3.82±0.00 | 3.89±0.00 | 3.69±0.00 | 17.04±0.00 | 3.83±0.00 | 3.89±0.00 | 3.67±0.00 |
| Gemini 2.5-pro | NBK | 17.32±2.04 | 4.67±0.03 | 2.84±0.04 | 4.57±0.06 | 16.26±1.30 | 4.60±0.00 | 2.67±0.02 | 4.55±0.02 | 18.33±2.08 | 4.75±0.03 | 2.97±0.02 | 4.65±0.02 | 17.04±0.65 | 4.66±0.01 | 2.79±0.04 | 4.58±0.01 |
| | BK | 20.92±3.44 | 4.64±0.01 | 2.94±0.03 | 4.50±0.00 | 24.14±2.61 | 4.59±0.02 | 2.68±0.06 | 4.51±0.04 | 22.33±2.08 | 4.76±0.03 | 2.86±0.06 | 4.64±0.04 | 22.88±0.14 | 4.64±0.02 | 2.79±0.04 | 4.54±0.03 |
| | SBK | 21.65±0.00 | 4.71±0.00 | 2.72±0.00 | 4.67±0.00 | 20.50±0.00 | 4.63±0.00 | 2.57±0.00 | 4.55±0.00 | 19.59±0.00 | 4.82±0.00 | 2.80±0.00 | 4.66±0.00 | 20.56±0.00 | 4.70±0.00 | 2.67±0.00 | 4.61±0.00 |
| | SABK | 20.59±0.00 | 4.71±0.00 | 2.81±0.00 | 4.60±0.00 | 19.21±0.00 | 4.70±0.00 | 2.51±0.00 | 4.66±0.00 | 17.00±0.00 | 4.77±0.00 | 2.78±0.00 | 4.69±0.00 | 19.01±0.00 | 4.72±0.00 | 2.66±0.00 | 4.65±0.00 |
| | FBK | 18.63±0.00 | 4.65±0.00 | 2.97±0.00 | 4.51±0.00 | 15.76±0.00 | 4.56±0.00 | 2.75±0.00 | 4.45±0.00 | 23.00±0.00 | 4.74±0.00 | 2.91±0.00 | 4.67±0.00 | 18.27±0.00 | 4.62±0.00 | 2.84±0.00 | 4.52±0.00 |
| Qwen 3 235B | NBK | 13.73±0.00 | 4.72±0.02 | 2.28±0.04 | 4.52±0.04 | 18.23±1.48 | 4.71±0.02 | 2.18±0.05 | 4.60±0.00 | 16.33±1.53 | 4.75±0.03 | 2.27±0.02 | 4.61±0.00 | 16.63±0.38 | 4.73±0.02 | 2.23±0.03 | 4.59±0.02 |
| | BK | 17.32±2.04 | 4.75±0.02 | 2.24±0.01 | 4.62±0.02 | 20.85±1.03 | 4.81±0.05 | 2.06±0.06 | 4.76±0.06 | 22.33±1.53 | 4.79±0.02 | 2.25±0.02 | 4.69±0.04 | 20.33±1.27 | 4.79±0.03 | 2.15±0.03 | 4.71±0.03 |
| | SBK | 14.71±0.00 | 4.76±0.00 | 2.17±0.00 | 4.64±0.00 | 19.21±0.00 | 4.83±0.00 | 2.00±0.00 | 4.83±0.00 | 16.16±0.00 | 4.79±0.00 | 2.21±0.00 | 4.66±0.00 | 17.33±0.00 | 4.80±0.00 | 2.09±0.00 | 4.74±0.00 |
| | SABK | 15.69±0.00 | 4.81±0.00 | 2.16±0.00 | 4.73±0.00 | 21.18±0.00 | 4.87±0.00 | 1.97±0.00 | 4.84±0.00 | 21.21±0.00 | 4.81±0.00 | 2.17±0.00 | 4.73±0.00 | 19.80±0.00 | 4.84±0.00 | 2.06±0.00 | 4.79±0.00 |
| | FBK | 12.75±0.00 | 4.61±0.00 | 2.35±0.00 | 4.45±0.00 | 21.18±0.00 | 4.55±0.00 | 2.30±0.00 | 4.45±0.00 | 19.00±0.00 | 4.75±0.00 | 2.32±0.00 | 4.63±0.00 | 18.52±0.00 | 4.62±0.00 | 2.32±0.00 | 4.49±0.00 |
| OpenAI O4-mini | NBK | 11.11±2.47 | 3.83±0.04 | 2.77±0.01 | 3.57±0.12 | 19.38±3.21 | 3.82±0.03 | 2.68±0.00 | 3.58±0.04 | 15.00±1.00 | 3.81±0.04 | 2.79±0.08 | 3.63±0.02 | 16.21±2.10 | 3.82±0.01 | 2.73±0.01 | 3.59±0.05 |
| | BK | 15.03±3.96 | 3.92±0.05 | 2.77±0.02 | 3.73±0.09 | 23.15±1.78 | 3.94±0.00 | 2.62±0.04 | 3.77±0.05 | 19.33±5.77 | 3.83±0.07 | 2.70±0.03 | 3.72±0.00 | 20.16±2.53 | 3.91±0.03 | 2.68±0.02 | 3.75±0.03 |
| | SBK | 13.73±0.00 | 3.93±0.00 | 2.75±0.00 | 3.85±0.00 | 16.75±0.00 | 3.97±0.00 | 2.63±0.00 | 3.85±0.00 | 10.00±0.00 | 3.88±0.00 | 2.78±0.00 | 3.91±0.00 | 14.32±0.00 | 3.94±0.00 | 2.70±0.00 | 3.86±0.00 |
| | SABK | 16.67±0.00 | 3.85±0.00 | 2.71±0.00 | 3.83±0.00 | 21.67±0.00 | 4.02±0.00 | 2.61±0.00 | 3.86±0.00 | 16.00±0.00 | 3.95±0.00 | 2.79±0.00 | 3.83±0.00 | 19.01±0.00 | 3.96±0.00 | 2.68±0.00 | 3.85±0.00 |
| | FBK | 14.71±0.00 | 3.85±0.00 | 2.81±0.00 | 3.68±0.00 | 21.18±0.00 | 3.85±0.00 | 2.69±0.00 | 3.65±0.00 | 17.00±0.00 | 3.77±0.00 | 2.84±0.00 | 3.66±0.00 | 18.52±0.00 | 3.83±0.00 | 2.76±0.00 | 3.66±0.00 |
| Llama 3.1 8B | NBK | 13.40±2.04 | 4.22±0.06 | 3.45±0.05 | 3.94±0.10 | 16.42±2.71 | 4.26±0.02 | 3.49±0.10 | 3.95±0.04 | 12.33±1.62 | 4.20±0.04 | 3.48±0.08 | 3.97±0.06 | 14.65±1.03 | 4.23±0.01 | 3.48±0.06 | 3.95±0.06 |
| | BK | 10.78±2.59 | 4.25±0.11 | 3.40±0.06 | 4.06±0.02 | 17.08±2.22 | 4.24±0.01 | 3.46±0.09 | 3.93±0.07 | 18.33±0.58 | 4.22±0.04 | 3.39±0.09 | 3.99±0.05 | 15.80±0.86 | 4.24±0.03 | 3.43±0.05 | 3.98±0.03 |
| | SBK | 9.80±0.00 | 4.29±0.00 | 3.28±0.00 | 4.05±0.00 | 16.75±0.00 | 4.21±0.00 | 3.49±0.00 | 3.97±0.00 | 17.00±0.00 | 4.22±0.00 | 3.50±0.00 | 4.01±0.00 | 15.06±0.00 | 4.23±0.00 | 3.44±0.00 | 4.00±0.00 |
| | SABK | 14.71±0.00 | 4.32±0.00 | 3.34±0.00 | 4.07±0.00 | 16.75±0.00 | 4.25±0.00 | 3.47±0.00 | 3.96±0.00 | 11.00±0.00 | 4.38±0.00 | 3.33±0.00 | 4.02±0.00 | 14.81±0.00 | 4.30±0.00 | 3.40±0.00 | 4.00±0.00 |
| | FBK | 15.69±0.00 | 4.19±0.00 | 3.43±0.00 | 3.98±0.00 | 16.75±0.00 | 4.19±0.00 | 3.52±0.00 | 3.91±0.00 | 15.00±0.00 | 4.31±0.00 | 3.56±0.00 | 3.88±0.00 | 16.05±0.00 | 4.22±0.00 | 3.51±0.00 | 3.92±0.00 |

All experiments reported in this work were conducted with web search capabilities disabled for all evaluated models. This design choice is critical to ensure our benchmark measures genuine predictive reasoning rather than information retrieval. Since our evaluation draws from papers published after March 2025, beyond the training cutoff of current frontier models, enabling web search would allow models to potentially locate and access the original publications, thereby converting the prediction task into a lookup task. This would fundamentally undermine our goal of assessing whether models can reason

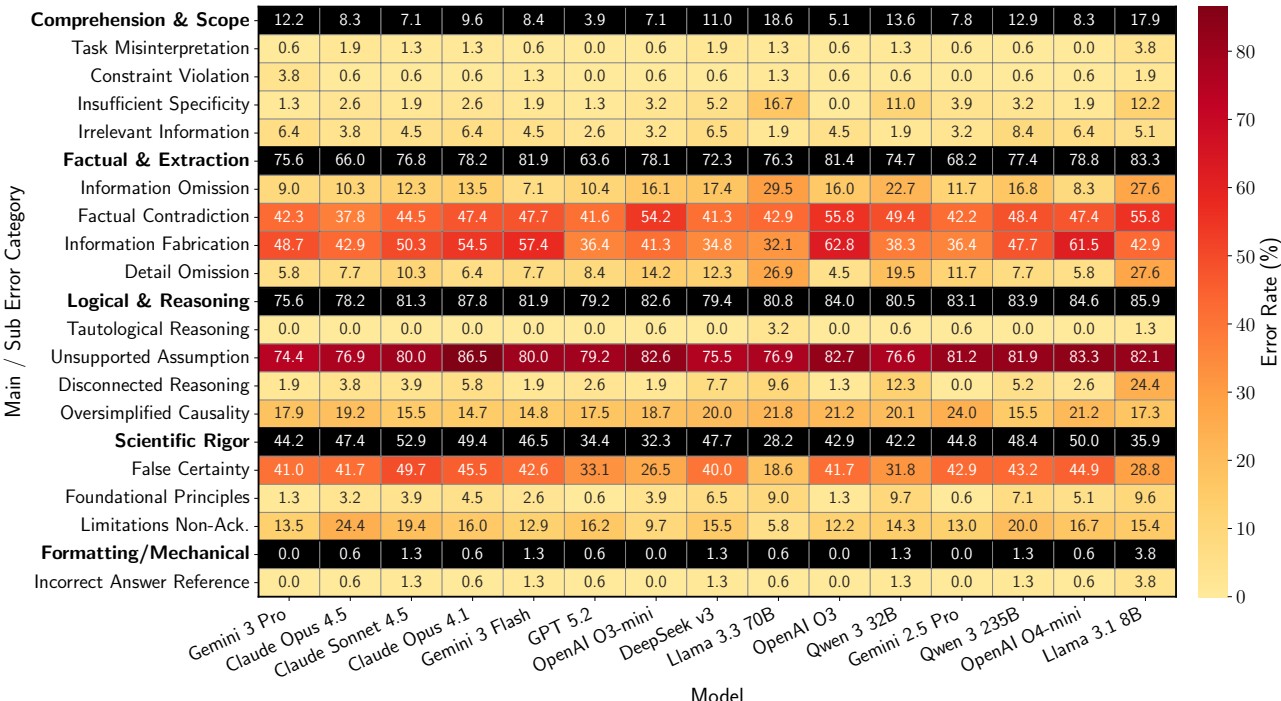

*Figure 13.* **Analysis of model errors for high feasible questions.** We employ an LLM judge to systematically classify errors in model predictions according to a hierarchical taxonomy spanning five top-level (in black background) categories and 16 specific error types. The heatmap shows the percentage of incorrect responses containing each error type for each evaluated model. Error categories progress from surface-level issues (Comprehension & Scope) to deeper reasoning failures (Logical & Reasoning Flaws) to fundamental scientific deficiencies (Deficiencies in Scientific Rigor). Models can exhibit multiple error types simultaneously, so accumulative percentage scores within top-level categories may exceed 100%. SciPredict tasks contribute to top-level category percentages if flagged with at least one underlying error type. Error analysis only considers the questions human experts marked as feasible to answer without running the practical experiment. Fig. 8 shows the same chart for all tasks. Tab. 8 provides comprehensive definitions for the error categories.

about experimental outcomes from first principles and provided context. By disabling web search, we ensure that model predictions reflect only their parametric knowledge, reasoning capabilities, and ability to leverage the provided experimental details and background knowledge, rather than their capacity to search for and retrieve the ground truth answers.

*Table 4.* **Question-format performance across LLM families.** Different versions of Gemini, OpenAI, Claude (Opus/Sonnet), Llama, Qwen, and DeepSeek evaluated across question formats. **Conf.** := Confidence Score; **Diff.** := Difficulty Level; **Feas.** := Feasibility Score.

| Model | Experimental Setup | MCQ Accuracy (%) | MCQ Calibration (1-5) Conf. | MCQ Calibration (1-5) Diff. | MCQ Calibration (1-5) Feas. | Numerical Accuracy (%) | Numerical Calibration (1-5) Conf. | Numerical Calibration (1-5) Diff. | Numerical Calibration (1-5) Feas. | Free form Accuracy Partial | Free form Accuracy Full | Free form Calibration (1-5) Conf. | Free form Calibration (1-5) Diff. | Free form Calibration (1-5) Feas. |
|---|---|---|---|---|---|---|---|---|---|---|---|---|---|---|
| | NBK | 36.21 ± 1.78 | 4.42 ± 0.03 | 3.45 ± 0.01 | 3.89 ± 0.06 | 12.80 ± 2.58 | 4.19 ± 0.05 | 4.01 ± 0.03 | 2.46 ± 0.11 | 37.70 ± 1.65 | 22.39 ± 2.45 | 4.62 ± 0.01 | 3.01 ± 0.03 | 4.19 ± 0.09 |
| Gemini 3-pro | BK | 42.39 ± 1.55 | 4.46 ± 0.01 | 3.36 ± 0.05 | 3.96 ± 0.02 | 12.80 ± 1.36 | 4.16 ± 0.03 | 4.03 ± 0.02 | 2.47 ± 0.06 | 36.04 ± 0.78 | 21.12 ± 2.33 | 4.62 ± 0.03 | 2.90 ± 0.05 | 4.25 ± 0.03 |
| | SBK | 37.04 ± 0.00 | 4.52 ± 0.00 | 3.17 ± 0.00 | 4.15 ± 0.00 | 10.71 ± 0.00 | 4.19 ± 0.00 | 3.98 ± 0.00 | 2.77 ± 0.00 | 38.90 ± 0.00 | 23.66 ± 0.00 | 4.71 ± 0.00 | 2.78 ± 0.00 | 4.47 ± 0.00 |
| | SABK | 41.36 ± 0.00 | 4.55 ± 0.00 | 3.17 ± 0.00 | 4.15 ± 0.00 | 12.50 ± 0.00 | 4.24 ± 0.00 | 3.91 ± 0.00 | 2.75 ± 0.00 | 35.90 ± 0.00 | 21.37 ± 0.00 | 4.69 ± 0.00 | 2.76 ± 0.00 | 4.39 ± 0.00 |
| | FBK | 38.89 ± 0.00 | 4.41 ± 0.00 | 3.44 ± 0.00 | 3.81 ± 0.00 | 8.93 ± 0.00 | 4.14 ± 0.00 | 4.04 ± 0.00 | 2.28 ± 0.00 | 36.41 ± 0.00 | 19.85 ± 0.00 | 4.62 ± 0.00 | 2.98 ± 0.00 | 4.10 ± 0.00 |
| | NBK | 33.54 ± 0.71 | 3.78 ± 0.05 | 3.88 ± 0.02 | 3.16 ± 0.01 | 13.99 ± 0.52 | 2.18 ± 0.03 | 4.59 ± 0.04 | 2.00 ± 0.02 | 34.69 ± 0.43 | 17.81 ± 1.17 | 3.74 ± 0.06 | 3.64 ± 0.01 | 3.50 ± 0.03 |
| Claude Opus 4.5 | BK | 39.09 ± 0.94 | 3.90 ± 0.03 | 3.80 ± 0.05 | 3.27 ± 0.02 | 15.77 ± 2.73 | 2.11 ± 0.02 | 4.58 ± 0.01 | 1.96 ± 0.02 | 38.37 ± 0.62 | 22.65 ± 1.17 | 3.79 ± 0.01 | 3.60 ± 0.03 | 3.54 ± 0.02 |
| | SBK | 36.88 ± 0.00 | 3.95 ± 0.00 | 3.77 ± 0.00 | 3.34 ± 0.00 | 15.18 ± 0.00 | 2.23 ± 0.00 | 4.52 ± 0.00 | 2.08 ± 0.00 | 35.13 ± 0.00 | 19.23 ± 0.00 | 3.81 ± 0.00 | 3.47 ± 0.00 | 3.61 ± 0.00 |
| | SABK | 35.80 ± 0.00 | 3.98 ± 0.00 | 3.69 ± 0.00 | 3.54 ± 0.00 | 17.86 ± 0.00 | 2.33 ± 0.00 | 4.53 ± 0.00 | 2.15 ± 0.00 | 36.34 ± 0.00 | 20.61 ± 0.00 | 3.91 ± 0.00 | 3.42 ± 0.00 | 3.64 ± 0.00 |
| | FBK | 37.04 ± 0.00 | 3.72 ± 0.00 | 3.85 ± 0.00 | 3.17 ± 0.00 | 16.96 ± 0.00 | 2.06 ± 0.00 | 4.59 ± 0.00 | 1.99 ± 0.00 | 34.88 ± 0.00 | 19.85 ± 0.00 | 3.80 ± 0.00 | 3.62 ± 0.00 | 3.52 ± 0.00 |
| | NBK | 29.01 ± 1.63 | 4.22 ± 0.03 | 3.60 ± 0.05 | 3.65 ± 0.01 | 16.07 ± 0.89 | 3.59 ± 0.07 | 4.15 ± 0.05 | 2.38 ± 0.01 | 35.75 ± 2.22 | 20.10 ± 1.17 | 4.23 ± 0.01 | 3.42 ± 0.01 | 3.94 ± 0.03 |
| Claude Sonnet 4.5 | BK | 36.01 ± 1.28 | 4.26 ± 0.02 | 3.55 ± 0.05 | 3.80 ± 0.04 | 15.77 ± 2.58 | 3.56 ± 0.04 | 4.12 ± 0.02 | 2.33 ± 0.05 | 40.83 ± 0.40 | 25.19 ± 3.05 | 4.26 ± 0.01 | 3.35 ± 0.00 | 4.01 ± 0.04 |
| | SBK | 25.62 ± 0.00 | 4.29 ± 0.00 | 3.50 ± 0.00 | 3.95 ± 0.00 | 11.93 ± 0.00 | 3.70 ± 0.00 | 4.11 ± 0.00 | 2.61 ± 0.00 | 36.26 ± 0.00 | 17.19 ± 0.00 | 4.23 ± 0.00 | 3.44 ± 0.00 | 3.95 ± 0.00 |
| | SABK | 33.33 ± 0.00 | 4.27 ± 0.00 | 3.55 ± 0.00 | 3.82 ± 0.00 | 13.39 ± 0.00 | 3.65 ± 0.00 | 4.13 ± 0.00 | 2.56 ± 0.00 | 38.91 ± 0.00 | 23.66 ± 0.00 | 4.26 ± 0.00 | 3.39 ± 0.00 | 4.05 ± 0.00 |
| | FBK | 29.63 ± 0.00 | 4.17 ± 0.00 | 3.67 ± 0.00 | 3.51 ± 0.00 | 14.29 ± 0.00 | 3.44 ± 0.00 | 4.12 ± 0.00 | 2.36 ± 0.00 | 39.25 ± 0.00 | 22.90 ± 0.00 | 4.12 ± 0.00 | 3.54 ± 0.00 | 3.90 ± 0.00 |
| | NBK | 29.01 ± 2.83 | 4.15 ± 0.02 | 3.64 ± 0.02 | 3.38 ± 0.06 | 16.07 ± 1.79 | 3.75 ± 0.07 | 4.00 ± 0.03 | 2.28 ± 0.07 | 34.38 ± 0.37 | 19.08 ± 0.76 | 4.13 ± 0.01 | 3.47 ± 0.01 | 3.68 ± 0.01 |
| Claude Opus 4.1 | BK | 35.39 ± 0.36 | 4.18 ± 0.01 | 3.54 ± 0.00 | 3.55 ± 0.04 | 14.58 ± 2.25 | 3.79 ± 0.05 | 4.01 ± 0.03 | 2.25 ± 0.05 | 37.52 ± 1.14 | 22.39 ± 0.44 | 4.12 ± 0.03 | 3.38 ± 0.03 | 3.76 ± 0.01 |
| | SBK | 28.75 ± 0.00 | 4.20 ± 0.00 | 3.52 ± 0.00 | 3.76 ± 0.00 | 17.27 ± 0.00 | 3.95 ± 0.00 | 3.96 ± 0.00 | 2.43 ± 0.00 | 34.63 ± 0.00 | 19.38 ± 0.00 | 4.22 ± 0.00 | 3.33 ± 0.00 | 3.98 ± 0.00 |
| | SABK | 30.86 ± 0.00 | 4.22 ± 0.00 | 3.52 ± 0.00 | 3.79 ± 0.00 | 16.96 ± 0.00 | 3.89 ± 0.00 | 3.99 ± 0.00 | 2.45 ± 0.00 | 38.24 ± 0.00 | 22.90 ± 0.00 | 4.18 ± 0.00 | 3.31 ± 0.00 | 4.03 ± 0.00 |
| | FBK | 30.25 ± 0.00 | 4.16 ± 0.00 | 3.63 ± 0.00 | 3.42 ± 0.00 | 15.18 ± 0.00 | 3.59 ± 0.00 | 4.07 ± 0.00 | 2.21 ± 0.00 | 32.03 ± 0.00 | 17.56 ± 0.00 | 4.08 ± 0.00 | 3.54 ± 0.00 | 3.65 ± 0.00 |
| | NBK | 29.22 ± 2.85 | 4.38 ± 0.02 | 3.38 ± 0.01 | 4.30 ± 0.02 | 11.90 ± 1.03 | 4.18 ± 0.01 | 3.92 ± 0.02 | 4.02 ± 0.02 | 35.71 ± 1.56 | 22.39 ± 1.17 | 4.63 ± 0.02 | 3.03 ± 0.06 | 4.55 ± 0.04 |
| Gemini 3-Flash | BK | 35.39 ± 1.28 | 4.44 ± 0.01 | 3.35 ± 0.02 | 4.32 ± 0.01 | 9.52 ± 1.86 | 4.18 ± 0.02 | 3.89 ± 0.02 | 4.03 ± 0.01 | 37.11 ± 3.11 | 22.14 ± 2.02 | 4.68 ± 0.03 | 2.92 ± 0.05 | 4.60 ± 0.02 |
| | SBK | 31.06 ± 0.00 | 4.50 ± 0.00 | 3.24 ± 0.00 | 4.35 ± 0.00 | 12.61 ± 0.00 | 4.20 ± 0.00 | 3.89 ± 0.00 | 4.04 ± 0.00 | 33.56 ± 0.00 | 18.75 ± 0.00 | 4.72 ± 0.00 | 2.90 ± 0.00 | 4.65 ± 0.00 |
| | SABK | 37.65 ± 0.00 | 4.49 ± 0.00 | 3.28 ± 0.00 | 4.37 ± 0.00 | 9.82 ± 0.00 | 4.22 ± 0.00 | 3.89 ± 0.00 | 4.05 ± 0.00 | 40.93 ± 0.00 | 25.95 ± 0.00 | 4.77 ± 0.00 | 2.79 ± 0.00 | 4.69 ± 0.00 |
| | FBK | 33.33 ± 0.00 | 4.33 ± 0.00 | 3.43 ± 0.00 | 4.26 ± 0.00 | 14.29 ± 0.00 | 4.21 ± 0.00 | 3.89 ± 0.00 | 3.98 ± 0.00 | 36.75 ± 0.00 | 22.14 ± 0.00 | 4.59 ± 0.00 | 3.11 ± 0.00 | 4.50 ± 0.00 |
| | NBK | 28.81 ± 3.40 | 3.95 ± 0.00 | 3.12 ± 0.03 | 3.95 ± 0.02 | 12.50 ± 2.36 | 2.77 ± 0.05 | 4.02 ± 0.01 | 2.84 ± 0.06 | 33.80 ± 1.00 | 17.30 ± 1.17 | 3.82 ± 0.02 | 3.21 ± 0.04 | 3.88 ± 0.02 |
| OpenAI GPT-5.2 | BK | 30.04 ± 2.49 | 3.98 ± 0.02 | 3.01 ± 0.01 | 4.01 ± 0.02 | 13.10 ± 3.72 | 2.94 ± 0.07 | 3.99 ± 0.02 | 3.04 ± 0.05 | 38.52 ± 1.46 | 22.14 ± 1.32 | 3.92 ± 0.03 | 3.11 ± 0.02 | 3.94 ± 0.03 |
| | SBK | 26.54 ± 0.00 | 3.92 ± 0.00 | 2.96 ± 0.00 | 4.01 ± 0.00 | 13.39 ± 0.00 | 2.97 ± 0.00 | 3.99 ± 0.00 | 3.13 ± 0.00 | 29.33 ± 0.00 | 14.50 ± 0.00 | 3.90 ± 0.00 | 3.03 ± 0.00 | 4.00 ± 0.00 |
| | SABK | 27.16 ± 0.00 | 3.97 ± 0.00 | 2.92 ± 0.00 | 4.03 ± 0.00 | 16.96 ± 0.00 | 3.00 ± 0.00 | 3.94 ± 0.00 | 3.24 ± 0.00 | 36.06 ± 0.00 | 22.90 ± 0.00 | 3.92 ± 0.00 | 3.04 ± 0.00 | 3.99 ± 0.00 |
| | FBK | 25.93 ± 0.00 | 3.93 ± 0.00 | 3.11 ± 0.00 | 3.93 ± 0.00 | 14.29 ± 0.00 | 2.76 ± 0.00 | 4.03 ± 0.00 | 2.82 ± 0.00 | 32.08 ± 0.00 | 16.03 ± 0.00 | 3.86 ± 0.00 | 3.26 ± 0.00 | 3.86 ± 0.00 |
| **Human Baseline** | NBK | 26.54 | 3.33 | 3.22 | 3.01 | 8.93 | 2.29 | 3.99 | 2.39 | 36.09 | 22.14 | 3.13 | 3.31 | 3.05 |
| | BK | 27.16 | 3.46 | 3.14 | 3.01 | 9.82 | 2.36 | 3.96 | 2.39 | 36.86 | 22.90 | 3.28 | 3.30 | 3.05 |
| | NBK | 26.54 ± 1.23 | 4.56 ± 0.02 | 2.76 ± 0.06 | 4.69 ± 0.02 | 14.58 ± 1.86 | 3.95 ± 0.03 | 3.73 ± 0.02 | 4.01 ± 0.03 | 29.95 ± 1.12 | 16.03 ± 2.75 | 4.51 ± 0.04 | 3.10 ± 0.02 | 4.51 ± 0.05 |
| OpenAI O3-mini | BK | 30.45 ± 2.17 | 4.70 ± 0.01 | 2.62 ± 0.03 | 4.81 ± 0.00 | 13.10 ± 1.03 | 3.98 ± 0.04 | 3.67 ± 0.03 | 4.02 ± 0.03 | 31.66 ± 1.04 | 17.30 ± 2.89 | 4.61 ± 0.02 | 2.97 ± 0.05 | 4.62 ± 0.02 |
| | SBK | 18.24 ± 0.00 | 4.69 ± 0.00 | 2.61 ± 0.00 | 4.77 ± 0.00 | 11.32 ± 0.00 | 3.95 ± 0.00 | 3.64 ± 0.00 | 4.07 ± 0.00 | 28.35 ± 0.00 | 14.06 ± 0.00 | 4.61 ± 0.00 | 2.96 ± 0.00 | 4.60 ± 0.00 |
| | SABK | 27.78 ± 0.00 | 4.69 ± 0.00 | 2.53 ± 0.00 | 4.78 ± 0.00 | 11.61 ± 0.00 | 3.92 ± 0.00 | 3.57 ± 0.00 | 4.02 ± 0.00 | 33.18 ± 0.00 | 17.56 ± 0.00 | 4.62 ± 0.00 | 2.95 ± 0.00 | 4.67 ± 0.00 |
| | FBK | 25.31 ± 0.00 | 4.60 ± 0.00 | 2.75 ± 0.00 | 4.73 ± 0.00 | 10.71 ± 0.00 | 3.90 ± 0.00 | 3.72 ± 0.00 | 4.00 ± 0.00 | 32.70 ± 0.00 | 18.32 ± 0.00 | 4.44 ± 0.00 | 3.20 ± 0.00 | 4.43 ± 0.00 |
| | NBK | 23.66 ± 1.43 | 4.70 ± 0.04 | 3.50 ± 0.01 | 4.37 ± 0.04 | 13.39 ± 0.00 | 3.85 ± 0.13 | 4.06 ± 0.10 | 4.17 ± 0.10 | 33.14 ± 0.81 | 18.58 ± 1.92 | 4.74 ± 0.02 | 3.34 ± 0.11 | 4.49 ± 0.03 |
| DeepSeek v3 | BK | 27.98 ± 4.20 | 4.72 ± 0.01 | 3.29 ± 0.07 | 4.45 ± 0.04 | 12.50 ± 2.36 | 3.98 ± 0.04 | 3.97 ± 0.17 | 4.18 ± 0.08 | 36.92 ± 1.16 | 22.65 ± 2.68 | 4.84 ± 0.02 | 3.20 ± 0.11 | 4.43 ± 0.08 |
| | SBK | 25.00 ± 0.00 | 4.75 ± 0.00 | 3.41 ± 0.00 | 4.44 ± 0.00 | 8.04 ± 0.00 | 4.10 ± 0.00 | 4.14 ± 0.00 | 4.11 ± 0.00 | 34.21 ± 0.00 | 18.32 ± 0.00 | 4.76 ± 0.00 | 3.44 ± 0.00 | 4.51 ± 0.00 |
| | SABK | 25.31 ± 0.00 | 4.73 ± 0.00 | 3.36 ± 0.00 | 4.46 ± 0.00 | 10.71 ± 0.00 | 3.96 ± 0.00 | 4.05 ± 0.00 | 4.13 ± 0.00 | 36.37 ± 0.00 | 21.37 ± 0.00 | 4.77 ± 0.00 | 3.32 ± 0.00 | 4.52 ± 0.00 |
| | FBK | 29.01 ± 0.00 | 4.65 ± 0.00 | 3.33 ± 0.00 | 4.48 ± 0.00 | 13.39 ± 0.00 | 3.76 ± 0.00 | 4.09 ± 0.00 | 4.25 ± 0.00 | 32.92 ± 0.00 | 16.79 ± 0.00 | 4.76 ± 0.00 | 3.49 ± 0.00 | 4.43 ± 0.00 |
| | NBK | 26.75 ± 0.71 | 3.92 ± 0.03 | 3.42 ± 0.02 | 3.83 ± 0.05 | 12.50 ± 0.89 | 2.73 ± 0.05 | 3.99 ± 0.02 | 2.67 ± 0.08 | 25.61 ± 1.94 | 12.47 ± 1.17 | 3.63 ± 0.08 | 3.74 ± 0.03 | 3.41 ± 0.01 |
| Llama 3.3 70B | BK | 28.81 ± 0.36 | 4.01 ± 0.04 | 3.33 ± 0.02 | 3.92 ± 0.10 | 14.29 ± 1.79 | 2.84 ± 0.04 | 3.98 ± 0.03 | 2.82 ± 0.07 | 26.58 ± 1.46 | 13.49 ± 1.17 | 3.72 ± 0.06 | 3.64 ± 0.03 | 3.60 ± 0.05 |
| | SBK | 24.68 ± 0.00 | 4.03 ± 0.00 | 3.35 ± 0.00 | 3.92 ± 0.00 | 10.19 ± 0.00 | 2.90 ± 0.00 | 3.99 ± 0.00 | 2.72 ± 0.00 | 24.00 ± 0.00 | 11.63 ± 0.00 | 3.64 ± 0.00 | 3.67 ± 0.00 | 3.48 ± 0.00 |
| | SABK | 27.78 ± 0.00 | 4.07 ± 0.00 | 3.31 ± 0.00 | 3.99 ± 0.00 | 9.82 ± 0.00 | 2.89 ± 0.00 | 3.99 ± 0.00 | 2.90 ± 0.00 | 26.27 ± 0.00 | 13.74 ± 0.00 | 3.80 ± 0.00 | 3.58 ± 0.00 | 3.69 ± 0.00 |
| | FBK | 27.16 ± 0.00 | 3.93 ± 0.00 | 3.51 ± 0.00 | 3.81 ± 0.00 | 13.39 ± 0.00 | 2.72 ± 0.00 | 3.99 ± 0.00 | 2.55 ± 0.00 | 25.05 ± 0.00 | 12.21 ± 0.00 | 3.59 ± 0.00 | 3.71 ± 0.00 | 3.39 ± 0.00 |
| | NBK | 21.40 ± 0.36 | 4.00 ± 0.01 | 2.97 ± 0.01 | 4.03 ± 0.02 | 11.31 ± 1.36 | 3.67 ± 0.03 | 3.29 ± 0.04 | 3.83 ± 0.02 | 35.69 ± 2.54 | 19.34 ± 3.09 | 3.99 ± 0.01 | 3.00 ± 0.01 | 4.03 ± 0.01 |
| OpenAI O3 | BK | 29.22 ± 2.57 | 4.00 ± 0.01 | 2.96 ± 0.02 | 4.06 ± 0.01 | 12.80 ± 2.87 | 3.74 ± 0.08 | 3.19 ± 0.01 | 3.91 ± 0.01 | 37.99 ± 2.50 | 21.37 ± 2.75 | 3.99 ± 0.01 | 3.00 ± 0.01 | 4.03 ± 0.01 |
| | SBK | 25.31 ± 0.00 | 4.02 ± 0.00 | 2.88 ± 0.00 | 4.12 ± 0.00 | 14.29 ± 0.00 | 3.80 ± 0.00 | 3.19 ± 0.00 | 3.94 ± 0.00 | 33.38 ± 0.00 | 17.56 ± 0.00 | 4.00 ± 0.00 | 2.95 ± 0.00 | 4.08 ± 0.00 |
| | SABK | 24.07 ± 0.00 | 4.01 ± 0.00 | 2.90 ± 0.00 | 4.09 ± 0.00 | 14.29 ± 0.00 | 3.83 ± 0.00 | 3.19 ± 0.00 | 3.96 ± 0.00 | 35.89 ± 0.00 | 20.61 ± 0.00 | 4.00 ± 0.00 | 2.95 ± 0.00 | 4.10 ± 0.00 |
| | FBK | 18.52 ± 0.00 | 3.99 ± 0.00 | 2.98 ± 0.00 | 4.00 ± 0.00 | 14.29 ± 0.00 | 3.65 ± 0.00 | 3.31 ± 0.00 | 3.79 ± 0.00 | 32.18 ± 0.00 | 16.79 ± 0.00 | 3.97 ± 0.00 | 3.04 ± 0.00 | 4.01 ± 0.00 |
| | NBK | 22.43 ± 1.98 | 4.04 ± 0.01 | 3.82 ± 0.03 | 3.88 ± 0.09 | 12.50 ± 4.46 | 3.48 ± 0.03 | 4.09 ± 0.04 | 3.24 ± 0.01 | 28.35 ± 0.43 | 14.25 ± 1.59 | 3.97 ± 0.03 | 3.78 ± 0.05 | 3.86 ± 0.04 |
| Qwen 3 32B | BK | 23.87 ± 2.34 | 4.10 ± 0.03 | 3.75 ± 0.02 | 3.99 ± 0.02 | 13.39 ± 3.09 | 3.64 ± 0.01 | 4.01 ± 0.03 | 3.45 ± 0.04 | 31.07 ± 2.49 | 18.07 ± 2.68 | 4.01 ± 0.01 | 3.75 ± 0.04 | 3.95 ± 0.01 |
| | SBK | 19.71 ± 0.00 | 4.06 ± 0.00 | 3.76 ± 0.00 | 3.94 ± 0.00 | 10.11 ± 0.00 | 3.66 ± 0.00 | 4.08 ± 0.00 | 3.51 ± 0.00 | 32.38 ± 0.00 | 17.65 ± 0.00 | 4.00 ± 0.00 | 3.73 ± 0.00 | 3.99 ± 0.00 |
| | SABK | 24.07 ± 0.00 | 4.12 ± 0.00 | 3.71 ± 0.00 | 4.03 ± 0.00 | 9.82 ± 0.00 | 3.60 ± 0.00 | 4.08 ± 0.00 | 3.31 ± 0.00 | 29.26 ± 0.00 | 14.50 ± 0.00 | 4.05 ± 0.00 | 3.66 ± 0.00 | 3.99 ± 0.00 |
| | FBK | 21.60 ± 0.00 | 4.01 ± 0.00 | 3.84 ± 0.00 | 3.79 ± 0.00 | 9.82 ± 0.00 | 3.46 ± 0.00 | 4.12 ± 0.00 | 3.22 ± 0.00 | 31.14 ± 0.00 | 17.56 ± 0.00 | 3.92 ± 0.00 | 3.75 ± 0.00 | 3.89 ± 0.00 |
| | NBK | 21.40 ± 0.71 | 4.79 ± 0.02 | 2.60 ± 0.05 | 4.73 ± 0.01 | 13.99 ± 1.03 | 4.43 ± 0.03 | 3.28 ± 0.02 | 4.25 ± 0.05 | 31.76 ± 1.36 | 14.25 ± 1.59 | 4.68 ± 0.02 | 2.59 ± 0.05 | 4.68 ± 0.02 |
| Gemini 2.5-pro | BK | 31.48 ± 1.07 | 4.79 ± 0.01 | 2.58 ± 0.05 | 4.71 ± 0.02 | 15.18 ± 0.00 | 4.38 ± 0.03 | 3.45 ± 0.04 | 4.07 ± 0.04 | 34.95 ± 2.70 | 18.83 ± 1.76 | 4.69 ± 0.03 | 2.50 ± 0.04 | 4.71 ± 0.04 |
| | SBK | 23.12 ± 0.00 | 4.86 ± 0.00 | 2.42 ± 0.00 | 4.83 ± 0.00 | 20.18 ± 0.00 | 4.41 ± 0.00 | 3.34 ± 0.00 | 4.16 ± 0.00 | 32.70 ± 0.00 | 17.60 ± 0.00 | 4.75 ± 0.00 | 2.38 ± 0.00 | 4.72 ± 0.00 |
| | SABK | 27.16 ± 0.00 | 4.83 ± 0.00 | 2.46 ± 0.00 | 4.82 ± 0.00 | 12.50 ± 0.00 | 4.45 ± 0.00 | 3.33 ± 0.00 | 4.26 ± 0.00 | 31.72 ± 0.00 | 14.50 ± 0.00 | 4.80 ± 0.00 | 2.33 ± 0.00 | 4.77 ± 0.00 |
| | FBK | 22.84 ± 0.00 | 4.80 ± 0.00 | 2.56 ± 0.00 | 4.76 ± 0.00 | 10.71 ± 0.00 | 4.38 ± 0.00 | 3.54 ± 0.00 | 4.06 ± 0.00 | 36.74 ± 0.00 | 19.08 ± 0.00 | 4.62 ± 0.00 | 2.60 ± 0.00 | 4.61 ± 0.00 |
| | NBK | 19.75 ± 1.07 | 4.88 ± 0.03 | 2.05 ± 0.04 | 4.83 ± 0.03 | 14.58 ± 1.03 | 4.36 ± 0.08 | 2.73 ± 0.09 | 4.09 ± 0.07 | 30.53 ± 0.25 | 14.50 ± 0.00 | 4.85 ± 0.03 | 2.03 ± 0.02 | 4.73 ± 0.04 |
| Qwen 3 235B | BK | 25.93 ± 2.23 | 4.92 ± 0.03 | 1.96 ± 0.03 | 4.90 ± 0.03 | 13.39 ± 0.89 | 4.44 ± 0.06 | 2.63 ± 0.03 | 4.28 ± 0.03 | 34.39 ± 1.30 | 19.34 ± 1.59 | 4.91 ± 0.02 | 1.98 ± 0.06 | 4.83 ± 0.04 |
| | SBK | 20.99 ± 0.00 | 4.95 ± 0.00 | 1.90 ± 0.00 | 4.94 ± 0.00 | 13.51 ± 0.00 | 4.49 ± 0.00 | 2.56 ± 0.00 | 4.32 ± 0.00 | 29.60 ± 0.00 | 16.03 ± 0.00 | 4.89 ± 0.00 | 1.94 ± 0.00 | 4.86 ± 0.00 |
| | SABK | 25.31 ± 0.00 | 4.97 ± 0.00 | 1.86 ± 0.00 | 4.94 ± 0.00 | 15.32 ± 0.00 | 4.52 ± 0.00 | 2.50 ± 0.00 | 4.40 ± 0.00 | 32.90 ± 0.00 | 16.79 ± 0.00 | 4.95 ± 0.00 | 1.95 ± 0.00 | 4.92 ± 0.00 |
| | FBK | 22.22 ± 0.00 | 4.86 ± 0.00 | 2.03 ± 0.00 | 4.82 ± 0.00 | 14.29 ± 0.00 | 4.15 ± 0.00 | 2.95 ± 0.00 | 3.84 ± 0.00 | 33.61 ± 0.00 | 17.56 ± 0.00 | 4.71 ± 0.00 | 2.13 ± 0.00 | 4.65 ± 0.00 |
| | NBK | 22.43 ± 5.25 | 4.02 ± 0.02 | 2.42 ± 0.04 | 3.95 ± 0.05 | 8.33 ± 1.03 | 3.33 ± 0.03 | 3.29 ± 0.04 | 2.75 ± 0.09 | 31.10 ± 2.28 | 15.27 ± 0.76 | 3.99 ± 0.01 | 2.63 ± 0.03 | 3.87 ± 0.06 |
| OpenAI O4-mini | BK | 28.81 ± 1.28 | 4.08 ± 0.02 | 2.37 ± 0.05 | 4.05 ± 0.01 | 10.12 ± 2.25 | 3.53 ± 0.08 | 3.18 ± 0.02 | 3.02 ± 0.06 | 33.11 ± 3.79 | 18.07 ± 4.34 | 4.02 ± 0.01 | 2.62 ± 0.06 | 3.99 ± 0.03 |
| | SBK | 16.05 ± 0.00 | 4.08 ± 0.00 | 2.40 ± 0.00 | 4.11 ± 0.00 | 8.93 ± 0.00 | 3.62 ± 0.00 | 3.23 ± 0.00 | 3.24 ± 0.00 | 32.37 ± 0.00 | 16.79 ± 0.00 | 4.03 ± 0.00 | 2.61 ± 0.00 | 4.09 ± 0.00 |
| | SABK | 26.54 ± 0.00 | 4.12 ± 0.00 | 2.40 ± 0.00 | 4.07 ± 0.00 | 8.04 ± 0.00 | 3.64 ± 0.00 | 3.17 ± 0.00 | 3.29 ± 0.00 | 34.87 ± 0.00 | 19.08 ± 0.00 | 4.05 ± 0.00 | 2.60 ± 0.00 | 4.05 ± 0.00 |
| | FBK | 24.07 ± 0.00 | 4.05 ± 0.00 | 2.41 ± 0.00 | 4.01 ± 0.00 | 9.82 ± 0.00 | 3.33 ± 0.00 | 3.39 ± 0.00 | 2.86 ± 0.00 | 35.84 ± 0.00 | 19.08 ± 0.00 | 3.98 ± 0.00 | 2.64 ± 0.00 | 3.90 ± 0.00 |
| | NBK | 21.81 ± 2.92 | 4.30 ± 0.00 | 3.41 ± 0.03 | 3.94 ± 0.04 | 8.63 ± 4.22 | 4.08 ± 0.02 | 3.53 ± 0.09 | 3.95 ± 0.04 | 21.58 ± 0.86 | 10.94 ± 1.76 | 4.28 ± 0.04 | 3.52 ± 0.10 | 3.97 ± 0.09 |
| Llama 3.1 8B | BK | 25.31 ± 0.62 | 4.29 ± 0.02 | 3.44 ± 0.07 | 3.96 ± 0.04 | 6.85 ± 2.06 | 4.07 ± 0.06 | 3.44 ± 0.13 | 3.95 ± 0.12 | 22.81 ± 1.80 | 11.70 ± 0.44 | 4.32 ± 0.08 | 3.42 ± 0.02 | 4.02 ± 0.02 |
| | SBK | 22.22 ± 0.00 | 4.30 ± 0.00 | 3.36 ± 0.00 | 4.07 ± 0.00 | 7.14 ± 0.00 | 4.08 ± 0.00 | 3.53 ± 0.00 | 3.97 ± 0.00 | 21.98 ± 0.00 | 12.98 ± 0.00 | 4.27 ± 0.00 | 3.44 ± 0.00 | 3.94 ± 0.00 |
| | SABK | 20.37 ± 0.00 | 4.34 ± 0.00 | 3.40 ± 0.00 | 4.03 ± 0.00 | 13.39 ± 0.00 | 4.15 ± 0.00 | 3.42 ± 0.00 | 3.99 ± 0.00 | 20.79 ± 0.00 | 9.16 ± 0.00 | 4.38 ± 0.00 | 3.38 ± 0.00 | 3.98 ± 0.00 |
| | FBK | 24.69 ± 0.00 | 4.31 ± 0.00 | 3.51 ± 0.00 | 3.91 ± 0.00 | 10.71 ± 0.00 | 4.03 ± 0.00 | 3.58 ± 0.00 | 3.78 ± 0.00 | 22.77 ± 0.00 | 9.92 ± 0.00 | 4.27 ± 0.00 | 3.44 ± 0.00 | 4.06 ± 0.00 |

*Table 5.* **Performance across LLM families by confidence, difficulty, and feasibility levels.** Different versions of Gemini, OpenAI, Claude, Llama, Qwen, and DeepSeek evaluated across different levels of confidence, difficulty, and feasibility scores.

*Table 6.* **Performance across LLM families by human-rated confidence, difficulty, and feasibility levels.** Different versions of Gemini, OpenAI, Claude, Llama, Qwen, and DeepSeek evaluated across different levels of human-rated confidence, difficulty, and feasibility scores.

*Table 7.* Different versions of Gemini, OpenAI, Claude Sonnet, Llama, Qwen, and Deepseek evaluated on their ability to answer questions based on required background knowledge needed to answer questions.

| Model | # Corr. | # Ques. | Acc (%) |
|---|---|---|---|
| Gemini 3-pro | 1268 | 1350 | 93.93 |
| Claude Opus 4.5 | 1277 | 1344 | 95.01 |
| Claude Sonnet 4.5 | 1232 | 1316 | 93.62 |
| Claude Opus 4.1 | 1228 | 1327 | 92.54 |
| Gemini 3-Flash | 1279 | 1350 | 94.74 |
| OpenAI GPT-5.2 | 1276 | 1350 | 94.52 |
| OpenAI O3-mini | 1250 | 1350 | 92.59 |
| DeepSeek v3 | 1234 | 1353 | 91.20 |
| Llama 3.3 70B | 1132 | 1350 | 83.85 |
| OpenAI O3 | 1261 | 1350 | 93.41 |
| Qwen 3 32B | 1149 | 1342 | 85.62 |
| Gemini 2.5-pro | 1246 | 1350 | 92.30 |
| Qwen 3 235B | 1222 | 1350 | 90.52 |
| OpenAI O4-mini | 1252 | 1350 | 92.74 |
| Llama 3.1 8B | 955 | 1329 | 71.86 |

*Table 8.* Definitions of prediction error categories and flags.

| Main Category | Specific Error | Description |
|---|---|---|
| Comprehension & Scope | Task Misinterpretation | The answer addresses a fundamentally different scientific question than the one that was asked. |
| | Constraint Violation | The answer ignores or violates a specific instruction or constraint mentioned in the question. |
| | Insufficient Specificity | The answer is overly generic, too high-level, or omits necessary details required to fully address the question. |
| | Irrelevant Information | The answer includes factually correct information that is not essential and does not help in answering the specific question. |
| Factual & Extraction | Information Omission | The answer fails to use a required piece of data that is explicitly present in the provided materials (e.g., experimental setup, measurements). |
| | Factual Contradiction | The answer misrepresents or directly contradicts facts, values, or relationships that are clearly stated in the provided materials. |
| | Information Fabrication | The answer invents data, formulas, or external "facts" that are not supported by the provided materials. |
| | Detail Omission | The answer's own reasoning is incomplete because it omits a critical piece of evidence from the provided materials needed to support its conclusion. |
| Logical & Reasoning | Tautological Reasoning | The answer's justification is circular, merely restating the conclusion in different words without providing independent evidence. |
| | Unsupported Assumption | The reasoning relies on a significant, unstated assumption that is not supported by the provided materials. |
| | Disconnected Reasoning | The answer lists correct facts but fails to logically connect them to form a coherent argument for the final conclusion. |
| | Oversimplified Causality | The reasoning focuses on a minor cause while ignoring a more critical or explicitly stated factor that impacts the conclusion. |
| Scientific Rigor | False Certainty | The answer presents a probabilistic, uncertain, or correlational outcome as a definitive fact, using absolute language where nuance is required. |
| | Foundational Principles | The reasoning violates a fundamental, universally accepted scientific principle to reach its conclusion. |
| | Limitations Non-Ack. | The answer presents a conclusion without acknowledging critical limitations, uncertainties, or sources of error evident from the experimental setup. |
| Formatting/Mechanical | Incorrect Answer Reference | Specific to multiple-choice questions, the reasoning correctly identifies one answer, but the final choice points to a different option letter. |

# D. Prompts

> **Prompt used for errors analysis judge**
>
> [SYS]
> Fields: domain, field
>
> Instructions: You are acting as a judge evaluating a 'suggested_answer' to a scientific 'question' (of type 'question_type') which corresponds to the prediction of the outcome of a scientific experiment in {domain} and the field of {field}. Your goal is to identify the reason(s) why the provided answer is flawed or incorrect when compared to the 'ground_truth_answer' and the provided 'experimental_setup', 'measurements_taken', and 'background_knowledge'. Carefully review the provided materials and provide your judgment based on the rigorous definitions below. Your judgment should be based on a detailed analysis of the 'suggested_answer''s reasoning and factual claims.
>
> Evaluation Materials and Terminology:
> - 'question': The scientific question posed to the responder for prediction of the experimental outcome.
> - 'experimental_setup': Details of the experimental design, conditions, and procedures relevant to the 'question' provided to the responder for prediction of the experimental outcome.
> - 'measurements_taken': Information about the measurements taken relevant to the 'question' provided to the responder for prediction of the experimental outcome.
> - 'background_knowledge' (if any): Additional scientific context or principles relevant to the 'question' provided to the responder for prediction of the experimental outcome.
> - 'suggested_answer': The responder's answer to the 'question', including any reasoning or justification provided.
> - 'ground_truth_answer': The ground truth answer to the 'question', representing the correct prediction of the experimental outcome based on the provided materials.
>
> Question Types:
> - Multiple-Choice (MCQ): Includes a set of possible answers from which one (1) OR more (>1) must be selected.
> - Free-Form: Requires a comprehensive but concise explanation of the expected experimental results.
> - Numerical: Requires a specific numerical value prediction based on the provided data for the outcome of the experiment described in the question.
>
> Error Analysis Categories:
> 1. Comprehension & Scope Errors: The answer fails because it fundamentally misunderstands the user's question or violates its core constraints. This is the primary error if the answer, regardless of its correctness, is for the wrong question.
> 2. Factual & Extraction Errors: The answer fails because it incorrectly handles explicit information from the provided 'experimental_setup', 'measurements_taken', or 'background_knowledge'. It omits, fabricates, or directly contradicts facts that are clearly stated.
> 3. Logical & Reasoning Flaws: The answer fails because the argument is logically unsound, even if the individual facts cited are correct. The connections between evidence and conclusion are invalid.
> 4. Deficiencies in Scientific Rigor: The answer fails because it lacks the necessary nuance and rigor expected in scientific communication. It may be factually correct but is presented with false certainty or violates a core scientific principle.
> 5. Formatting & Mechanical Bug: The answer fails due to a non-substantive formatting error.
>
> Detailed Analysis Flags:
> First, choose a PRIMARY ERROR CATEGORY from the five main categories above that best explains WHY the 'suggested_answer' is flawed or incorrect. For this choice of the primary error category, provide a comprehensive justification (4-5 sentences) explaining your judgment.
>
> Second, for EACH flag below (INCLUDING from ALL categories, NOT just the one you selected), choose

YES, NO, or N/A based on the strict definitions provided:

1. Comprehension & Scope Errors
  - 'flag_task_misinterpretation':
    - Evidence Source: 'question', 'suggested_answer'.
    - Definition: Whether the 'suggested_answer' addresses a fundamentally different question than the one posed.
    - Prerequisite: None.
    - 'YES': The answer's core purpose is different from the question's intent or it addresses a different scientific question than was asked.
    - 'NO': The conditions for 'YES' are NOT satisfied.

  - 'flag_constraint_violation':
    - Evidence Source: 'question', 'suggested_answer'.
    - Definition: Whether the 'suggested_answer' ignores a specific instruction or constraint mentioned in the 'question'.
    - Prerequisite: The 'question' contains an explicit constraint.
    - 'YES': The answer violates an explicit constraint in the question.
    - 'NO': The prerequisite IS met, but the conditions for 'YES' are NOT satisfied.
    - 'N/A': The prerequisite is NOT met.

  - 'flag_insufficient_specificity':
    - Evidence Source: 'question', 'suggested_answer', 'ground_truth_answer'.
    - Definition: Whether the 'suggested_answer' is overly generic or lacks the required detail.
    - Prerequisite: None.
    - 'YES': The answer is too high-level and omits details that are necessary to fully address the question, as evidenced by the 'ground_truth_answer'.
    - 'NO': The conditions for 'YES' are NOT satisfied.

  - 'flag_irrelevant_information':
    - Evidence Source: 'question', 'suggested_answer', 'ground_truth_answer'.
    - Definition: Whether the 'suggested_answer' includes factually correct but non-essential information.
    - Prerequisite: None.
    - 'YES': The answer contains information that does not help answer the specific 'question'.
    - 'NO': The conditions for 'YES' are NOT satisfied.

2. Factual & Extraction Errors
  - 'flag_information_omission':
    - Evidence Source: 'experimental_setup', 'measurements_taken', 'background_knowledge', 'suggested_answer'.
    - Definition: Whether the 'suggested_answer' fails to extract or reports as "missing" a REQUIRED piece of data explicitly present in the provided materials.
    - Prerequisite: The information is explicitly stated in the 'experimental_setup', 'measurements_taken', or 'background_knowledge' AND the information is REQUIRED for answering the question.
    - 'YES': A key fact, value, or condition from the provided materials is missing from, or was ignored in the 'suggested_answer'.
    - 'NO': The prerequisite IS met, but the conditions for 'YES' are NOT satisfied.
    - 'N/A': The prerequisite is NOT met.

  - 'flag_factual_contradiction':
    - Evidence Source: 'experimental_setup', 'measurements_taken', 'background_knowledge', 'suggested_answer'.
    - Definition: Whether the 'suggested_answer' directly misrepresents or contradicts facts, values, or relationships stated in the provided materials.
    - Prerequisite: None.
    - 'YES': A statement in the 'suggested_answer' is verifiably FALSE when checked against the provided materials.
    - 'NO': The conditions for 'YES' are NOT satisfied.

  - 'flag_information_fabrication':
    - Evidence Source: 'experimental_setup', 'measurements_taken', 'background_knowledge', 'suggested_answer'.

- Definition: Whether the 'suggested_answer' invents data, formulas, or external "facts" not supported by the provided materials.
        - Prerequisite: None.
        - 'YES': The answer includes specific information that cannot be found in or reasonably inferred from the provided materials.
        - 'NO': The conditions for 'YES' are NOT satisfied.

- 'flag_detail_omission_in_reasoning':
        - Evidence Source: 'experimental_setup', 'measurements_taken', 'background_knowledge', 'suggested_answer'.
        - Definition: Whether the reasoning in the 'suggested_answer' omits a CRITICAl piece of evidence from the provided materials that is necessary to logically support its OWN conclusion.
        - Prerequisite: The 'suggested_answer' presents a logical argument or reasoning.
        - 'YES': The argument or reasoning provided for the answer is incomplete because a necessary premise from the provided materials is missing.
        - 'NO': The prerequisite IS met, but the conditions for 'YES' are NOT satisfied.
        - 'N/A': The prerequisite is NOT met.

3. Logical & Reasoning Flaws
    - 'flag_tautological_reasoning':
        - Evidence Source: 'suggested_answer'.
        - Definition: Whether the justification restates the conclusion without providing independent evidence.
        - Prerequisite: The 'suggested_answer' provides a justification or reasoning.
        - 'YES': The reasoning is circular, using the conclusion as its own evidence.
        - 'NO': The prerequisite IS met, but the conditions for 'YES' are NOT satisfied.
        - 'N/A': The prerequisite is NOT met.

    - 'flag_unsupported_assumption':
        - Evidence Source: 'experimental_setup', 'measurements_taken', 'background_knowledge', 'suggested_answer'.
        - Definition: Whether the reasoning relies on a significant, unstated assumption that is NOT supported by the provided materials.
        - Prerequisite: The 'suggested_answer' presents a logical argument or reasoning.
        - 'YES': The logical leap from evidence to conclusion requires an assumption that is NOT provided or justified by the provided materials.
        - 'NO': The prerequisite IS met, but the conditions for 'YES' are NOT satisfied.
        - 'N/A': The prerequisite is NOT met.

    - 'flag_disconnected_reasoning':
        - Evidence Source: 'suggested_answer'.
        - Definition: Whether the 'suggested_answer' lists correct facts but fails to logically connect them to the final conclusion.
        - Prerequisite: The 'suggested_answer' presents more than one (>1) piece of evidence in its reasoning.
        - 'YES': NO logical connection is made between the evidence presented and the conclusion drawn.
        - 'NO': The prerequisite IS met, but the conditions for 'YES' are NOT satisfied.
        - 'N/A': The prerequisite is NOT met.

    - 'flag_oversimplified_causality':
        - Evidence Source: 'experimental_setup', 'measurements_taken', 'background_knowledge', 'suggested_answer'.
        - Definition: Whether the reasoning focuses on a minor cause while ignoring a more critical or explicitly stated factor impacting the conclusion to be made from the provided materials.
        - Prerequisite: The provided materials present multiple potential causal factors.
        - 'YES': The reasoning incorrectly prioritizes a secondary factor over the primary factor described in the provided materials.
        - 'NO': The prerequisite IS met, but the conditions for 'YES' are NOT satisfied.
        - 'N/A': The prerequisite is NOT met.

4. Deficiencies in Scientific Rigor
    - 'flag_false_certainty':
        - Evidence Source: 'experimental_setup', 'measurements_taken', 'background_knowledge', 'suggested_

answer'.
- Definition: Whether the 'suggested_answer' presents a probabilistic, correlational, or uncertain outcome as a definitive fact.
- Prerequisite: The outcome described in the provided materials or 'ground_truth_answer' is NON-deterministic.
- 'YES': The answer uses absolute language where uncertainty or probability is warranted.
- 'NO': The prerequisite IS met, but the conditions for 'YES' are NOT satisfied.
- 'N/A': The prerequisite is NOT met.

- 'flag_violation_of_foundational_principles':
- Evidence Source: 'suggested_answer'.
- Definition: Whether the reasoning in the 'suggested_answer' is scientifically invalid because it violates a fundamental, universally accepted scientific principle.
- Prerequisite: The 'suggested_answer' invokes reasoning related to a known scientific principle.
- 'YES': The reasoning makes a statement that is verifiably FALSE according to a FOUNDATIONAL scientific principle.
- 'NO': The prerequisite IS met, but the conditions for 'YES' are NOT satisfied.
- 'N/A': The prerequisite is NOT met.

- 'flag_failure_to_acknowledge_limitations':
- Evidence Source: 'experimental_setup', 'suggested_answer'.
- Definition: Whether the 'suggested_answer' presents a conclusion without acknowledging critical limitations or uncertainties evident from the 'experimental_setup'.
- Prerequisite: The 'experimental_setup' contains CLEAR limitations or sources of error.
- 'YES': The answer presents its conclusion as robust WITHOUT mentioning the known limitations.
- 'NO': The prerequisite IS met, but the conditions for 'YES' are NOT satisfied.
- 'N/A': The prerequisite is NOT met.

5. Formatting & Mechanical Bugs
- 'flag_incorrect_answer_reference':
- Evidence Source: 'question', 'suggested_answer', 'ground_truth_answer'.
- Definition: Whether the provided justification or reasoning identifies the correct answer option(s), BUT then a different option letter is given as the final answer.
- Prerequisite: The 'question' IS a multiple-choice question (MCQ).
- 'YES': The justification or reasoning provided refers to one option letter while discussing the content of another.
- 'NO': The prerequisite IS met, but the conditions for 'YES' are NOT satisfied.
- 'N/A': The prerequisite is NOT met.

Output: Provide your evaluation in the specified JSON format, including the single 'primary_error_category' and the choice ('YES', 'NO', or 'N/A') for every 'flag_'. Note that for some flags the ONLY possible choices as 'YES' and 'NO' (NOT 'N/A'). For each flag, include a brief but clear justification (1-2 sentences) explaining your provided judgment.

[USER]
Fields: outcome_prediction_question, pq_format, experimental_setup, measurement_taken, required_background_knowledge, answer, reasoning_for_answer, clean_gta

Given the following 'experimental_setup' and 'measurements_taken' and 'background_knowledge' (if any):
- 'experimental_setup':
    """
    {experimental_setup}
    """

- 'measurements_taken':
    """
    {measurement_taken}
    """

- 'background_knowledge':

```
    """
    {required_background_knowledge}
    """

And for the following 'question' (of type 'question_type') and its 'ground_truth_answer':
- 'question_type': {pq_format}

- 'question' (along with choices if applicable):
    """
    {outcome_prediction_question}
    """

- 'ground_truth_answer':
    """
    {clean_gta}
    """

Evaluate the following 'suggested_answer' with respect to the provided materials as instructed:
- 'suggested_answer':
  """
  {answer}

  REASONING: {reasoning_for_answer}
  """
```

## Prompt for generating responses with background knowledge

```
[SYS]
Fields: domain, field, experimental_setup, measurement_taken, required_background_knowledge

Instructions: You are tasked with predicting the outcome of a scientific experiment in {domain} and the field
    of {field} given the provided 'experimental_setup' and 'measurements_taken'. You must analyze the
    user's scientific 'question' very carefully, and forecast the results AS ACCURATELY AS POSSIBLE
    given the inputs provided. Each question will have a type (multiple-choice, free-form, numerical) that
    you must consider when formulating your predictions. Ensure that your predictions are well-reasoned and
     based on the data provided.

Inputs :
    - 'domain': {domain}
    - 'field': {field}
    - 'experimental_setup': {experimental_setup}
    - 'measurements_taken': {measurement_taken}
    - 'required_background_knowledge': {required_background_knowledge}

Question Types:
    - Multiple-Choice: Choose the most likely outcome from the list of provided options.
    - Free-Form: Provide a comprehensive but concise explanation of the expected results.
    - Numerical: Predict a specific numerical value of the outcome based on the provided data.

Output: Depending on the 'question_type' provided by the user and based on the provided background
    knowledge, output the appropriate prediction in the following output fields:
    - 'answer'
        - Multiple-Choice: Write ONLY the letter(s) corresponding to the most likely outcome in the 'answer'
              field (e.g., "X"). If choosing multiple letters (items) is allowed by the 'question' and desired,
              separate them with commas (e.g., "X, Y, Z").
        - Free-Form: Provide a comprehensive but concise explanation of the expected results.
        - Numerical: Write ONLY the predicted numerical value in the 'answer' field (e.g., "1.234").
```

- 'reasoning_for_answer': A detailed explanation of how you arrived at your prediction, including any relevant calculations, assumptions, or scientific principles applied.
- 'confidence': Choose between the levels provided. "Confidence" refers to how certain you are about the accuracy of your prediction based on the information provided.
- 'difficulty': Choose between the levels provided. "Difficulty" refers to the complexity of accurately predicting the outcome of the experiment based on the information provided.
- 'feasibility': Choose between the levels provided. "Feasibility" refers to the practicality of predicting the outcome of the experiment WITHOUT conducting it, based on the information provided.
- 'reasoning_for_feasibility': A detailed explanation of how you arrived at your feasibility assessment, considering factors such as experimental design, measurement accuracy, and potential sources of error.

Ensure that your predictions are clear, concise, and directly address the user's scientific 'question'.

[USER]
Fields: pq_format, outcome_prediction_question

Answer the following 'question' as accurately as possible:
- 'question_type': {pq_format}
- 'question': {outcome_prediction_question}

---

**Prompt for generating responses without background knowledge**

[SYS]
Fields: domain, field, experimental_setup, measurement_taken

Instructions: You are tasked with predicting the outcome of a scientific experiment in {domain} and the field of {field} given the provided 'experimental_setup' and 'measurements_taken'. You must analyze the user's scientific 'question' very carefully, and forecast the results AS ACCURATELY AS POSSIBLE given the inputs provided. Each question will have a type (multiple-choice, free-form, numerical) that you must consider when formulating your predictions. Ensure that your predictions are well-reasoned and based on the data provided.

Inputs :
- 'domain': {domain}
- 'field': {field}
- 'experimental_setup': {experimental_setup}
- 'measurements_taken': {measurement_taken}

Question Types:
- Multiple-Choice: Choose the most likely outcome from the list of provided options.
- Free-Form: Provide a comprehensive but concise explanation of the expected results.
- Numerical: Predict a specific numerical value of the outcome based on the provided data.

Output: Depending on the 'question_type' provided by the user, output the appropriate prediction in the following output fields:
- 'answer'
    - Multiple-Choice: Write ONLY the letter(s) corresponding to the most likely outcome in the 'answer' field (e.g., "X"). If choosing multiple letters (items) is allowed by the 'question' and desired, separate them with commas (e.g., "X, Y, Z").
    - Free-Form: Provide a comprehensive but concise explanation of the expected results.
    - Numerical: Write ONLY the predicted numerical value in the 'answer' field (e.g., "1.234").
- 'reasoning_for_answer': A detailed explanation of how you arrived at your prediction, including any relevant calculations, assumptions, or scientific principles applied.
- 'confidence': Choose between the levels provided. "Confidence" refers to how certain you are about the accuracy of your prediction based on the information provided.
- 'difficulty': Choose between the levels provided. "Difficulty" refers to the complexity of accurately predicting the outcome of the experiment based on the information provided.

- 'feasibility': Choose between the levels provided. "Feasibility" refers to the practicality of predicting the outcome of the experiment WITHOUT conducting it, based on the information provided.
- 'reasoning_for_feasibility': A detailed explanation of how you arrived at your feasibility assessment, considering factors such as experimental design, measurement accuracy, and potential sources of error.

Ensure that your predictions are clear, concise, and directly address the user's scientific 'question'.

[USER]
Fields: pq_format, outcome_prediction_question

Answer the following 'question' as accurately as possible:
- 'question_type': {pq_format}
- 'question': {outcome_prediction_question}

---

**Prompt used for judge**

[SYS]
Fields: domain, field, rubric_criteria_lines

Instructions: You are acting as an impartial judge evaluating a suggested answer ('suggested_answer') to a scientific prediction question in the {domain} domain and the field of {field}. Your goal is to determine how well the 'suggested_answer' aligns with the 'ground_truth_answer' based on a set of specific 'rubric_criteria' (a list of >=1 criterion items). Each criterion will need to be evaluated independently. Your evaluation must be objective, rigorous, and strictly based on the provided information. The 'question' was asked given the context information of a scientific experiment as defined by the provided 'experimental_setup' and 'measurements_taken'.

Evaluation Requirements:
1. First, carefully read and understand the scientific context (domain, field) and the specific 'question'. Use the provided 'experimental_setup' and 'measurements_taken' to inform your understanding.
2. Compare the 'suggested_answer' with the 'ground_truth_answer' and reason about the overall correctness and completeness of the 'suggested_answer'.
3. For EACH criterion (INDEPENDENTLY) provided in the 'rubric_criteria' list (could be 1 or more criterion items), you must meticulously assess if the 'suggested_answer' satisfies it ("true" or "false"). The ground truth answer should be used as the reference as the overall correct answer to the 'question'. Provide the output in the corresponding '_satisfied' fields.
4. Your judgment must be objective. Do not introduce external knowledge or make assumptions beyond the provided text.
5. Provide a concise yet clear justification for EACH criterion's determined satisfaction status ("true"/"false") in the corresponding '_reasoning' field.

Inputs:
- 'domain': {domain}
- 'field': {field}
- 'rubric_criteria': Provided below as a list.

Evaluation Criteria:
{rubric_criteria_lines}

Output Format:
You MUST provide your evaluation in a strict JSON format. For each criterion, you will output two fields: one boolean ('_satisfied') and one string ('_reasoning').

[USER]
Fields: outcome_prediction_question, predicted_answer, clean_gta, experimental_setup, measurement_taken

Given the following 'experimental_setup' and 'measurements_taken':
- 'experimental_setup':

```
                """
            {experimental_setup}
                """

- 'measurements_taken':
                """
            {measurement_taken}
                """

Evaluate the following 'question' with respect to the provided 'suggested_answer' and 'ground_truth_answer'
        as instructed:
    - 'question': {outcome_prediction_question}
    - 'suggested_answer': {predicted_answer}
    - 'ground_truth_answer': {clean_gta}
```

---

**Prompt to generate responses for questions on background knowledge**

[SYS]
Fields: domain, field

Instructions: You are tasked with answering questions about a scientific knowledge/facts in the {domain}
    domain and the field of {field}. You will be provided with the experimental setup ('experimental_setup')
    and the measurements taken ('measurement_taken') as additional context that are relevant to the
    questions. Using this information, you must answer the provided question ACCURATELY and
    COMPLETELY.

Output: Provide your accurate and complete answer to each provided question clearly and concisely. Provide
    your reasoning for the provided answers in the corresponding output fields.

[USER]
Fields: bkg_to_qa, experimental_setup, measurement_taken

Given the following 'experimental_setup' and 'measurements_taken':
- 'experimental_setup':
        """
    {experimental_setup}
        """

- 'measurements_taken':
        """
    {measurement_taken}
        """

Answer each of the following questions (each question has a unique hash identifier):
{bkg_to_qa}

---

**Prompt to generate questions on background knowledge**

[SYS]
Fields: domain, field

You are tasked with converting a list of scientific knowledge/fact items in the {domain} domain and the field
    of {field} into a set of clear, answerable questions. You will be provided with the description of the
    experimental setup and the measurements taken, the purpose of the given scinetific knowledge/fact items
    is to help predict the outcome of the experiment. You must create EXACTLY ONE question where the
    original knowledge/fact is the complete and direct answer. DO NOT MAKE any direct references to the

experimental setup, and the measurements taken in the questions.

Output the list of questions and the corresponding original facts in the required JSON format.

[USER]
Fields: experimental_setup, measurement_taken, required_background_knowledge_hashed

Given the following 'experimental_setup' and 'measurements_taken':
- 'experimental_setup':
    """
    {experimental_setup}
    """

- 'measurements_taken':
    """
    {measurement_taken}
    """

List of knowledge/fact items to convert:
{required_background_knowledge_hashed}

## Prompt for generating synthetic background knowledge

[SYS]
Fields: domain, field

Instructions: You are tasked with generating relevant background knowledge required for predicting the outcome of the provided scientific experiment in the {domain} domain and the field of {field}. Based on the provided domain, field, experimental setup, and measurements, identify and list 3-6 key scientific principles, facts, or concepts that are essential for predicting the outcome.

Output: Your output must match the required JSON format. Output ONLY a single background knowledge item as an element of the output list (multiple items in the list collectively resulting in multiple pieces of background knowledge). Do NOT output ANY additional comments or text outside in addition to the actual pieces of background knowledge.

Example Output:
{
  "generate_bkg": [
    "Background sentence 1.",
    "Background sentence 2."
  ]
}

[USER]
Fields: domain, field, experimental_setup, measurement_taken

Please generate the background knowledge for the following experimental direction:
- Domain: {domain}
- Field: {field}
- Experimental Setup: {experimental_setup}
- Measurements Taken: {measurement_taken}

## Prompt for judging answers to questions on background knowledge

[SYS]
Fields: domain, field

Instructions: You are acting as an impartial judge evaluating a list of answers ('answers') to questions and if those answers capture the corresponding ground truth facts ('ground_truth_facts') for that question in the context of a scientific experiment in the {domain} and the field of {field}. You will also be provided with the experimental setup ('experimental_setup') and measurements taken ('measurements_taken') as additional context that are relevant to the questions. Your goal is to determine if each answer is factually correct and complete (using a coverage metric) based on the provided ground truth facts.

Output: Output your evaluation in the provided JSON format. Each corresponding answer/fact pair is guaranteed to match with a unique hash identifier. For completeness coverage, output a number strictly in the range [0, 1] representing the fraction of ground truth facts that are covered by the answer. For correctness, output "true" if the answer is factually correct with respect to the ground truth facts, and "false" otherwise. Provide a concise yet clear justification for each judgment in the corresponding 'reasoning' fields.

[USER]
Fields: answer_bkg_qa, experimental_setup, measurement_taken, required_background_knowledge_hashed

Given the following 'experimental_setup' and 'measurements_taken':
- 'experimental_setup':
    """

    {experimental_setup}
    """

- 'measurements_taken':
    """

    {measurement_taken}
    """

And the following 'ground_truth_facts' (IDs provided in the start of the lines):
{required_background_knowledge_hashed}

Provide your judgments strictly matching the above criteria on the correctness and completeness coverage of each ANSWER against the ground truth (ANSWERs need to be evaluated NOT the ground truth facts):
{answer_bkg_qa}

---

Prompt for converting MCQ to FF

[SYS]
Fields: domain, field

Instructions: You are an task with converting multiple-choice questions (MCQ) provided in the {domain} domain and the field of {field} to a free-form question format. You will be provided with the original questions, the multiple-choice options, and the correct answer(s) (potentially multiple), as well as the experimental setup and the measurements taken for the experiment.

Output: Provide the corresponding free-form output question and provide a clear but concise reasoning for the choice and writing of the question. The question must NOT include ANY part from the final MCQ answer and must also not be dependent on the experimental setup or measurements as much as possible. The goal is to have a responder answer the output free-form question, and for a judge to then be able to check whether the free-form question was answered correctly and completely or not based on the original correct answer(s) to the original MCQ question. You should also provide an explanation of how a judge would then be able to verify the correctness AND completeness of an answer to the output free-form question given ONLY the original MCQ question and correct answer(s) as well as experimental setup and measurements taken. Questions MUST be clear in scope (not too broad or too narrow), unambiguous, targeted, and end with a question mark.

[USER]
Fields: outcome_prediction_question, experimental_setup, measurement_taken, clean_gta

Given the following 'experimental_setup' and 'measurements_taken':
- 'experimental_setup':
    """

    {experimental_setup}
    """

- 'measurements_taken':
    """

    {measurement_taken}
    """

Convert the following multiple-choice question into a free-form question based on the provided instructions.
{outcome_prediction_question}

Correct answer(s) for this question (NOT to be included in the output free-form question):
{clean_gta}

Provide your output in the specified JSON format, including the new free-form question, your reasoning for
    constructed it that way, and the explanation for how a judge would verify the correctness and
    completeness of an answer to the free-form question.

