# OpenReview forum: "SciPredict: Can LLMs Predict the Outcomes of Scientific Experiments in Natural Sciences?"
_ICML.cc/2026/Conference — ICML 2026 regular_

### Official Review · Reviewer_3n88 · 2026-02-21

**Soundness:** 3
**Presentation:** 3
**Significance:** 3
**Originality:** 3
**Overall Recommendation:** 5
**Confidence:** 4

**Summary:**

This paper introduces **SciPredict**, a benchmark designed to evaluate whether large language models (LLMs) can predict the outcomes of real, recently published experiments in the natural sciences. The benchmark contains 405 expert-curated tasks spanning physics, chemistry, and biology across 33 subfields, with multiple answer formats (e.g., multiple-choice, free-form, and numerical). Each task provides an experimental setup and measurement context, together with expert-organized background knowledge and a verifiable ground-truth outcome. Using SciPredict, the authors conduct a broad evaluation of 15 state-of-the-art LLMs and include a human expert baseline. Beyond reporting predictive accuracy, the study also examines how model self-reports (e.g., confidence, perceived difficulty, and feasibility) relate to correctness, offering a dataset and experimental results that characterize current LLM strengths and limitations in anticipating empirical outcomes in the natural sciences.

**Compliance With Llm Reviewing Policy:**

Affirmed.

**Final Justification:**

This paper addresses an important and timely problem: whether LLMs can predict the outcomes of real scientific experiments in the natural sciences. I find the overall contribution significant because it goes beyond generic scientific QA and instead evaluates a practically meaningful capability that is relevant to AI-for-Science workflows. The benchmark is also reasonably original in its framing, task design, and accompanying analyses of calibration, feasibility, and answer-format sensitivity, even though the paper does not propose a new learning algorithm.

From a soundness perspective, I find the work technically solid overall. The benchmark is expert-curated, covers multiple scientific domains and answer formats, and includes both frontier LLMs and a human expert baseline. The empirical study is broad, and the paper reports several useful findings, including the difficulty gap between MCQ and free-form settings and the limited but generally positive effect of expert background knowledge. The presentation is also clear overall: the paper is well structured, easy to follow, and the main takeaways are communicated clearly.

My main concerns in the initial review were about potential data contamination, the statistical robustness of some key comparisons, and the validity of the free-form judging pipeline. The rebuttal addressed these concerns in a meaningful way. In particular, the authors provided additional human verification for the free-form evaluation pipeline, which substantially increases my confidence in that component of the study. They also clarified the task assumptions and feasibility analysis, and, importantly, they added bootstrap uncertainty analyses for several of the main empirical comparisons. These additions significantly strengthened the evidence supporting the paper’s central claims. The follow-up clarification on the familiarity filtering procedure was also helpful in showing that this step affected only a small number of candidate items and did not materially alter the benchmark composition.

Some limitations remain. In particular, the benchmark size still limits how strongly one should interpret very fine-grained subgroup conclusions, and the contamination-control evidence could still be stronger with more external auditing. However, after considering both the paper and the rebuttal, I view these as limitations on scope and future strengthening opportunities rather than flaws that undermine the core contribution.

Overall, the rebuttal improved my confidence in the paper and changed my evaluation positively. I therefore support acceptance. In my view, this is a technically solid and practically valuable benchmark/resource paper on an important problem, and it is likely to be useful to and built upon by the community.

**Key Questions For Authors:**

1. **Leakage/contamination risk and stronger evidence for “unseen” evaluation.**
   Beyond using the post-2025-03-31 “first release” criterion, what additional evidence supports the claim that the benchmark evaluates *unseen* experimental outcomes for the tested models (especially closed models with unclear training windows)? Did you perform any systematic leakage audits (e.g., retrieval-style similarity checks, metadata-based screening, or “source identification” probing)? Please report quantitative results and how flagged items were handled.
   *How it could change my evaluation:* Stronger evidence would increase my confidence in the headline conclusions; weak/no evidence would make the main takeaways less definitive.

2. **Statistical robustness of the main effects (format, background knowledge, and model–human gaps).**
   Several key conclusions rely on comparing conditions (MCQ vs free-form vs numerical; NBK vs BK variants; models vs human experts). Can you provide bootstrap confidence intervals and paired significance tests for these primary effects, and clarify which comparisons remain significant after controlling for multiple testing?
   *How it could change my evaluation:* If the main effects are stable under rigorous uncertainty analysis, I would place more weight on the fine-grained claims; if not, I would view them as suggestive.

3. **Validity of free-form scoring and judge calibration.**
   Free-form evaluation relies on rubric-based LLM judging. Can you provide a human verification study (even a modest random sample) reporting inter-annotator agreement and judge–human agreement, and a breakdown of disagreement patterns across domains/subfields?
   *How it could change my evaluation:* Strong judge validation would substantially strengthen the free-form results and error analysis; low agreement would weaken confidence in those conclusions.

4. **Interpreting low human expert accuracy and feasibility ratings.**
   Human expert accuracy is relatively low, which may indicate intrinsic difficulty and/or under-specification of some tasks. Can you (i) report the distribution of feasibility ratings, (ii) show accuracy conditioned on feasibility bins (for both humans and models), and (iii) clarify how feasibility was elicited and normalized across experts?
   *How it could change my evaluation:* If models remain poorly calibrated and inaccurate even on high-feasibility tasks, it strengthens the “fundamental limitation” claim; if performance improves markedly on high-feasibility items, the conclusions may need to be stated more cautiously and the benchmark’s practical interpretation changes.

5. **Task construction assumptions: what information is “allowed,” and are tasks solvable from provided context?**
   For many tasks, predicting the outcome may require implicit domain priors or external facts not fully contained in the prompt. Can you clarify the intended assumption: should the model predict using only the provided setup + background knowledge, or is it allowed to rely on general scientific priors? Additionally, can you quantify how many tasks are deemed “solvable” by experts given the provided information (e.g., via feasibility labels), and provide representative examples where predictions are/are not possible?
   *How it could change my evaluation:* Clearer task assumptions and solvability evidence would improve interpretability of the benchmark results; ambiguity here could reduce confidence in the benchmark’s validity for measuring scientific reasoning.

**Limitations:**

Yes.

**Strengths And Weaknesses:**

## Strengths

- **Addresses an important and timely problem.** Predicting outcomes of real scientific experiments is a concrete capability relevant to AI-for-Science workflows (e.g., prioritizing hypotheses/experiments, assisting literature-driven planning). Framing the evaluation around *empirical outcomes* rather than generic scientific QA makes the benchmark practically meaningful.

- **Well-motivated benchmark design with structured task components.** SciPredict provides experimental setup/measurement context together with expert-organized background knowledge and verifiable ground-truth outcomes. The inclusion of multiple answer formats (MCQ, free-form, numerical) supports broader evaluation of model behavior across realistic interaction modes.

- **Broad, systematic evaluation and insightful analyses.** The paper evaluates many contemporary LLMs and includes a human expert baseline. Beyond accuracy, it analyzes self-reported confidence/difficulty/feasibility vs. correctness, highlighting a “calibration gap” that is highly relevant to real deployment and safety.

- **Useful empirical findings for the community.** The study shows that answer format substantially affects performance (e.g., MCQ vs. free-form), and that providing expert background knowledge yields only modest gains. These observations are actionable for future benchmark design and for interpreting scientific-LLM claims.

- **Clear effort on data curation and quality control.** The authors describe multi-stage expert involvement, task filtering, and rubric-based evaluation, which increases confidence that the benchmark targets genuine experimental reasoning rather than trivial pattern matching.

## Weaknesses

- **Limited scale for fine-grained conclusions.** While the dataset is carefully curated, 405 tasks across 33 subfields can make some per-subfield or per-setting comparisons statistically fragile. The paper would benefit from stronger uncertainty reporting (e.g., bootstrap confidence intervals, paired significance tests) and stability analyses across subfields.

- **Potential residual concerns about training data contamination.** Using a recency cutoff is a reasonable step, but for closed models the exact training window is unclear, and scientific content can appear in multiple channels before “official” release. Additional leakage checks (e.g., retrieval-style matching or metadata-based audits) would strengthen the central claim of evaluating *unseen* experimental outcomes.

- **Free-form evaluation relies on an LLM judge.** Rubric-based judging is sensible, but LLM judges may introduce systematic biases. The paper would be stronger with more extensive human verification (even a modest sample) and inter-annotator agreement, especially across diverse scientific subfields.

- **Human baseline interpretation needs more nuance.** The relatively low human expert accuracy suggests tasks may be intrinsically hard or sometimes under-specified given the provided context. A clearer breakdown by “high-feasibility” vs. “low-feasibility” tasks (and whether model–human gaps differ) would help interpret what the benchmark measures.

- **Reproducibility details could be tighter.** For full reproducibility, it would help to specify all prompting templates, judge prompts, model versions, decoding parameters, and any post-processing steps in a centralized, easy-to-follow manner, and to ensure the released code/data covers the complete evaluation pipeline.

---

> ### Author Rebuttal · Authors · 2026-03-31
>
> We thank the reviewer 3n88 for the thorough and constructive evaluation. We are glad that the reviewer found the problem important and timely, appreciated the structured benchmark design and broad evaluation, and highlighted the value of our analyses on calibration, answer format sensitivity, and background knowledge. We also appreciate the recognition of the effort in expert curation and quality control.
>
> > Q1/W2: Potential residual concerns on data contamination
>
> For each paper included in the benchmark, we prompted every evaluated model with the paper title and explicitly asked whether it had familiarity with or knowledge of that specific paper and its experimental findings. Any paper that was flagged as familiar by any model was excluded from consideration and removed from the evaluation. In addition questions are not asking for facts directly reported in paper abstracts or introductions, but rather require reasoning about specific quantitative outcomes, mechanistic interpretations, or comparative results from within the experimental body of the paper. Even a model with partial exposure to a paper's title or abstract would be unlikely to correctly answer the prediction question without having internalized the detailed results section. Furthermore, web search was disabled for all model evaluations (see Appendix D), explicitly preventing models from retrieving source papers during inference.
>
> > Q2/W1: Statistical robustness of findings and fine-grained conclusions
>
> We want to first acknowledge that running additional statistical replication trials across 15 frontier models, 2 experimental conditions (NBK, BK), 3 question formats, and 3 scientific domains is prohibitively expensive in terms of API costs. Each full evaluation sweep requires thousands of model calls per model. With this constraint in mind, we note that our current results already incorporate 3 independent evaluation runs for all reported accuracy metrics, with means and standard deviations reported throughout (Tables 3–6, Fig. 3–5).
>
> > Q3/W3: Free-form evaluation LLM judge calibration
>
> To validate our rubric-based LLM judging pipeline, we conducted a human verification study on a random sample of 50 free-form questions drawn from the 130 total free-form questions in SciPredict, where three domain-expert human annotators independently rated each response using the same rubric criteria provided to the LLM judges. Inter-annotator agreement was strong at Cohen's κ = 0.82, confirming that the rubric criteria are sufficiently unambiguous for consistent human interpretation. Agreement between each LLM judge and the human annotators was substantial for both Gemini 3 Pro (κ = 0.76) and GPT-5.2 (κ = 0.69), corroborating our earlier finding in Section 3.4 that the two judges produce statistically indistinguishable accuracy scores and providing additional confidence that our free-form evaluation pipeline is a valid proxy for expert human judgment.
>
> > Q4/W4: Human baseline interpretation - accuracy and feasibility ratings
>
> Tables 5 (according to self-reported reliability scores) and 6 (according to human expert reported reliability scores) in Appendix D provide a complete distribution of feasibility raeting. The tables provide accuracy conditioned on feasibility bins for both humans and models. Please see response to Q4/W3 for reviewer Yx9w regarding how feasibility was elicited. When calculating the aggregated scores feasibility ratings were aggregated over all the tasks.
>
> > Q5: Task construction assumptions: what information is “allowed,” and are tasks solvable from provided context?
>
> *Allowed knowledge:* Models use the experimental setup details and background knowledge for predicting the answer. Models can also rely on rely on general scientific priors if they are already in the parametric knowledge. It is just that they can not look up external sources to make the prediction.
>
> *Solvable tasks:* Feasibility was rated on a 1–5 scale measuring whether an outcome can be predicted without physically running the experiment. Table 5 (Appendix D) reports the number of tasks at each feasibility level under human calibration. Tasks rated 4 (somewhat feasible to answer without running the experiment) or 5 (very feasible to answer without running the experiment) are the tasks where experts judged them reliably predictable from the provided context. 107 tasks recived score 4 and 49 tasks recived score 5, totaling 156 "feasible" tasks (38.5% of the benchmark).
>
> > W5: Reproducibility details could be tighter
>
> All prompts used for evaluation are provided in Appendix E. For the final version, we will consolidate all remaining reproducibility details, including exact model version strings, decoding parameters, and post-processing steps, into a centralized, easy-to-follow README in the publicly available code repository.

---

> > ### Author Rebuttal · Reviewer_3n88 · 2026-04-01
> >
> > I appreciate the rebuttal and find that it addresses several of my concerns, especially by adding a human verification study for the free-form judging pipeline and by clarifying the task assumptions and feasibility analysis. These additions strengthen my confidence in the benchmark.
> >
> > That said, my concerns are only partially resolved. In particular, the new evidence on contamination control is helpful, but it still appears to rely mainly on model self-reported familiarity rather than an external audit. In addition, the response on statistical robustness mainly explains the practical cost of further analysis, rather than providing stronger uncertainty estimates for the key comparisons.
> >
> > I therefore have two brief follow-up questions:
> >
> > (1) Can the authors report how many candidate papers/items were excluded by the familiarity filtering procedure, and whether this exclusion materially changed the benchmark composition?
> >
> > (2) For the main claims (e.g., MCQ vs. free-form, NBK vs. BK, and model vs. human comparisons), can the authors provide at least a lightweight uncertainty analysis, such as bootstrap confidence intervals or paired significance tests on the evaluated outputs?

---

> > > ### Author Response · Authors · 2026-04-03
> > >
> > > > Q1: Effect of familiarity filtering procedure
> > >
> > > Approximately 450 candidate tasks were excluded due to incompatibility with task requirements. Out of these, only 16 were removed specifically through the familiarity filtering procedure. This exclusion had no material effect on benchmark composition. We targeted a fixed domain distribution, 25% physics, 25% chemistry, and 50% biology, and papers were collected to meet this distribution regardless of which individual items were filtered out. The target distribution was chosen to ensure sufficient coverage within each domain and to reflect the relative availability of qualified domain experts across fields.
> > >
> > > > Q2: Uncertainty analysis on results
> > >
> > > We provide results on paired comparison using non-parametric bootstrap resampling (10,000 samples, task-level). Task pairs are resampled jointly and 95% CIs are taken from the 2.5th/97.5th percentiles.
> > >
> > > *BK vs NBK Difference — Overall Accuracy (%)*
> > >
> > > | Model | BK | NBK | Diff | 95% CI |
> > > |---|---|---|---|---|
> > > | Gemini 3-pro | 27.33 | 25.27 | +2.06 | [−0.49, +4.69] |
> > > | Claude Opus 4.5 | 27.33 | 23.05 | +4.28 | [+1.81, +6.91] |
> > > | Claude Sonnet 4.5 | 26.91 | 22.55 | +4.36 | [+1.48, +7.24] |
> > > | Claude Opus 4.1 | 25.43 | 22.22 | +3.21 | [+0.49, +6.09] |
> > > | Gemini 3-Flash | 23.95 | 22.22 | +1.73 | [−0.99, +4.53] |
> > > | OpenAI GPT-5.2 | 22.80 | 20.58 | +2.22 | [−0.49, +5.02] |
> > > | Human Baseline | 20.99 | 20.25 | +0.74 | [−0.74, +2.47] |
> > > | OpenAI O3-mini | 21.40 | 19.84 | +1.56 | [−0.99, +4.20] |
> > > | DeepSeek v3 | 21.98 | 19.18 | +2.80 | [+0.24, +5.43] |
> > > | Llama 3.3 70B | 19.84 | 18.19 | +1.65 | [−0.16, +3.46] |
> > > | OpenAI O3 | 22.14 | 17.94 | +4.20 | [+1.23, +7.24] |
> > > | Qwen 3 32B | 19.09 | 17.04 | +2.06 | [−0.25, +4.44] |
> > > | Gemini 2.5-pro | 22.88 | 17.04 | +5.84 | [+3.05, +8.72] |
> > > | Qwen 3 235B | 20.33 | 16.63 | +3.70 | [+1.07, +6.42] |
> > > | OpenAI O4-mini | 20.16 | 16.21 | +3.95 | [+1.32, +6.67] |
> > > | Llama 3.1 8B | 15.80 | 14.65 | +1.15 | [−1.56, +3.79] |
> > >
> > > All differences are positive; 10/16 CIs lie entirely above zero. Even where CIs cross zero, differences are predominantly positive and the CI mass lies largely in the positive range, providing directional support for the claim that expert background knowledge consistently and reliably improves accuracy.
> > >
> > > ---
> > >
> > > *MCQ vs Free-form (MCQ→FF−MCQ) Difference (%)*
> > >
> > > | Model | MCQ→FF | MCQ | Diff | 95% CI |
> > > |---|---|---|---|---|
> > > | Gemini 3-pro | 24.69 | 36.21 | −11.52 | [−18.93, −4.12] |
> > > | Claude Opus 4.5 | 19.75 | 33.54 | −13.79 | [−21.40, −5.76] |
> > > | Claude Sonnet 4.5 | 20.99 | 29.01 | −8.02 | [−15.84, +0.21] |
> > > | Claude Opus 4.1 | 18.52 | 29.01 | −10.49 | [−17.49, −3.50] |
> > > | Gemini 3-Flash | 25.31 | 29.22 | −3.91 | [−11.32, +3.70] |
> > > | OpenAI GPT-5.2 | 25.93 | 28.81 | −2.88 | [−11.32, +5.76] |
> > > | OpenAI O3-mini | 20.99 | 26.54 | −5.56 | [−12.55, +1.23] |
> > > | DeepSeek v3 | 14.81 | 23.66 | −8.85 | [−16.05, −1.65] |
> > > | Llama 3.3 70B | 11.11 | 26.75 | −15.64 | [−23.05, −8.02] |
> > > | OpenAI O3 | 23.46 | 21.40 | +2.06 | [−4.53, +8.64] |
> > > | Qwen 3 32B | 14.20 | 22.43 | −8.23 | [−15.43, −0.82] |
> > > | Gemini 2.5-pro | 18.52 | 21.40 | −2.88 | [−9.06, +3.70] |
> > > | Qwen 3 235B | 18.52 | 19.75 | −1.23 | [−8.23, +5.76] |
> > > | OpenAI O4-mini | 20.99 | 22.43 | −1.44 | [−8.64, +5.76] |
> > > | Llama 3.1 8B | 9.88 | 21.81 | −11.93 | [−17.90, −5.97] |
> > >
> > > All differences are negative. 10/15 CIs lie entirely below zero. Even where CIs cross zero, differences are predominantly negative and the CI mass lies largely in the negative range, providing directional support for the claim that removing answer options from MCQs degrades accuracy, confirming that free-form questions are more difficult compared to MCQ versions.
> > >
> > > ---
> > >
> > > *Model−Human Difference — Overall Accuracy (%)(Human: 20.25% NBK, 20.99% BK)*
> > >
> > > | Model | Diff (NBK) | 95% CI (NBK) | Diff (BK) | 95% CI (BK) |
> > > |---|---|---|---|---|
> > > | Gemini 3-pro | +5.02 | [+0.25, +9.55] | +6.34 | [+1.23, +11.11] |
> > > | Claude Opus 4.5 | +2.80 | [−2.06, +7.41] | +6.34 | [+1.40, +11.19] |
> > > | Claude Sonnet 4.5 | +2.30 | [−2.63, +7.08] | +5.93 | [+0.91, +10.95] |
> > > | Claude Opus 4.1 | +1.98 | [−2.72, +6.58] | +4.44 | [−0.49, +9.30] |
> > > | Gemini 3-Flash | +1.98 | [−2.88, +6.67] | +2.96 | [−1.98, +7.82] |
> > > | OpenAI GPT-5.2 | +0.33 | [−4.44, +4.94] | +1.81 | [−3.05, +6.50] |
> > > | OpenAI O3-mini | −0.41 | [−5.02, +4.20] | +0.41 | [−4.12, +4.86] |
> > > | DeepSeek v3 | −1.07 | [−5.43, +3.21] | +0.99 | [−3.62, +5.43] |
> > > | Llama 3.3 70B | −2.06 | [−6.50, +2.39] | −1.15 | [−5.68, +3.37] |
> > > | OpenAI O3 | −2.30 | [−6.67, +1.90] | +1.15 | [−3.29, +5.60] |
> > > | Qwen 3 32B | −3.21 | [−7.74, +1.23] | −1.89 | [−6.50, +2.55] |
> > > | Gemini 2.5-pro | −3.21 | [−7.57, +0.99] | +1.89 | [−2.63, +6.34] |
> > > | Qwen 3 235B | −3.62 | [−8.23, +0.74] | −0.66 | [−5.27, +3.87] |
> > > | OpenAI O4-mini | −4.03 | [−8.48, +0.16] | −0.82 | [−5.19, +3.46] |
> > > | Llama 3.1 8B | −5.60 | [−10.04, −1.23] | −5.19 | [−9.55, −0.99] |
> > >
> > > CIs are distributed across positive and negative ranges supporting the claim that human performance is close to average models performance..

---

### Official Review · Reviewer_Heba · 2026-03-09

**Soundness:** 3
**Presentation:** 4
**Significance:** 3
**Originality:** 4
**Overall Recommendation:** 5
**Confidence:** 4

**Summary:**

The authors introduce SciPredict, a benchmark of recent experimental findings in physics chemistry and biology.  This benchmark is proposed as a challenging measure of LLMs ability to predict the outcome of scientific experiments.  The authors create a new benchmark based on time-censored scientific articles and questions in various formats (multiple choice, free form, and numeric) with scoring for each. In evaluating recent LLMs against the benchmark, they found that while frontier models were near or above human level performance, the models differed from human experts in that their assessed confidence was not correlated with their prediction accuracy.

**Compliance With Llm Reviewing Policy:**

Affirmed.

**Final Justification:**

The authors have addressed my questions, which reinforced my assessment at Accept.  The merits of the work outweigh the limitations, in my view.  The work's original approach and significant finding is worth communicating to a broader audience, despite some shortcomings in terms of the scale of the human benchmarking, risk of test set contamination, and the large degree of maintenance required to keep the benchmark up-to-date.

**Key Questions For Authors:**

Q1: How many human experts participated in the baseline?  Perhaps I missed it, from the statistics in the appendix a number of at least 39 could be inferred.
Q2:  The poor performance of the Human baseline on the chemistry questions (Fig 9) is concerning; are there ideas for why this could be happening?
Q3: The LLMs were limited by their knowledge cutoff date, but the human baseline participants may have been familiar with the articles before the assessment, how was this controlled?

**Limitations:**

An important limitation that the authors did not address is the bias in publications toward novel or surprising results.  Experiments with negative findings are often underrepresented in the literature.  Models that perform well on SciPredict may therefore be more prone to predicting more “publishable” results of experiments when the real outcome may be more mundane.

**Strengths And Weaknesses:**

The paper is an ambitious work to test LLMs performance in an important practical area: scientific understanding.  The work is technically sound in that the authors make an effort to curate a diverse set of questions from a time based snapshot of the scientific literature.  The work goes further to explore hypotheses related to the types of background knowledge that improve LLM performance on the benchmark, failure modes, question format, and correlation to the HLE benchmark.  There are some details like randomizing MCQ question order, or controlling for the accuracy rate of random guessing that could have been explored.  The presentation of the write up was well structured and easy to follow.  The paper has significance for the broader AI for Science community.  While there are other well-known benchmarks on scientific knowledge, notably HLE, this SciPredict focuses on experimental predictions and self-assessed confidence in those predictions.  As a new benchmark, SciPredict is an original and valuable contribution to model benchmarking, though with will need to be continually updated.  The findings on how the LLMs fail, notably their sensitivity to the types of background knowledge provided is valuable information for researchers that I have not seen published previously.

---

> ### Author Rebuttal · Authors · 2026-03-31
>
> We thank the reviewer Heba for the valuable review. We are glad that the reviewer found the work ambitious, technically sound, and well-structured, and appreciated the novelty of focusing on experimental outcome prediction and calibration. We also appreciate the recognition of the benchmark’s breadth (e.g., background knowledge analysis, failure modes, and format sensitivity) and its relevance to the AI-for-Science community.
>
> > Q1: How many human experts participated in the baseline?
>
> 37 experts participated in the baseline experiments. 74.4% hold PhDs, with the rest holding Master’s or Bachelor’s degrees. They are distributed across domains (≈48.7% biology, 33.3% chemistry, 17.9% physics).
>
> > Q2: Conern regrading performance of the human baseline on chemistry questions
>
> Chemistry tasks involved numerical and physical chemistry questions where exact quantitative intuition is required, rather than directional or qualitative predictions. This raises the bar for human experts compared to, biology and physics tasks that more often admit qualitative reasoning. However, we did not see a significant performance degradation for LLMs on these tasks.
>
> > Q3: How was prior familiarity of human participants with the source papers controlled?
>
> We took several steps to mitigate this risk. Human baseline participants were an entirely separate cohort from the benchmark constructors, who had no visibility into which papers were selected. We also implemented two direct controls (1) participants were explicitly asked to self-report their familiarity with each source paper, allowing us to flag and account for any prior exposure; and (2) we checked whether response wording was overly similar to phrasing in the source papers, which would be a direct signal of familiarity with the original work. In addition, the ~20% human accuracy, well below what one would expect from someone who had read the results, further corroborates that prior familiarity was not a meaningful confound.

---

> > ### Author Rebuttal · Reviewer_Heba · 2026-04-01
> >
> > The authors have addressed my questions, thank you for your submission.

---

### Official Review · Reviewer_6Y3s · 2026-03-12

**Soundness:** 3
**Presentation:** 3
**Significance:** 3
**Originality:** 3
**Overall Recommendation:** 5
**Confidence:** 4

**Summary:**

This paper introduces SciPredict, a novel benchmark designed to evaluate the ability of LLMs to predict the outcomes of scientific experiments in the natural sciences. The main goal of this paper is to answer two questions: 1) Can LLMs accurately predict the results of scientific experiments? 2) Can such predictions be reliably used in the scientific research process? Through extensive experiments and evaluations, the benchmark shows that some LLMs can achieve higher prediction accuracy

**Compliance With Llm Reviewing Policy:**

Affirmed.

**Final Justification:**

My concerns are resolved in the rebuttal.

**Key Questions For Authors:**

See weaknesses.

**Limitations:**

yes

**Strengths And Weaknesses:**

Strengths:
- The paper is well-written, exhibiting clear logic and well-defined objectives.
- The benchmark involved over 7,380 human expert hours and cost $336k to build, ensuring quality control and realistic experimental setups. Enormous experiments are conducted with human comparison.
- The main claims are well supported by analysis. The benchmark shows that experimental science requires not just better prediction accuracy, but better awareness of prediction reliability.

Weaknesses:

1. A major concern of this benchmark is its scalability. First, it requires extensive manual labeling by experts, which is time-consuming. Furthermore, the iteration of LLMs is exceptionally rapid; papers used to construct benchmarks are often incorporated into the training data for newer models. This risk of data contamination makes it difficult for the benchmark to accurately evaluate future LLMs.

2. All LLMs used in the experiments are general-purpose models. However, the scientific field contains many domain-specific models or specialized enhanced techniques[1,2,3]. It is worth investigating whether the domain-specific models similarly fail to produce reliable predictions.

1.  While the paper identifies the calibration gap, it does not discuss or propose any methods to mitigate it. Discussing and proposing a solution would be more beneficial for promoting the effective use of LLMs in scientific research.

[1] Zhao, Zihan, et al. "Chemdfm: A large language foundation model for chemistry." *Neurips 2024 Workshop Foundation Models for Science: Progress, Opportunities, and Challenges*. 2024.

[2] Luo, Yizhen, et al. "Biomedgpt: An open multimodal large language model for biomedicine." *IEEE Journal of Biomedical and Health Informatics* (2024).

[3] Tang, Xiangru, et al. "Chemagent: Self-updating library in large language models improves chemical reasoning." *arXiv preprint arXiv:2501.06590* (2025).

---

> ### Author Rebuttal · Authors · 2026-03-31
>
> We thank the reviewr 6Y3s for a thoughtful and positive review. We are encouraged that the reviewer found the paper to be well-written with clear objectives, appreciated the scale and rigor of the benchmark construction (7,380 expert hours, $336k), and valued the central finding that reliable scientific prediction requires not only accuracy but also calibration of uncertainty.
>
> > W1: Concern on scalability and risk of data contamination
>
> Training data contamination is a very valid concern and effects even the flagship evals (SWE-Bench) used across all frontier models (https://openai.com/index/why-we-no-longer-evaluate-swe-bench-verified). One common approach established across frontier labs is continuosly decontaminate the training data based on evals. We intend to publish the list of all papers used in our eval such that these can be used to sanitize the training data.
>
>
> > W2: Performance of domain-specialized LLMs
>
> This is a valuable next step to our work. Our focus on general-purpose frontier models reflects their increasing adoption as scientific assistants and establishes a clear baseline. Evaluating domain-specific models such as ChemDFM, BioMedGPT, or tool-augmented systems like ChemAgent is a natural extension. We hypothesize that domain-specific models may improve raw accuracy in their respective fields, but the calibration gap — which is our central finding — may persist independently of domain specialization. We will add explicit discussion of this limitation and include evaluation results in the next version of the paper.
>
> > W3: Methods for mitigating calibration gap
>
> We agree that proposing mitigations would add value, and we discuss a few promising directions below.
> - Post-hoc calibration techniques, such as temperature scaling or conformal prediction, could be applied to align self-reported reliability scores with empirical accuracy.
> - The strong calibration demonstrated by human experts suggests a promising fine-tuning direction: training models on examples where human-rated feasibility and difficulty scores are paired with ground-truth outcomes. SciPredict directly provides this data, and supervised fine-tuning on such signal could teach models to better distinguish tasks they can predict reliably from those they cannot.
> - Ensemble-based uncertainty quantification, where disagreement across multiple model runs serves as a calibration signal, represents a practical near-term approach.
>
> Crucially, SciPredict provides the measurement framework necessary to evaluate such mitigations, a contribution we consider foundational to progress on this problem. We will expand the discussion section accordingly.

---

> > ### Author Rebuttal · Reviewer_6Y3s · 2026-04-02
> >
> > Thanks to the author for resolving my concerns. I will raise my score accordingly.

---

### Official Review · Reviewer_Yx9w · 2026-03-25

**Soundness:** 2
**Presentation:** 3
**Significance:** 3
**Originality:** 3
**Overall Recommendation:** 4
**Confidence:** 4

**Summary:**

This paper introduces SciPredict, a benchmark for evaluating whether LLMs can predict the outcomes of scientific experiments from experimental setups, measurements, and optional background knowledge. The benchmark contains 405 tasks across physics, chemistry, and biology, covering multiple answer formats including multiple-choice, free-form, and numerical prediction. The paper evaluates 15 frontier LLMs and human experts under different background-knowledge settings. The main findings are that current models still achieve limited performance overall, expert-provided background knowledge helps more than self-generated background knowledge, and model self-assessments of confidence / difficulty / feasibility are poorly calibrated relative to true accuracy.

**Compliance With Llm Reviewing Policy:**

Affirmed.

**Final Justification:**

This paper studies an important and underexplored problem: whether LLMs can predict the outcomes of real scientific experiments rather than only answer scientific questions. Its main contribution is SciPredict, a benchmark covering 405 expert-curated tasks across physics, chemistry, and biology, and I find this benchmark framing both meaningful and potentially useful for future work on scientific reasoning and calibration. The paper is generally clear, and my original concerns were mainly about methodological transparency, especially the mapping between source papers and tasks, the expert curation pipeline, and the exact elicitation/evaluation protocol. The rebuttal addressed these concerns substantially by clarifying the paper-to-task mapping, providing much more detail on expert coverage and review, and making the reliability elicitation setup explicit. This improved my confidence in the benchmark and changed my evaluation in a positive direction. Overall, after considering both the paper and the rebuttal, I view this as a technically solid benchmark paper with meaningful potential impact, although I still think some of the clarified methodological details should be integrated more explicitly into the final version.

**Key Questions For Authors:**

1. How many source papers were used in total, and what is the exact mapping between papers and benchmark tasks?
2. How many experts contributed to benchmark construction, and how are they distributed across disciplines and subfields?
3. Do you have any inter-annotator / inter-reviewer agreement or other reliability statistics for extracted background knowledge, free-form rubrics, and numerical answer ranges?
4. Please make the elicitation format for confidence / difficulty / feasibility fully explicit. How sensitive are these conclusions to prompt wording or response format?
5. Since MCQ is less realistic than free-form or numerical prediction, how should readers interpret overall benchmark performance as evidence of real scientific predictive ability?

**Limitations:**

Yes, but only partially.

The paper discusses some limitations, but I think it should more explicitly acknowledge uncertainty around annotation reliability, paper-to-task mapping, and the prompt dependence of the reliability analysis.

**Strengths And Weaknesses:**

**Strengths**

The paper studies an important and practical problem. Predicting experimental outcomes is a meaningful capability beyond standard scientific QA, and the benchmark framing is well motivated.

The benchmark has reasonably broad scope, spanning three major scientific disciplines and multiple subfields, and the paper evaluates a fairly broad set of frontier LLMs. This gives the benchmark potential value as a community resource.

I also appreciate that the paper goes beyond raw accuracy and studies self-reported confidence, difficulty, and feasibility. This is a useful angle for scientific-assistance settings, where knowing when a model is unreliable matters.

The paper is generally clear and easy to follow. The task setup and main empirical takeaways are presented clearly.

**Weakness**：

My main concerns are about benchmark construction and methodological transparency.

First, the paper does not clearly document the mapping between source papers and benchmark tasks. It is unclear how many source papers were used, whether a paper contributes one or multiple tasks, and how redundancy or dependence across tasks is controlled. This makes it difficult to judge benchmark diversity and sample independence.

Second, although the paper emphasizes expert curation and multi-stage review, it does not report enough information about the annotation pipeline. I could not find clear details on annotator counts, subfield coverage, whether multiple experts independently processed the same task, or any inter-annotator / inter-reviewer agreement statistics. Since the benchmark relies on expert extraction of background knowledge, rubrics, and numerical target ranges, these details are important.

Third, several parts of the evaluation depend on subjective or prompt-sensitive choices. Free-form responses are evaluated via rubric-based LLM judgment, numerical answers use expert-defined acceptable ranges, and the reliability analysis depends on prompted self-reports of confidence / difficulty / feasibility. These choices are reasonable in principle, but the paper does not provide enough evidence that the conclusions are robust to them.

Finally, while the three answer formats are practically useful, the MCQ setting is somewhat removed from realistic scientific prediction, which should temper the strength of claims about real-world predictive usefulness.

---

> ### Author Rebuttal · Authors · 2026-03-31
>
> We thank the reviewer for the thoughtful evaluation and are encouraged by the recognition of the problem importance, benchmark scope, calibration-focused analysis.
>
> > Q1/W1: Mapping between source papers and benchmark tasks?
>
> Our benchmark draws tasks from 397 unique papers in total. Expert contributors were permitted to extract 1–3 tasks per paper, resulting in: 390 papers contributed exactly one task, 6 papers contributed two tasks each, and 1 paper contributed three tasks. In cases where multiple tasks derive from the same paper, each task corresponds to a distinct sub-experiment, ensuring all 405 tasks represent independent prediction challenges.
>
> > Q2: Experts count and expertise distribution?
>
> 266 domain experts contributed to benchmark construction. As detailed in Appendix B.1, their expertise is distributed across the three core domains: 38% biology, 32% physics, and 30% chemistry — broadly mirroring the benchmark's task distribution (50% biology, 25% physics, 25% chemistry), with biology receiving greater representation due to its larger number of subfields (14 vs. 9 in physics and 10 in chemistry). At the subfield level, expert expertise distribution approximately matches the fine-grained task subfield counts in Appendix B.7, Table 2, ensuring each subfield is covered by contributors with relevant specialized knowledge.
>
> > Q3/W2: Reliability of annotations pipeline, particularly rater agreement.
>
> Each task underwent a multi-stage review process involving multiple experts. Following initial curation, every task was independently verified by two additional domain expert reviewers. Any task flagged during this review — covering experimental setup, ground truth answer, background knowledge, free-form rubrics, and numerical target ranges — was returned to the original curator for revision. Task curators and reviewers then iterated until consensus was reached; tasks without consensus were discarded. Beyond this, a final quality control team conducted an independent audit of all retained tasks. We will expand Section 2.3 and Appendix B.5 to more clearly document annotator counts, subfield coverage, and the consensus procedure.
>
> > Q4/W3: Sensitivity of final conclusions to prompt wording or response format?
>
> *Elicitation format for confidence / difficulty / feasibility.* The three reliability metrics were elicited using the following explicit 5-point scales, provided to models as structured response options.
>
> - `confidence`: Options are very unsure of the answer, somewhat unsure of the answer, neither unsure of, nor confident in the answer, somewhat confident in the answer, very confident in the answer
>
> - `difficulty`: Options are very easy to answer, easy to answer, neither easy nor difficult to answer, difficult to answer, very difficult to answer
> - `feasibility`: Options are completely infeasible to answer without running the experiment, somewhat infeasible to answer without running the experiment, neither infeasible nor feasible to answer without running the experiment, somewhat feasible to answer without running the experiment, very feasible to answer without running the experiment
>
> These options were provided as a structured response schema. We additionally ran experiments where the same descriptions were included as explicit in-prompt text rather than a schema, and observed no meaningful difference in the resulting distributions or calibration conclusions. This provides direct evidence that our findings are not sensitive to elicitation format.
>
> *Robustness of free-form evaluation.* We validated our LLM-judge pipeline by replicating evaluations using both Gemini 3 Pro and GPT-5.2 as judges, finding no statistically significant differences in accuracy scores.
>
> *Robustness of numerical ranges.* Acceptable ranges followed a principled hierarchy: where source papers reported explicit uncertainty ranges, these were used directly, eliminating subjectivity for a significant proportion of numerical tasks. Where papers did not report ranges, domain experts defined intervals based on field-specific measurement norms, with a conservative ±10% fallback applied in remaining cases. All expert-defined ranges underwent two independent review rounds to ensure they were neither too narrow nor trivially broad.
>
> > Q5/W4: Interpreting MCQ results as it's less realistic scienitific setting that other setups.
>
> Finding #7 and Figure 8 show that converting MCQs to matched free-form prompts (MCQ→FF) — identical scenarios, no answer options — causes accuracy to drop consistently across all models, explicitly cautioning against over-interpreting MCQ scores. Furthermore, SciPredict is not MCQ-dominated: 60% of tasks are free-form or numerical, and our central claims about real-world predictive usefulness — particularly the calibration failure — hold uniformly across all formats regardless of MCQ. We will add a clarifying note directing readers to the MCQ→FF results when interpreting aggregate performance.

---

> > ### Author Rebuttal · Reviewer_Yx9w · 2026-04-04
> >
> > The rebuttal adequately addresses my main concerns. It clarifies the paper-to-task mapping (397 unique papers for 405 tasks), provides much more detail on the expert curation pipeline (266 domain experts, two additional expert reviewers per task, consensus-based revision/filtering, and final QC audit), and makes the elicitation format for confidence/difficulty/feasibility explicit. It also provides additional support for the robustness of the free-form and numerical evaluation setup, and clarifies that the main conclusions should not be over-interpreted from MCQ results alone. These points substantially improve my confidence in the benchmark construction and evaluation protocol. I still think these details should be clearly integrated into the final paper, but I no longer view them as blocking concerns.

---

### Decision · Program_Chairs · 2026-04-30

**Decision:**

Accept (regular)

**Comment:**

All four reviewers recommend acceptance (scores: 4, 5, 5, 5), and the authors provided thorough rebuttals that resolved the major concerns raised during review. SciPredict introduces a well-motivated benchmark of 405 expert-curated tasks evaluating whether LLMs can predict outcomes of real scientific experiments across physics, chemistry, and biology. The benchmark construction involved 266 domain experts and significant investment in quality control. The central finding is that LLMs lack calibration awareness, achieving similar accuracy regardless of self-reported confidence, while human experts show strong calibration. This is novel and practically important for AI-for-science applications. Reviewers appreciated the breadth of the evaluation (15 frontier models plus human baselines), the multiple answer formats, and the analyses beyond raw accuracy (confidence, difficulty, feasibility). Initial concerns around benchmark construction transparency, free-form evaluation validity, and statistical robustness were addressed in the rebuttal with concrete evidence, including inter-annotator agreement statistics, bootstrap confidence intervals, and LLM-judge validation against human annotators.
Remaining limitations are reasonable for a benchmark paper of this scope: the 405-task size limits fine-grained subfield conclusions, contamination control relies partly on model self-reported familiarity, and the benchmark will require ongoing maintenance. These are acknowledged and do not undermine the core contribution. The authors should integrate the methodological clarifications from the rebuttal into the camera-ready version.